# Paleontological and paleoecological significance of the oldest highly productive Upper Cretaceous (lowermost Maastrichtian) bonebed of Haţeg Basin (western Romania; Densuş-Ciula Formation)

Gábor Botfalvai[1,2☯*], Zoltán Csiki-Sava[1,3☯], János Magyar[1,4], Barna Páll-Gergely[5], Levente Koczó[1], Daniel Ţabără[6], Gergő Konecsni[1], Soma Budai[7]

1 Institute of Geography and Earth Sciences, Department of Palaeontology, ELTE Eötvös Loránd University, Budapest, Hungary, 2 HUN-REN–MTM–ELTE Research Group for Paleontology, Budapest, Hungary, 3 Department of Geology, Mineralogy and Palaeontology, University of Bucharest, Bucharest, Romania, 4 Department of Geology and Paleontology, Hungarian National Museum Public Collection Centre, Hungarian Natural History Museum, Budapest, Hungary, 5 Plant Protection Institute, HUN-REN Centre for Agricultural Research, Martonvásár, Hungary, 6 Department of Geology, Al. I. Cuza University of Iaşi, Iaşi, Romania, 7 Dipartimento di Scienze della Terra e dell'Ambiente, Università degli Studi di Pavia, Pavia, Italy

☯ These authors contributed equally to this work
* botfalvai@staff.elte.hu

## Abstract

Recent extensive fieldwork in the Densuş-Ciula Formation in Haţeg Basin has led to the discovery of several important high-diversity bonebeds. Among the excavated locations, site K2 is by far the most significant, as based on its stratigraphical position it is considered the oldest known (earliest Maastrichtian) highly diversified vertebrate site in the entire Haţeg Basin, and thus provides a good starting point for paleofaunistic, paleoecological and biostratigraphic comparisons with other similar sites across the Transylvanian area. During this study, detailed sedimentological, palynological, invertebrate- and vertebrate paleontological investigations were conducted to reconstruct the former paleoenvironment and the different depositional processes that allowed the formation of this productive bonebed. More than 800 vertebrate fossils were collected from an approximately 4.75 m² area of the bonebed horizon of site K2 representing at least 17 species including fish, amphibians, turtles, squamates, crocodyliforms, dinosaurs, pterosaurs and mammals, ranking this site among the most taxonomically diverse ones within the basin. The sedimentological investigation points towards a lacustrine depositional environment in which a high-diversity, multitaxic, multidominant mixed assemblage was accumulated on a flood-related delta due to a sudden drop in transport energy. Based on its stratigraphical position, site K2 represents the oldest vertebrate site within the Haţeg area and suggests a remarkable large-scale faunal stability on the Haţeg Island during the Maastrichtian.

**Data availability statement:** All data are in the manuscript and/or supporting information files.

**Funding:** Financial support for this project was provided by Hungarian Scientific Research Fund and National Research, Development and Innovation Office (Grants no. NKFIH OTKA FK 146097 awarded to G.B.) and Romanian Ministry of Research, Innovation and Digitization (Grants no. PN-III-P4-ID-PCE-2020-2570, within PNCDI III awarded to Z.Cs.-S.). The funders had no role in study design, data collection and analysis, decision to publish, or preparation of the manuscript.

**Competing interests:** All authors have read and approved the manuscript, and have no conflicts of interest to declare.

The dominant elements of the local fauna were already present in the earliest Maastrichtian, and no significant differences in faunal composition can be detected between this oldest and other, younger vertebrate assemblages of Haţeg Basin, at least at the level of higher taxa. Furthermore, just as the faunal composition, the dominance spectrum of the different taxa has not changed significantly among the Maastrichtian sites of Haţeg Basin.

## 1. Introduction

Since the beginning of the 20th century, Haţeg Basin has been known for its well-preserved Late Cretaceous vertebrate fossils. The more than century long research efforts in the area provided valuable material for a deeper understanding of the paleontology, paleoecology, paleobiogeography and paleobiology of Late Cretaceous European vertebrate faunas [1–3 and references therein]. Currently, thousands of vertebrate fossils are known from dozens of sites spread across Haţeg Basin [3]; however, a large part of this extensive fossil material resulted from chance discoveries of sporadic occurrences often yielding isolated specimens [4,5]. Thus, due to the lack of systematic fossil collecting and associated careful facies analysis, the detailed sedimentology and stratigraphy of these sites as well as the precise faunal content/ diversity of the investigated bone-bearing beds remained poorly understood.

This is particularly the case for the vertebrate-bearing sites located in the Densuş-Ciula Formation cropping out in the northwestern part of the basin, where, apart from the Tuştea nesting site [6–8], systematic vertebrate excavations were only carried out at the beginning of the last century [9]. Thus, the stratigraphic and sedimentological context of the fossiliferous occurrences described from this formation is not yet understood in great details [5,10–12]. Since the late 1990s and 2000s, an extensive survey for microvertebrates has been carried out in different parts of the Densuş-Ciula Formation, leading to the identification of several important localities and providing a large quantity of microvertebrate remains [3,13–19]. These results highlighted an outstanding faunal diversity within the unit and indicated a high paleontological potential hence justifying a more comprehensive study in the area. However, in contrast to the growing number of microvertebrate sites, larger-scale excavations of important macrovertebrate accumulations (i.e., bonebeds) in the Densuş-Ciula Formation have been largely limited to the Tuştea locality.

To address these gaps in paleontological, sedimentological and stratigraphical knowledge, since 2019 large-scale and systematic excavations were carried out in the northwesternmost part of Haţeg Basin in the vicinity of Vălioara village (Fig 1). During this multi-annual fieldwork several potential locations were investigated, resulting in the identification of important fossil accumulations and the collection of hundreds of vertebrate remains. Initially the aim of this fieldwork was to rediscover and document the historical fossil sites from which an important vertebrate material had been collected by Ottokár Kadić in the early 20th century [11]. However, during the rediscovery of these historical sites, several other new fossiliferous outcrops

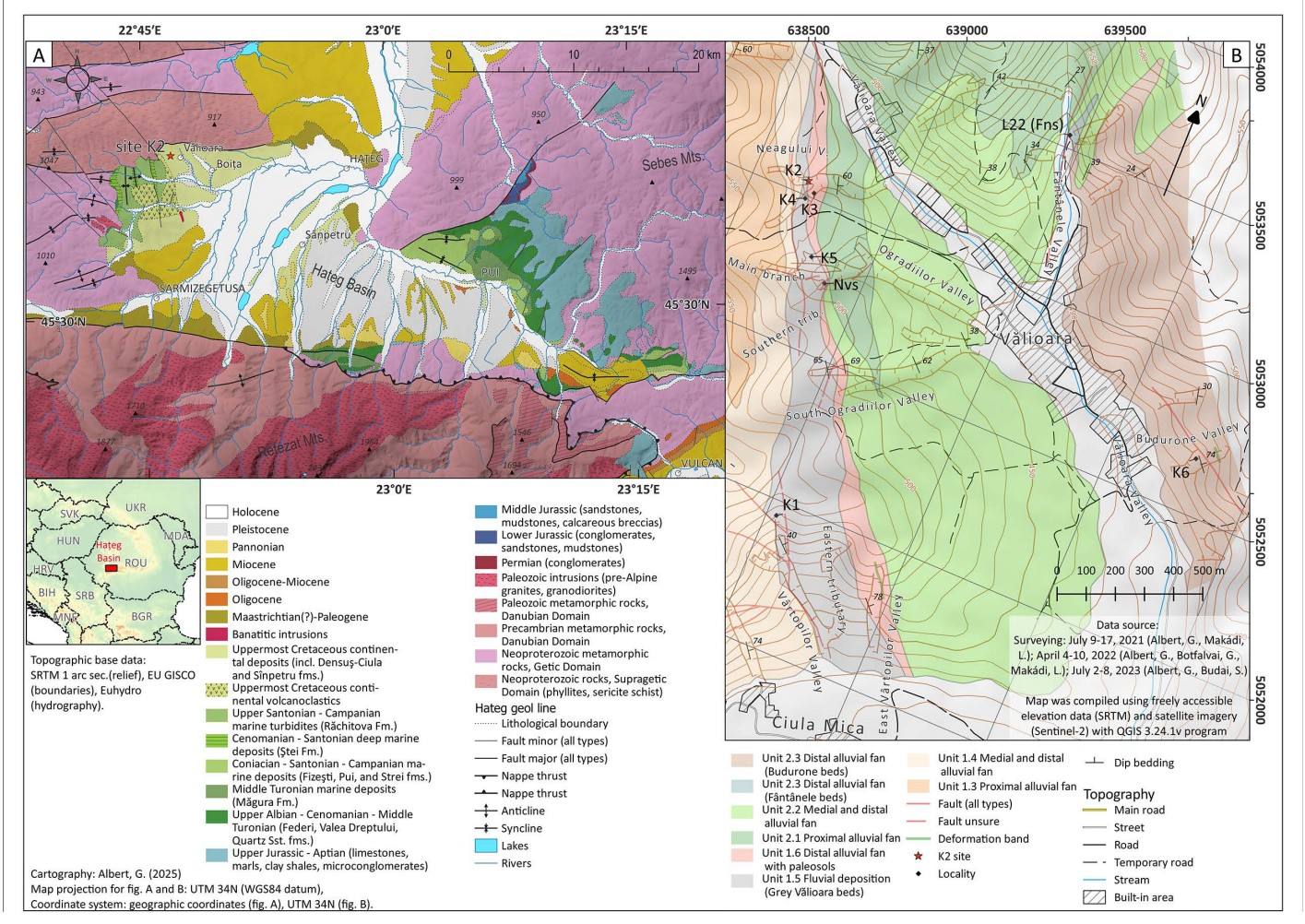

**Fig 1. Geological maps. (A)** Geologic map of Haţeg Basin (based on [20] and references therein). **(B)** Detailed geological map of the Vălioara area showing the main vertebrate sites (based on [20]. Base maps were compiled using openly licensed (CC BY 4.0 or public domain) datasets: Sentinel-2 (Copernicus, 2023.01.08), SRTM (NASA/JPL-Caltech), EU GISCO (Eurostat), and EU-Hydro (Copernicus/EEA).

were identified, in which subsequent excavations were accompanied by thorough sedimentological, taphonomical, and structural geological analyses [11,12,20]. These investigations revealed the detailed sedimentological context as well as stratigraphic relationships between the different bone-bearing horizons, highlighting important differences in age and depositional environments.

Among the sites excavated during these years in the Vălioara study area, site K2 [11,20] was the most productive and most diverse one (Fig 1B and Fig 2). From its only 0.5 m thick bonebed more than 800 isolated, associated, or even articulated vertebrate remains belonging to at least 17 different taxa were excavated and collected, in addition to significant invertebrate and palynomorph assemblages. As a result of the multi-year research activity carried out at site K2, the fossil vertebrate accumulation identified here ranks among the most diverse and best documented ones from the entire Densuş-Ciula Formation, matching the previously investigated Tuştea-Oltoane site, and as such provides an excellent data point for in-depth paleofaunistic, paleoecological, paleoenvironmental and biostratigraphical comparisons with other important sites known across Haţeg Basin, but also from the entire Transylvanian region.

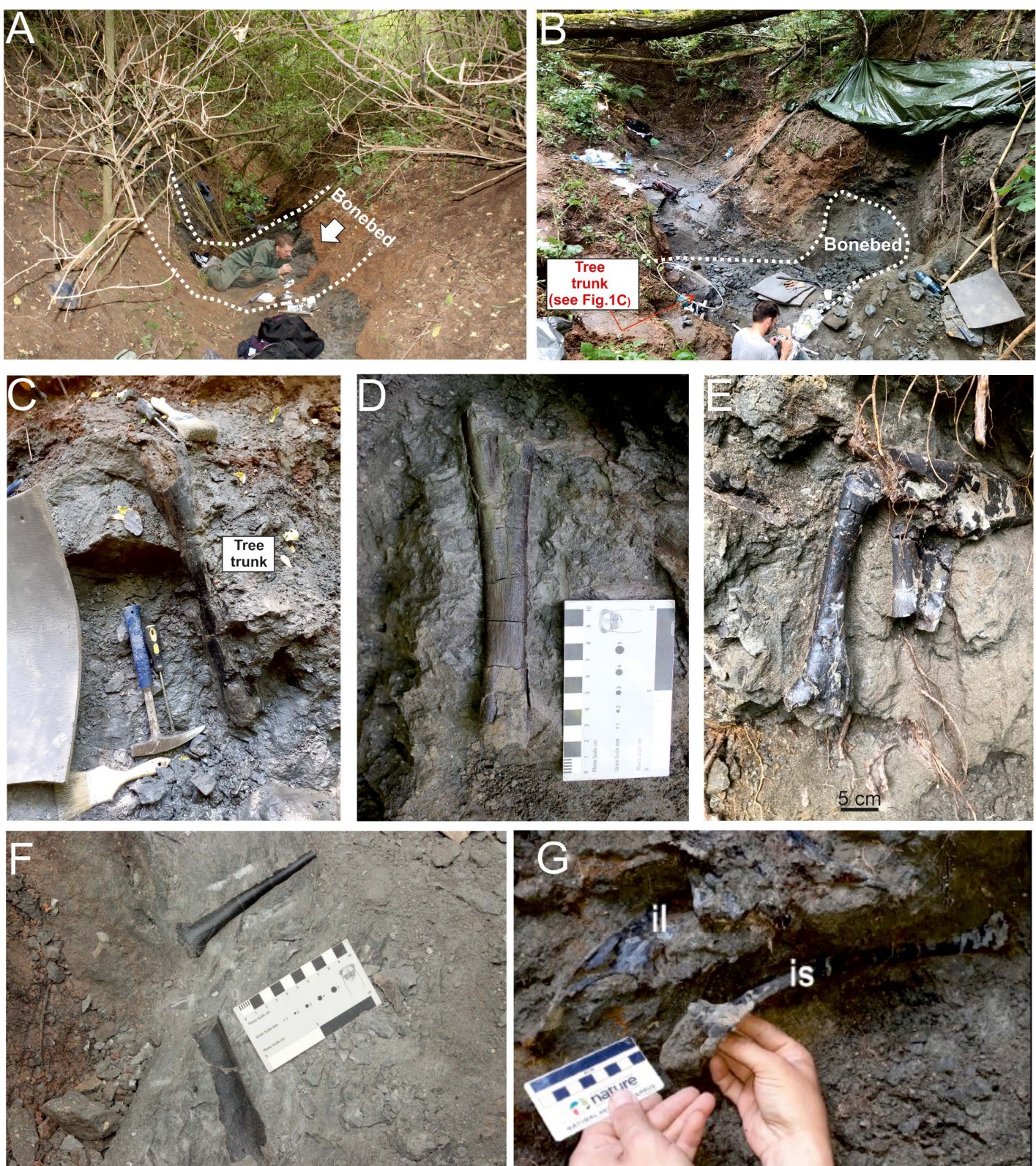

**Fig 2. Site K2, Vălioara (Densuş-Ciula Formation). (A)** Status of the investigated bonebed in 2019 (B) and in 2022. **(C)** Vertically oriented tree trunk in the bonebed horizon of site K2. **(D-F)**. *In situ* well-preserved dinosaur bones at site K2. **(G)**. *In situ* associated pelvic elements of Rhabdodontidae indet. [12].

The aim of this report is to give a detailed taphonomical, paleontological, sedimentological and stratigraphical account of this remarkably rich vertebrate fossil site, with the following specific objectives: (i) to document the vertebrate fossil assemblage collected at this site; (ii) to provide a detailed sedimentological interpretation, allowing the identification of the paleoenvironment within which the bonebed was formed; (iii) to review the palynomorph and invertebrate assemblages recovered here in order to supplement the sedimentological interpretation; (iv) to establish the relative stratigraphic position of the site within the local basin-fill succession, in order to constrain its age as well as to determine its spatial and temporal relationships to other fossiliferous sites, either from the study area or from farther away within Haţeg Basin and the wider Transylvanian area; (v) to reconstruct in details the taphonomic history of the K2 vertebrate assemblage, and evaluate its implications for understanding the composition of the local paleoecosystem(s); and (vi) to assess the relevance of site K2 for interpreting latest Cretaceous faunal composition and evolution in the Transylvanian area.

## 2. Geological setting

Situated in the western part of the Southern Carpathians, southwestern Romania, Haţeg Basin is an intramountain depression whose evolution was shaped by several Late Cretaceous tectonic events that generated the nappe structure of this Carpathian segment. It was formed first as a piggy-back basin developed atop an actively accreting nappe stack and filled up with mainly deep-marine (even turbiditic) Upper Cretaceous (Cenomanian to Campanian) deposits [21–23], after which the tectonic regime shifted to an extensional one from the latest Cretaceous onwards. At that time, concomitantly with localized orogenic collapse and unroofing that affected the already raising Southern Carpathian orogenic segment, Haţeg Basin transitioned towards a transtensional strike-slip basin that accumulated a thick pile of largely continental deposits resulting from the ongoing erosion of the surrounding uplifted basement units [20,23].

These uppermost Cretaceous continental deposits, hosting the different vertebrate-rich fossiliferous localities, have been studied for more than a century [3,24–27]. They are outcropping discontinuously over the northwestern, central and central-eastern parts of the basin, and had been categorized into several lithostratigraphic units that were often regarded as being roughly synchronous, representing the Maastrichtian [3,28]; nevertheless, recently it has been shown that, at least locally, continental sedimentation started significantly earlier than the Maastrichtian [20 and see below]. Overall, the continental beds are built up by sediments deposited in diverse intercalated environments, from proximal alluvial fan deposits to braided channel fills, well- or poorly drained floodplain sediments, with common paleosol horizons, and lacustrine, swampy (coal-bearing) deposits. Volcanoclastic successions are also present, but remain only locally important and well-represented [see review in 3]. Together with isofacial deposits cropping out in the larger Transylvanian Basin to the northeast and the Rusca Montană Basin to the west, these uppermost Cretaceous continental deposits were accumulated within an insular environment, the 'Haţeg Island' [1,25,29], part of the Late Cretaceous European Archipelago [3].

Of the uppermost Cretaceous continental units of Haţeg Basin, the Densuş-Ciula Formation groups deposits that crop out in the northwestern part of the basin, between Ciula Mică and Ştei, in the western part, and Crăguiş in the east (Fig 1A). Previously, both the age/stratigraphic position as well as the lithological make-up and subdivisions of the deposits currently included into the Densuş-Ciula Formation had been contentious [see discussions in 3,18,20,28,30–32] but currently it is thought that this unit encompasses all continentally accumulated (i.e., both sedimentary and volcano-sedimentary/pyroclastic) uppermost Cretaceous rocks from northwestern Haţeg Basin [3,28]. The unit is divided into three vertically superimposed sub-units that are also spatially arrayed from west to east: the oldest mixed volcanoclastic-siliciclastic Lower Member in the westernmost outcropping areas, followed by the siliciclastic Middle Member that hosts both the major vertebrate-bearing localities as well as reworked volcanic material, and, towards the east, the siliciclastic Upper Member that was until two decades ago regarded as being devoid of vertebrate fossils and reworked volcano-clasts. On account of these paleontological and sedimentological characteristics, the latter subunit was once considered to potentially extend into the lower Paleogene as well [28,29], although more recent discoveries have verified both its latest Cretaceous age and its fossiliferous nature [18]. Until very recently, the age of the entire formation (excluding its

controversial uppermost part, see above) had been considered to be Danian (sensu latest Cretaceous; [33]), then late Maastrichtian [28,32], or, subsequently, as encompassing the entire Maastrichtian [3,34]. Nevertheless, new research – discussing both the continental deposits of the formation itself as well as the underlying marine beds – has shown that deposition of the oldest Densuș-Ciula deposits (i.e., the volcano-sedimentary successions) began as early as the middle part of the Campanian [20,25,35].

The focus of this study is represented by the uppermost Cretaceous continental deposits cropping out in the neighbourhood of Vălioara in the northwestern extremity of Haţeg Basin (Fig 1) and belonging to the basal part of the Middle Member of the Densuș-Ciula Formation. Recent mapping combined with multi-proxy investigations [20] has shown that the Densuș-Ciula Formation in this area overlays, on top of an erosional and slightly angular unconformity, the uppermost Santonian to lower Campanian turbiditic beds of the marine Răchitova Formation, and consists of two major fining-upward depositional sequences which both grade, laterally and vertically, from dominantly matrix-supported conglomerate-breccias and conglomerates to dominantly finer-grained, sandy-silty alluvial and floodplain units with paleosol horizons [20]. The lowermost part of the first sequence also contains volcaniclastic and pyroclastic intercalations, allowing its identification as the local development of the mixed siliciclastic-volcaniclastic Lower Member of the formation, whereas the remaining depositional units separated in the larger study area belong to the lower part of the Middle Member. Site K2, discovered in 2019 along the upper reaches of Neagului Valley, west of Vălioara [11], is located in the upper, finer-grained section of the first depositional sequence (Fig 3A), identified informally as the 'grey Vălioara beds', or Unit 1.5., by [20]. This unit is considered to have been formed within the distal, fluvially-dominated sectors of alluvial fans fed through the erosion of nearby uplifted basement rocks followed by short-distance transport of the resulting clastic material.

## 3. Materials and methods

Between 2021 and 2023, ten day-long follow-up excavations were carried out in the area of site K2. In addition to the vertebrate excavations, samples were taken from different beds of the site for palynological analyses, and invertebrate remains (gastropods and bivalves) from the bone-bearing succession were also collected for a more detailed study of the local paleofauna and a better understanding of the paleoenvironment. All invertebrate and vertebrate remains mentioned in the text are curated as part of the collections of the Laboratory of Paleontology of the Faculty of Geology and Geophysics, University of Bucharest, and are accessioned here under LPB (FGGUB) numbers; meanwhile, the palynological samples are inventoried at the 'Alexandru Ioan Cuza' University, Iaşi. The described study complied with all relevant regulations, and the fieldwork in the Vălioara area had been conducted under permits issued by the Haţeg Country UNESCO Global Geopark and by the National Agency for Natural Protected Areas (REF: 437/STDH/08.04.2022 and 468/STDH/09.04.2024).

Two samples collected from fine-grained deposits of site K2 were analysed for organic matter and palynological content assessment. Sample 6B has a somewhat lower stratigraphic position compared to sample P2, both of these coming from a succession of sandy clays and clays that have a thickness of about 2–2.5 m (Fig 3B). These samples were treated using traditional palynological techniques [36] involving the HCl–HF acid maceration method. The organic residue was mounted on microscopic slides using glycerin jelly and subsequently examined under transmitted light microscopy for palynomorph identification and palynofacies recognition. An organic geochemistry investigation was also performed on sample 6B (Table 1), including analysis of the bitumen extract using gas chromatography–mass spectrometry (GC–MS), as well as evaluation of the total organic carbon (TOC) content. These geochemical analyses were carried out at the Institute of Earth Science of the University of Silesia in Sosnowiec (Poland). The paleoenvironmental interpretation of the samples is based on geochemical data (e.g., cross-plots of phytane/n-C18 versus pristane/n-C17 components, and Pr/Ph against Pr/n-C17), as well as on palynofacies analyses (i.e., relative abundance of different terrestrial organic matter components; shape and size of opaque phytoclasts; presence of woody tissues and cuticles) following the studies of [37–40].

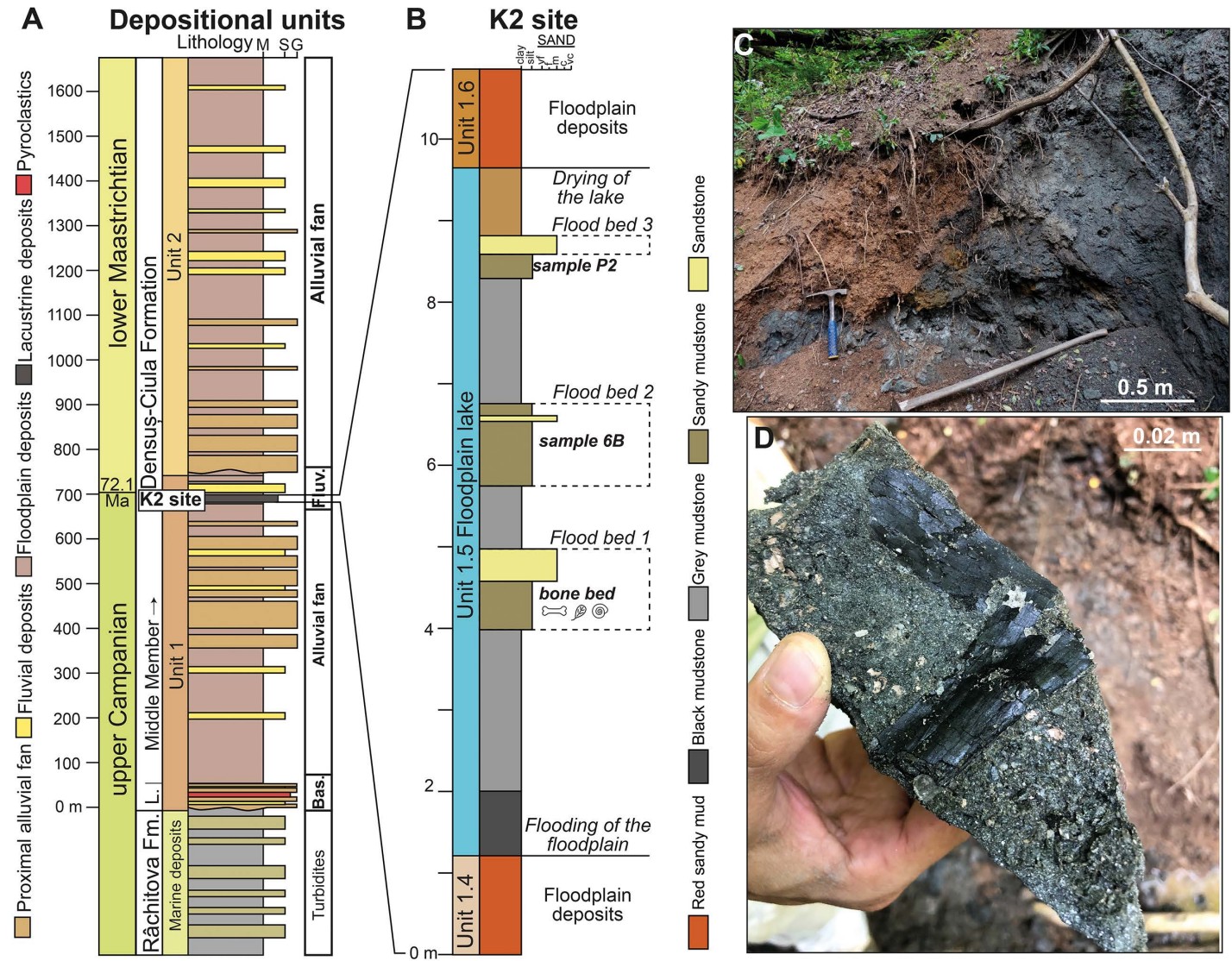

**Fig 3. Geological background and sedimentology of site K2.** (A) Sedimentary architecture, depositional environments and depositional units of the lower Densuș-Ciula Formation in the Vălioara area, for details see [20] (Bas.: basal deposits, Fluv.: fluvial deposits). **(B)** Sedimentary log of the K2 site, with important surfaces and sample locations. (C) Lower portion of the K2 outcrop showing structureless black and grey mudstone. **(D)** Texture of the heterolithic bonebed, plant fragments and molluscs.

**Table 1. Results of the geochemical analyses (TOC, TS and n-alkanes) on sample 6B, site K2, Vălioara, Densuș-Ciula Formation.**

| Sample | TC (%) | TIC (%) | TrefereOC (%) | TS (%) | *n*-Alkanes and acyclic isoprenoids | | | |
|---|---|---|---|---|---|---|---|---|
| | | | | | Pr/*n*-C$_{17}$ | Ph/*n*-C$_{18}$ | Pr/Ph | CPI (C$_{24-34}$) |
| 6B, site K2, Pârâul Neagului, Vălioara | 2.304 | 0.577 | 1.727 | 1.541 | 2.45 | 0.98 | 2.54 | 3.72 |

Abbreviations: TC – Total Carbon; TIC – Total Inorganic Carbon; TOC – Total Organic Carbon; TS – Total Sulfur; Pr – pristane; Ph – phytane; CPI – Carbon Preference Index.

A relatively rich mollusc fauna was collected from site K2, and the recovered specimens were prepared with common hand-held preparation tools. The largest part of the invertebrate material came from the bonebed horizon of site K2 (Fig 3B), although a few well-preserved gastropod shells were also collected from the sediments overlaying the bone-bearing horizon. The gastropod and bivalve shells are in the millimetre size range, so that the successful extraction of individual specimens from the rocky matrix was usually done using a binocular microscope. Photos of the most important specimens were taken with a Nikon SMZ25 digital microscope and the compatible Nikon Nis-Elements software.

The recovered macrovertebrate remains were prepared mechanically in the technical laboratory of the Department of Palaeontology of the Eötvös Loránd University (ELTE; Budapest, Hungary), using a Fiac Compact 106 oilless air compressor with W 224 pneumatic preparation head, Dremel engraver 290−02, and preparation needles. The bones were fixed with cyanoacrylic glue. Photos were taken with a Canon EOS 600D DS126311 camera. Beside the excavation of the abundant macrovertebrate material, about half a ton of sediments was also collected for screen-washing, a procedure that yielded a microvertebrate assemblage. During the screen-washing procedure three different mesh sizes (0.5 mm, 1.25 mm, and 2 mm) were used, of which the 1.25 mm one yielded the most productive fraction. Photos of the recovered microvertebrate remains were taken with a NIKON H550L camera and a Phenom XL Scanning Electron Microscope.

In order to collect as much of the taphonomic information as possible about the K2 fossil vertebrate assemblage, we followed the methodology described by [8,41–43]. In addition to the preliminary observations recorded in the field directly during excavations, the taphonomic investigation of the assemblage also entailed further examination of each vertebrate remain in the laboratory in order to determine as precisely as possible its skeletal and taxonomic identity. In addition, any taphonomic modification feature (for instance weathering stage, degree of abrasion, breakage pattern, etc.) was carefully documented for the macrovertebrate specimens. Data such as taxonomic and skeletal identity, shape, size, state of preservation, and taphonomic modifications observed on the specimens were summarized in a comprehensive taphonomic observation file (see S1 File and S2 Table). Selected specimens from site K2, especially those discussed in this report, are summarised in S3 File with their provisional and/or final inventory numbers.

## 4. Results

### 4.1. Sedimentology of site K2

The uppermost Cretaceous basin-fill succession hosting the vertebrate sites in the Vălioara area is dominated by vertically and laterally interfingering gravelly and muddy deposits that were deposited in an alluvial fan environment, and represent proximal and, respectively, distal reaches of fan bodies, with the addition of sandy-gravelly fluvial channel deposits [20]. The deposits were built up by a vast amount of coarse material that originated from the then already subaerially exposed and weathering, dominantly metamorphic basement blocks to the north and northwest.

The sedimentary succession hosting site K2 crops out in the upper reaches of Neagului Valley, west of Vălioara (Fig 2B). It represents a unique mud-dominated sequence devoid of gravelly material within the otherwise primarily coarse basin-filling succession, and thus stands out in this regard even compared to the other more finer-grained, dominantly fluvial depositional sequences mapped in this area by [20]. Its ca. 8.5 m thick section is dominated by clean grey to black mudstones, interrupted by three sandy-silty interbeds (Fig 3B).

The first such bed is 1 m thick and overlies a ca. 3 m thick black and grey mudstone-dominated interval that itself is deposited along a sharp but non-erosive contact on top of a red sandy mudstone containing calcareous nodules (Fig 3B). The coarser bed itself begins with a basal, 0.1 to 0.4 m thick sandy-siltstone that is characterised by a sharp basal boundary and contains abundant vertebrate fossils, molluscs and plant fragments. The proportions between sand and silt as well as the thickness of the bonebed changes laterally. This heterolithic facies is overlain by a 0.2 m thick medium-grained sand layer occasionally also containing vertebrate fossils. The first and the second coarser beds are separated from each other by 0.75 m of grey mudstone. The second bed shows similar internal facies to the previous one, as it starts with sandy silt containing vertebrate and mollusc fossils along with plant fragments, followed by a clean, 0.05 m thick medium-grained

sandstone. However, unlike in the previous, first fossiliferous bed, the muddy sand facies is repeated once more above the clean sand facies, albeit with a limited thickness, and it is followed by sandy mudstones. Although this fossiliferous interval is thicker than the lowermost one (1 m vs. 0.6 m), it yielded a significantly lower number of vertebrate fossils.

This second fossiliferous sandy interval is again followed by clean grey mudstones (2 m) atop a sharp, non-erosional boundary, overlain by a cross-laminated medium grained sandy facies representing the third coarser bed (0.25 m), although this one does not contain any fossils. The third sandy bed is finally overlain by ca. 1 m thick brown mudstone containing no coarse detritic material.

This entire richly fossiliferous section is covered, over a sharp boundary, by several meters thick red, poorly sorted mudstones reminiscent of the deposits immediately underlying it, and these are eventually followed by trough cross-laminated pebbly sandstones interpreted as floodplain and, respectively, fluvial channel fills by [20]. Remarkably, deposits similar in terms of sedimentary characteristics to those representing the fossiliferous interval at site K2 do not return vertically within the local succession, nor were we able to track or identify them in other parts of the study area.

### 4.1.1. Interpreted depositional environment based on the sedimentological investigations.

Given the absence of gravelly material as well as that of pedogenic features, both of which otherwise characterize the different locally separated units of the Densuș-Ciula succession [20], in combination with the presence of freshwater molluscs (see below), the studied sedimentary sequence was most likely deposited in a quiet, subaqueous environment in which sedimentation was dominated by suspension settling. The sharp but non-erosional basal contact with the underlying red mudstones that display pedogenic features indicates abrupt low-energy flooding of the previously subaerially exposed depositional area, which lead to a formation of a floodplain lake [44,45] (Fig 3B).

Meanwhile, the sandy intervals of the fossiliferous section indicate recurring events that were capable of transporting sandy material and, occasionally, vertebrate fossils into the otherwise quiet, mud-dominated environment. The origin of the silty-sandy and sandy material could be thus linked to a series of fluvial flooding events that moved and deposited coarse material into the lake. The coarsening-upward and coarsening-fining upward nature of Flood Bed 1 and 2, respectively (Fig 3B), can indicate episodes of increase, peak, and decrease of flow velocity that are characteristics to river floods [46–48].

The deposition of the uppermost sandstone bed marks the cessation of lacustrine conditions in the studied section, as it is overlain by red sandy-gravelly mudstones with abundant pedogenic features. Cross-stratification present in this sandstone bed might indicate either a fluvial origin or a proximal delta front environment.

### 4.2. Palynological assemblage

Two samples (P2, 6B) collected from fine-grained deposits of site K2 (Fig 3B) were analyzed for organic matter and palynological content assessment. Sample 6B (mentioned in [20]) has a lower stratigraphic position compared to sample P2 (first reported preliminarily in [11]), although both are coming from this fossiliferous succession of sandy clays and clays that have a thickness of about 2–2.5 m. The recovered microfloral assemblages consist of 62 terrestrial taxa (S3 File), including 33 pteridophyte spores, 8 gymnosperm and 17 angiosperm pollen, as well as minor occurrences of bryophytes (2 taxa) and freshwater algae (2 taxa). The identified palynomorphs (Fig 4) show a good preservation state in both samples, and consist mainly of pteridophyte spores such as Polypodiaceae (*Polypodiaceoisporites* div. sp.; Figs 4B-E), Cyatheaceae (*Deltoidospora australis* – Fig 4A, *Deltoidospora minor, Deltoidospora toralis*), Gleicheniaceae (*Gleicheniid- ites senonicus*), and Schizaeales (*Cicatricosisporites* sp.). The fern spore *Lusatisporites dettmanniae* (Fig 4G), cited here for the first time from the Upper Cretaceous of Hațeg Basin, occurs only in sample 6B. Gymnosperm pollen grains are mainly represented by high-altitude species such as *Pinuspollenites* sp. (Fig 4K) and *Araucariacites australis* (Fig 4M), as well as by *Classopollis* sp. (Fig 4L), a palynomorph taxon often regarded as derived from saltmarsh vegetation [49,50].

The low-diversity angiosperm pollen assemblage consists of specimens assigned to the Normapolles group (e.g., *Trudopollis nonperfectus* – Fig 4O, *Trudopollis minimus* – Fig 4S, *Oculopollis praedicatus* – Fig 4N, *Pseudopapillopollis praesubhercynicus* – Fig 4R). A remarkable occurrence is that of *Proteacidites* (Fig 4P), a taxon typical for the Southern

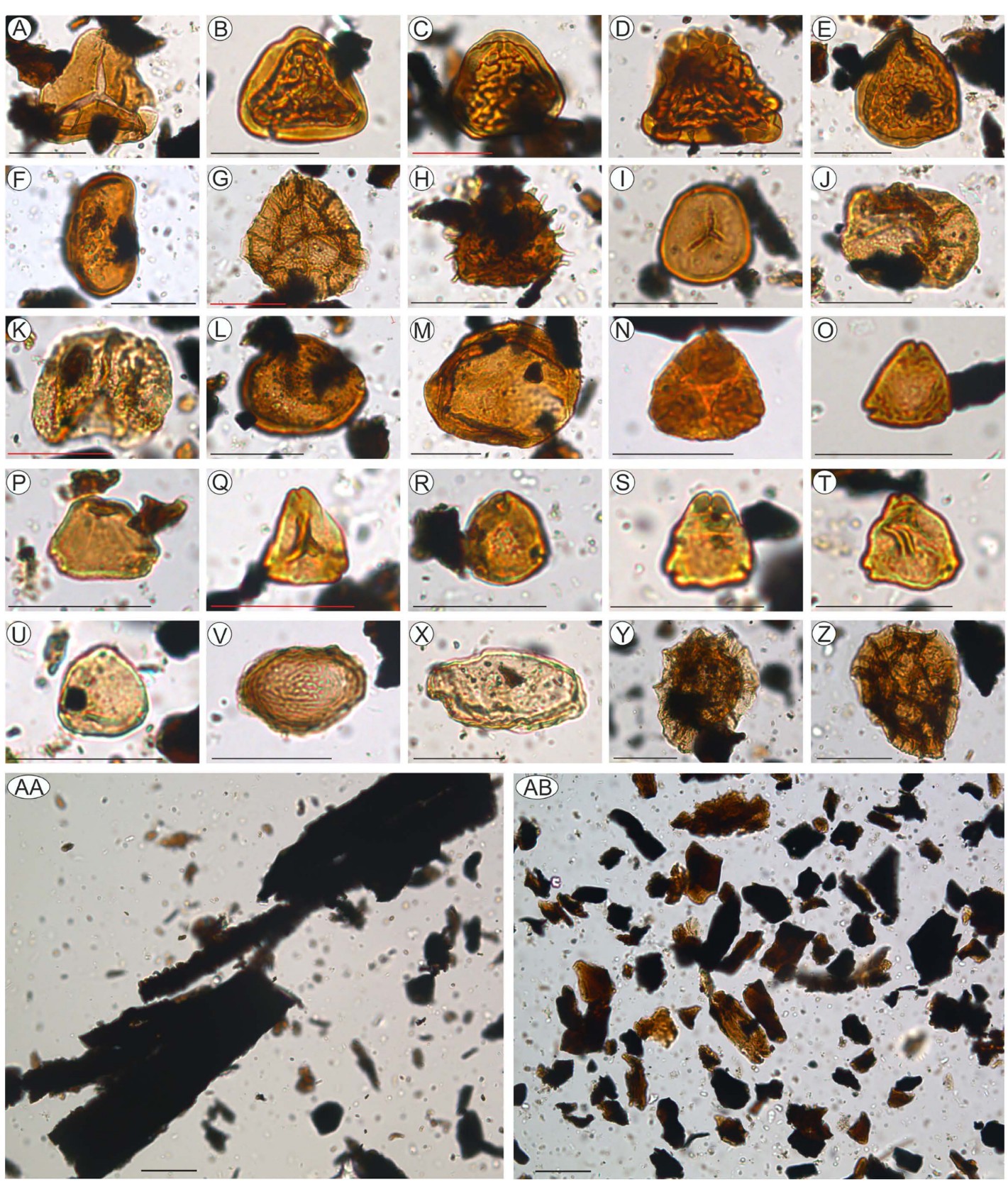

**Fig 4. Selected terrestrial palynomorphs, freshwater algae, and typical constituents of organic matter recognized at site K2, Vălioara (Densuș–Ciula Formation). (A)** *Deltoidospora australis* (sample 6B). **(B)** *Polypodiaceoisporites* sp. (sample P2). **(C-D)** *Polypodiaceoisporites* sp. (sample 6B). **(E)** *Polypodiaceoisporites hojrupensis* (sample 6B). **(F)** *Laevigatosporites ovatus* (sample 6B). **(G)** *Lusatisporites dettmanniae* (sample 6B). **(H)** *Echinatisporis* sp. (sample 6B). **(I)** *Stereisporites antiquasporites* (sample 6B). **(J)** *Inaperturopollenites dubius* (sample 6B). **(K)** *Pinuspollenites* sp. (sample 6B). **(L)** *Classopollis* sp. (sample 6B). **(M)** *Araucariacites australis* (sample 6B). **(N)** *Oculopollis praedicatus* (sample P2). **(O)** *Trudopollis nonperfectus* (sample P2). **(P)** *Proteacidites* sp. (sample 6B). **(Q)** *Plicapollis serta* (sample 6B). **(R)** *Pseudopapillopollis praesubhercynicus* (sample 6B). **(S)** *Trudopollis minimus* (sample 6B). **(T)** *Myricipites bituitus* (sample 6B). **(U)** *Subtriporopollenites anulatus* (sample P2). **(V)** *Chomotriletes fragilis* (sample P2). **(X)** *Ovoidites* sp. (sample 6B). **(Y-Z)** *Pterodinium cingulatum* cf. *granulatum* (sample 6B; reworked). **(AA)** large lath-shaped opaque phytoclasts (sample P2). **(AB)** a mixture of equidimensional opaque and translucent phytoclasts (sample 6B). (scale bar: 30 μm).

Hemisphere [51] and which is yet to be reported from other Upper Cretaceous deposits in Europe; currently, it could be considered as a local, Hațeg Basin endemics within the Northern Hemisphere. Very rare occurrences of freshwater algae such as *Chomotriletes fragilis* (Fig 4V) and *Ovoidites* sp. (Fig 4X) were also identified.

In addition to the autochthonous terrestrial palynomorphs, sample 6B also contains specimens of *Pterodinium cingulatum* cf. *granulatum* (Figs 4Y and 4Z), a marine phytoplankton taxon that can be considered reworked from older marine deposits, as it was reported from the lower Upper Cretaceous (Cenomanian–Santonian) Ștei Formation [52] currently cropping out farther to the west. The same sample also yielded a number of reworked spore specimens typical for the Jurassic and Early Cretaceous, easily recognizable under the microscope as such due to their darker colour.

### 4.2.1. Age interpretation and paleoenvironmental reconstruction based on the palynological investigations.

These newly identified palynological assemblages contain several biostratigraphically relevant taxa that allow an age refinement of the K2 site deposits, previously considered by [11] to belong to the lower Maastrichtian. Indeed, according to the biostratigraphic interpretations of [53–55] two taxa identified in sample 6B, *Polypodiaceoisporites hojrupensis* and *Pseudopapillopollis praesubhercynicus*, restrict the age of the K2 local assemblage to the late Campanian to earliest Maastrichtian time interval. Considered together with other litho- and biostratigraphic constraints reported recently from the Vălioara area by [20], the age of the locality can be confidently constrained to the earliest part of the Maastrichtian, around or very close to the Campanian/Maastrichtian boundary. Accordingly, the fossil assemblage recovered from site K2 ranks among the oldest ones known from Hațeg Basin, and from the entire Hațeg Island as well (see fig 20 in [20]), and it is definitively the best-sampled, most diverse early biotic assemblage currently on record in Hațeg Basin, documenting the earliest stages of its insular faunal evolution.

Overall, the K2 samples yielded a palynological assemblage dominated by hygrophytic fern spores such as *Polypodiaceoisporites*, *Deltoidospora* and *Laevigatosporites*, suggestive of plant communities growing along riverbanks or around lakes [56], under rather warm climatic conditions [56]. Their dominance, together with the presence of *Proteacidites* and *Inaperturopollenites* (i.e., *Taxodium*-related) pollen as well as that of rare phytoplankton specimens referable to freshwater algae, indicates the presence of perennially to temporarily wet, low-elevation fluvio-deltaic systems such as wetlands, ponds, swamps or temporary lakes [37,57,58]. Roughly similar palynological assemblages have previously been described from Maastrichtian continental deposits of Hațeg Basin [59] and southwestern Transylvanian Basin [60].

The palynofacies assemblage identified in sample 6B contains a fairly large amount of continental organic matter (TOC = 1.727%; Table 1) represented mainly by translucent phytoclasts (~60–65% of the total kerogen; Fig 4AB), mixed with equidimensional opaque phytoclasts that suggest a longer transport, and some large cuticles (Fig 5B) typical of deltaic deposits [37]. Furthermore, the $Pr/n-C_{17}$ versus $Ph/n-C_{18}$ ratios indicate a type III kerogene (Fig 5D) deposited in a peat coal environment (Fig 5C) under suboxic conditions (Fig 5E).

## 4.3. Mollusc fauna

Molluscan shells representing several taxonomic groups were found at site K2 (Figs 6 and 7), such as freshwater bivalves, freshwater "pulmonates" (Lymnaeidae and Physidae), and terrestrial snails (Caenogastropoda: Cyclophoroidea

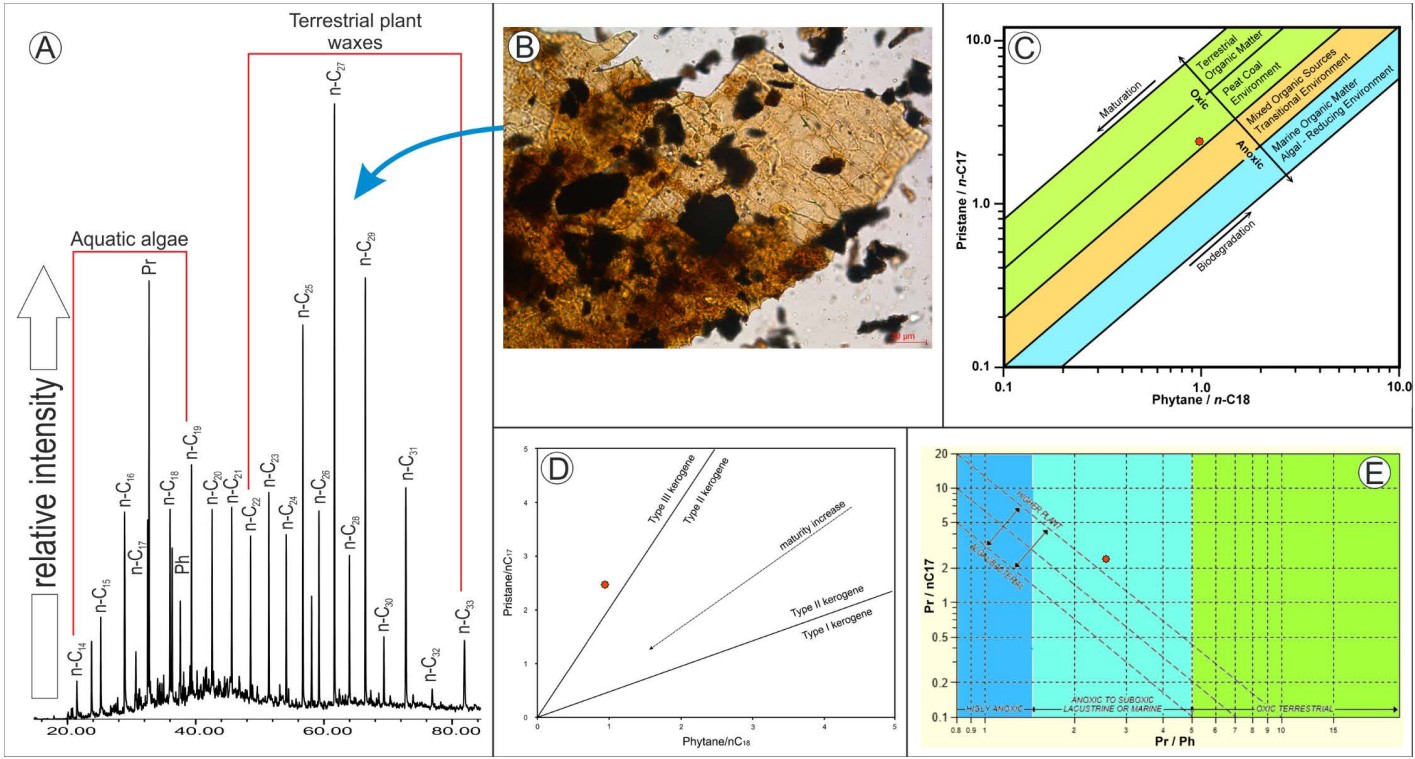

**Fig 5. Gas chromatogram–mass spectrometry spectra from sample 6B, site K2, Vălioara (Densuş–Ciula Formation) (A), and diverse cross-plots: C, D. phytane/n-C18 versus pristane/n-C17 indicating the type of organic matter and depositional environment.** (B) fragment of large-sized leaf cuticle considered to be composed mainly of long-chain n-alkanes (n-C23 – n-C31).

and Eupulmonata: Stylommatophora). The preservation of the shells is, however, mediocre. Most of them could be identified up to family level, while a minority of these fossils showing character-poor shells could not be assigned to any family with certainty. Since probably the largest part of the K2 mollusc fauna is new to science, its detailed description will be done together with similar fossils from other Haţeg Basin sites.

**Freshwater gastropods:**

Some middle-sized (ca. 5–10 mm) conical shells with irregular shell sculpture can be classified in the freshwater family Lymnaeidae (Fig 6A). However, their classification is uncertain given that some land snails (e.g., extant Enidae, see [61]) have similar shell shapes. A single, strongly damaged shell of a freshwater limpet (Fig 6B) was found. Freshwater limpets are classified in the subfamily Ancylinae of the family Planorbidae, and in the family Acroloxidae [62]. The peak of our specimen is bent to the left and to the posterior side, similarly to some extant *Acroloxus* species belonging to the Acroloxidae. However, the position of the peak is reminiscent of the similarly Late Cretaceous freshwater limpet taxon *Ancylina cretacea* [63] from Ajka, Hungary, which is currently classified in the Ancylinae [64].

Small, pointed, sinistral shells (Figs 6C–E) were frequently found at site K2, although the majority of them are poorly preserved. They undoubtedly belong to the family Physidae, a referral that is also supported, besides the shell shape and the coiling direction, by the characteristic beaded sculpture [65], which is clearly visible on most K2 specimens. Based on their shell width, it appears that two or three species of this group have lived in the K2 area, although diagnosis and description of these will have to be done together with further material from nearby sites. Large (ca. 1 cm) sinistral-shelled Physidae (Fig 6F–J) were frequently found in the K2 material; based on the recorded abundance, these were probably

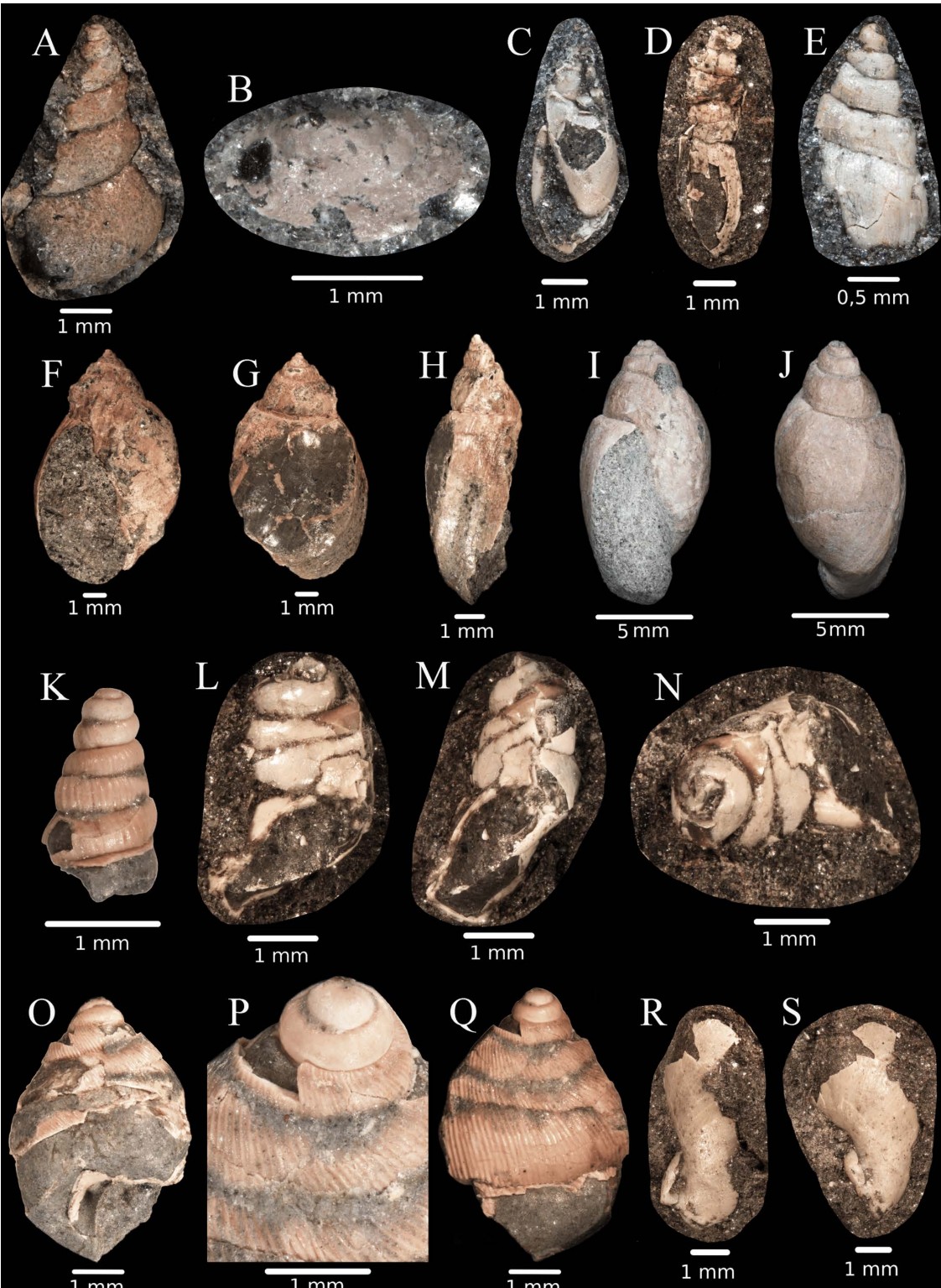

**Fig 6. Shells of the terrestrial and freshwater gastropods from site K2, Vălioara (Densuş–Ciula Formation). (A)** Lymnaeidae sp. (abapertural view). **(B)** *Acroloxus* sp. (apical view). **(C)** Physidae sp. 1 (apertural view). **(D)** Physidae sp. 2 (lateral view). **(E)** Physidae sp. 3 (abapertural view). **(F)** Physidae sp. 4 (apertural view). **(G)** Physidae sp. 4 (abapertural view). **(H)** Physidae sp. 4 (lateral view). **(I)** Physidae sp. 4, specimen2 (apertural view).

(J) Physidae sp. 4, specimen2 (abapertural view). (K) *Ajkaia* sp. (abapertural view). (L) Pupinidae sp. 1 (lateral view). (M) Pupinidae sp. 1 (abapertural view). (N) Pupinidae sp. 1 (apical view). (O) Pupinidae sp. 2 (lateral view). (P) Pupinidae sp. 2 (enlarged view of the protoconch and the initial whorls of the teleoconch). (Q) Pupinidae sp. 2 (lateral view). (R) Laminiferinae sp. (abapertural view). (S) Laminiferinae sp. (lateral view).

the most abundant fossil snails locally. Since most shells were not perfectly preserved, reliably assessing the number of species present is challenging. Due to the overall low conchological variability, we provisionally handle the large sinistral shells as representing a single physid species.

**Terrestrial Caenogastropoda:**

 A single shell of a slender, regularly ribbed snail was found, which is putatively classified in the genus *Ajkaia* [63], because it shows strong similarities with the similarly strongly ribbed *Ajkaia gracilis* [63] from Ajka, Hungary. [64] noted that the type species of *Ajkaia*, *A. gregaria* [63]cannot be distinguished from shells he attributed to the genus *Megalomastoma*. However, there is a large size gap between the two groups; shell height of *Megalomastoma* species described by [63] measure 13–18 mm, while that of *A. gregaria* is only 3 mm. *Megalomastoma* belongs to the Central American family Megalomastomatidae (see [66]), and the Cretaceous European species are probably more closely related to the Southeast Asian family Pupinidae. [64] also mentioned that *A. gracilis* is similar to extant Diplommatinidae. We agree that a small ribbed shell with rounded aperture is probably either a diplommatinid, or a pupinid [cf. [67,68]]. The systematic position of our K2 shell is even more questionable as the aperture, which would contain the taxonomically most important characters, is not preserved.

At least two species of the Pupinidae have been found. This family is currently most speciose in Southeast Asia (see, [66,69]), and they mostly inhabit humid forests with limestone outcrops [70–72]. A small (ca. 3 mm), ovoid, whitish, glossy morphotype (Fig 6L–N), and a middle-sized (adult size ca. 6 mm), ribbed, conical-ovoid one (Fig 6O–Q) are classified in this family, probably representing two different species. Most extant and fossil pupinid genera are defined based on the morphology of the aperture (see [69, 73]), which is unfortunately not preserved in the K2 specimens.

**Stylommatophora:** A single last whorl fragment of a sinistral shell with a strongly reflected peristome clearly indicates that it belonged to the family Clausiliidae, and the aperture shape suggests that it may be a representative of subfamily Laminiferinae (Fig 6R–S). This specimen would be then one of the oldest clausiliid fossils, as currently only two *Proalbinaria* species are known from the Cretaceous [74], and the so far oldest member of the Laminiferinae [75] is known from the Upper Paleocene [75].

The genus *Lychnus* is characterized by an unusually enlarged last whorl, and the currently known 16 species have been described mostly from France, Spain and Austria [76,77]. A single *Lychnus* specimen was found at site K2 (Fig 7A–B). Several specimens are characterized by a flat shell, a large number of whorls, and a rounded body whorl (Fig 7C–G). Unfortunately, most of these are not in a good preservation state, and therefore, not all important shell characters are visible. Nevertheless, the ratio of shell height to shell width is different enough to state that we have probably three species of this group represented at site K2. Species similar in shape to the K2 specimens have been described by [76] as *Proterocorilla europaea* and *Gosavidiscus acutimarginatus* (now belonging to the family Anostomopsidae, see [78]), but they are characterised by the presence of several longitudinal apertural barriers, while no such barriers were found in the flat helicoid shells we collected. Finally, several individuals with elongated ovoid shells (Fig 7H–K) were found at K2, but due to their rather poor state of preservation and the presence of few useful characters, their taxonomic position remains questionable. They may belong either to the terrestrial group Orthurethra (e.g., Cochlicopidae, see, [61]), to the Pupinidae, or even to the freshwater family Lymnaeidae.

**Bivalvia:** The majority of the bivalve shells found at site K2 belong to the family Sphaeriidae, which are characterized by small, rounded valves with a rather centrally located umbo. As nearly all valves are attached to the bedrock with their concave side (where the teeth that are crucial for species recognition are located), we had to rely on their sculpture

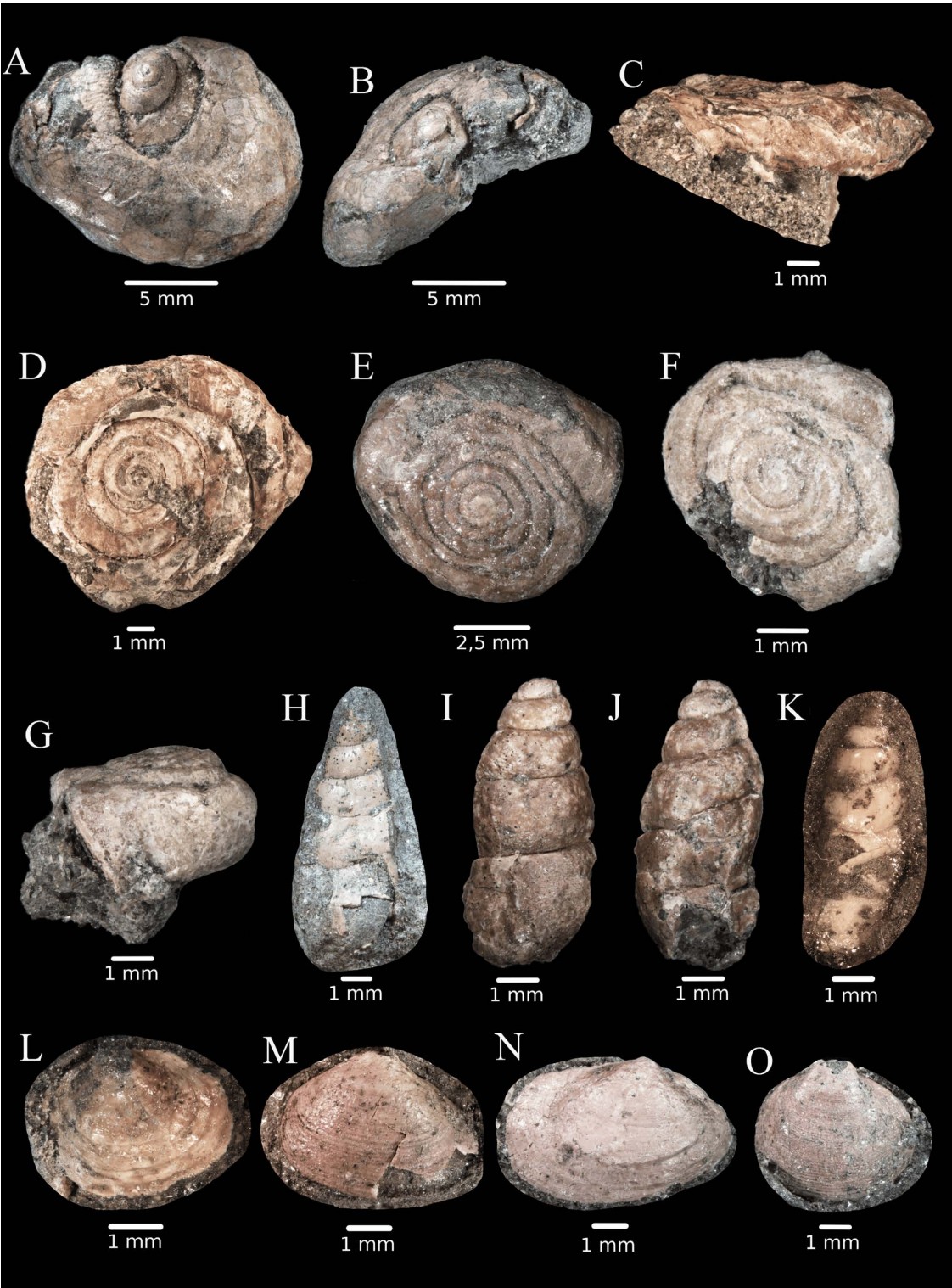

**Fig 7. Shells of terrestrial gastropods and freshwater bivalves from site K2, Vălioara (Densuș–Ciula Formation). (A)** *Lychnus* sp. (apical-dorsal view). **(B)** *Lychnus* sp. (apical view). **(C)** Helicoidea sp. 1 (lateral view). **(D)** Helicoidea sp. 1 (apical view). **(E)** Helicoidea sp. 2 (apical view). **(F)** Helicoidea sp. 3 (apical view). **(G)** Helicoidea sp. 3 (lateral view). **(H)** slender gastropod sp. 1 (dorsal view). **(I)** slender gastropod sp. 2 (abapical view). **(J)** slender gastropod sp. 2 (apertural view). **(K)** slender gastropod sp. 3 (lateral view). **(L)** Sphaeriidae sp. 1. **(M)** Sphaeriidae sp. 2. **(N)** Cyrenidae sp. **(O)** Sphaeriidae sp. 3.

(density of growth lines) and to a lesser extent, their overall shell shape to identify potential lower-level taxa. Based on these characters, we recognize possibly as many as 3 different species (Fig 7 L, M, O). Some other valves are elongated with posteriorly located valves, which may indicate that they belong to another family (possibly Cyrenidae). The shape of the shell on Fig 7N is similar to *Cyrena concinna* illustrated by [79].

### 4.3.1. Paleoenvironmental reconstruction and paleoecological interpretation based on the mollusc fauna.

The deposits exposed at site K2 contained numerous molluscan fossils. Although the preservation state of the shells is mostly mediocre, in the majority of the cases it is possible to assign them to families. The identified freshwater fauna contains approximately 4 bivalve taxa, 4 species of Physidae, 1 limpet (most probably Acroloxidae), and a lymnaeid. Shells with sinistral coiling direction belong to the family Physidae, which is currently the most widespread freshwater family with 80 known extant and 135 extinct species ([80,81] and MolluscaBase Eds., 2025). Physidae, together with small shelled bivalves, are undoubtedly characteristic for lentic environments (lakes, ponds, swamps, slowly flowing rivers). Due to their sinistral coiling direction, it was possible to identify even tiny (smaller than 1 mm) embryos to family level, while the same accuracy was not achievable for most shell fragments and juvenile shells representing other groups. In order to avoid a resulting bias, we do not undertake any quantitative analysis of the mollusc fauna, but nevertheless emphasize that bivalves and physids were among the most numerous fossils at site K2 site. The snails are considered as feeding on algae and detritus, and many of their extant relatives tolerate eutrophic conditions [82].

The number of identified terrestrial taxa is similar to that of the freshwater ones, but their fossils are less numerous than those of the physids and the bivalves. We could identify three indeterminate species of Pupinidae (Caenogastropoda), ca. three species belonging to Helicoidea (Stylommatophora), and one *Lychnus* species (Stylommatophora). Similarly to our site, terrestrial operculate snails (see [67]) dominate other Upper Cretaceous European sites as well [63,76,79]. Recent faunas reminiscent to that identified at site K2 are distributed nowadays in the humid tropical forests of Southeast Asia, where comparable snails are mostly found in the vicinity of limestone rocks, in the leaf litter, and under decaying logs [70,72].

### 4.4. Vertebrate assemblage

During our fieldwork between 2019 and 2023 a large amount of vertebrate fossils, most of them well-preserved (Fig. 2D-G and Figs. 8-11), were collected from site K2; the more than 800 specimens represent at least 17 different species, albeit not all of these can be identified yet at a low taxonomic level (see Tables 2-3 and S3 File).

Overall, the faunal composition of the K2 assemblage is similar to those reported from other vertebrate localities of the Densuș-Ciula Formation [3], but nevertheless some of the newly identified K2 occurrences are of paleontological and/ or paleoecological significance (see [5,11,12]). Furthermore, asmentioned above, the assemblage originating from site K2 represents the oldest best-sampled paleofauna from the entire Hațeg Basin. Here, we will briefly review the entire K2 vertebrate assemblage, and comment on the representation and significance of the different taxa identified. We also note, nonetheless, that some of the taxa represented at the site already were [12] or currently are [5,12,83] under more in-depth investigation, given their better and more complete skeletal representation and/or specific importance for tracking faunal composition and evolution on Hațeg Island.

**Fishes:** Just as reported in other fossiliferous sites of Hațeg Basin, fish remains are extremely rare in the K2 material as well. Only four rhomboid ganoid scales (Fig 8A) most probably attributable to lepisosteiforms are considered to represent this group in the recovered material.

**Amphibians:** Similarly to fish remains, only a small number of amphibian ones are known from site K2. In addition to characteristic limb bones belonging to indeterminate Anura, a well-preserved and 8 mm long albanerpetontid dentary was also recovered (LPB (FGGUB) v.901; Fig 8B-C), representing the best-preserved such element currently on record from Hațeg Basin. Based on the preliminary investigation of this specimen, [83] suggested that it is similar to the dentaries of significantly older representatives of the clade such as *Anoualerpeton unicum* described from the Berriasian of Morocco [84], or *Shirerpeton isaji* from the Barremian of Japan [85]. A more detailed study of this specimen is ongoing.

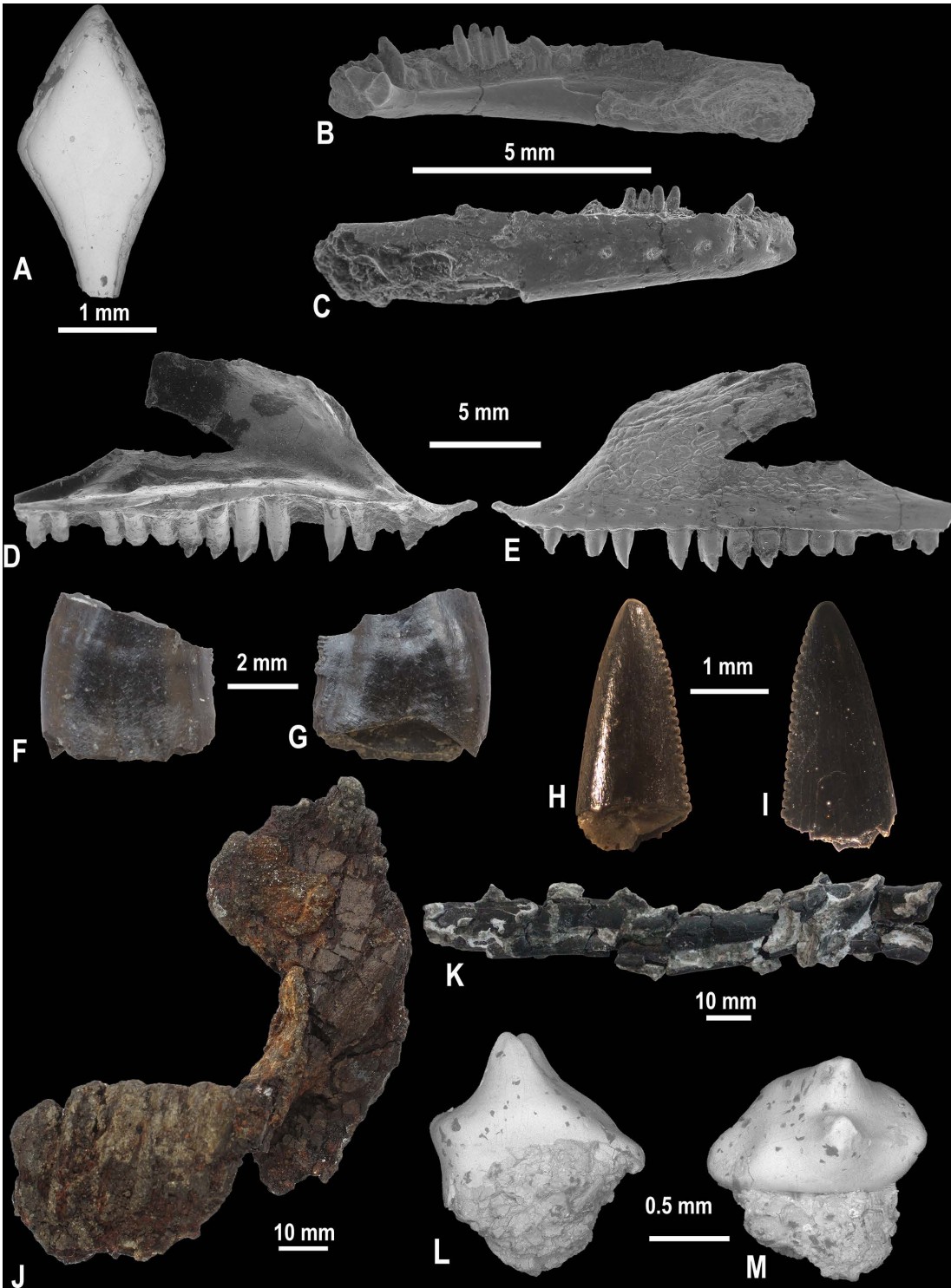

**Fig 8. Isolated vertebrate remains, site K2, Vălioara (Densuș–Ciula Formation). (A)** Lepisosteiformes indet., scale LPB [FGGUB] v.900 in external view. **(B)** Albanerpetontidae indet., right dentary LPB [FGGUB] v.901 in lingual and **(C)** labial views. **(D)** Barbatteidae indet., maxilla LPB [FGGUB] v.902 in lingual and **(E)** labial views. **(F)** Velociraptorine dromaeosaurid tooth, LPB [FGGUB] R.2885 in labial and **(G)** lingual views. **(H)** *Richardoestesia* sp. tooth, (LPB [FGGUB] R.2884) in labial and (I) lingual views. **(J)** Hadrosauroidea indet. right dentary fragment LPB [FGGUB] R.2887 in medial view. **(K)** Possible pterosaurian limb bone fragment LPB [FGGUB] R.2891. **(L)** Kogaionidae indet., left P1 (LPB [FGGUB] M.1710) in lingual and **(M)** oblique occlusal views.

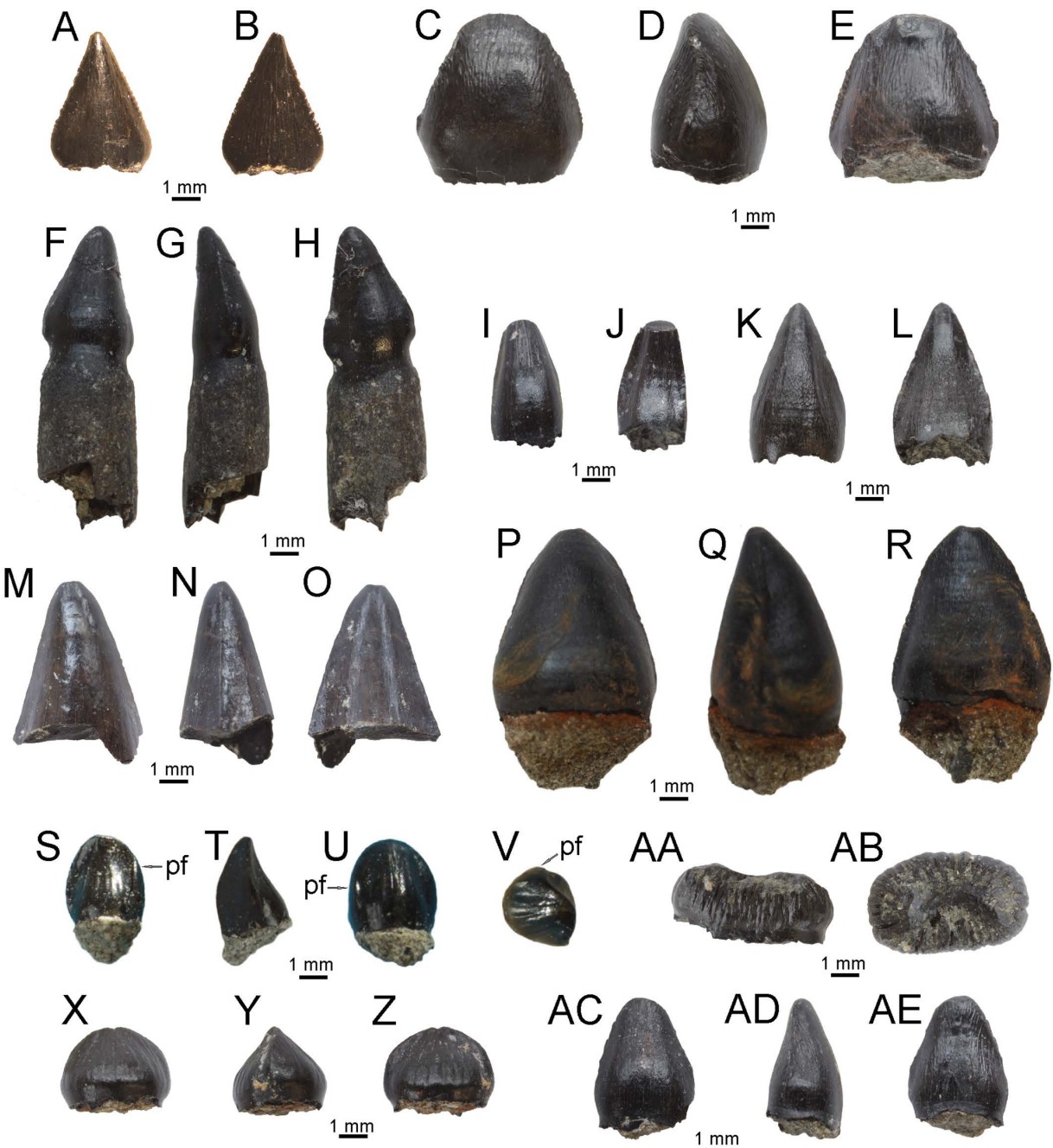

**Fig 9. Isolated crocodyliform teeth, site K2, Vălioara (Densuș–Ciula Formation).** *Doratodon* sp., anterior-middle maxillary/dentary tooth (LPB [FGGUB] v.910) in **(A)**, labial, and **(B)**, lingual views. *Doratodon* sp., posterior maxillary/dentary tooth (LPB [FGGUB] v.907) in **(C)**, labial, **(D)**, lateral, and (E) lingual views. *Theriosuchus*-like taxon, caniniform premaxillary/anterior maxillary/dentary tooth (LPB [FGGUB] v.915) in **(F)**, labial, **(G)**, lateral, and **(H)**, lingual views. *Theriosuchus*-like taxon, hypertrophied caniniform?maxillary tooth (LPB [FGGUB] v.924) in **(I)**, labial, and **(J)**, lingual views. *Theriosuchus*-like taxon, middle-posterior maxillary/dentary tooth (LPB [FGGUB] v.918) in **(K)**, labial, and **(L)**, lingual views. *Allodaposuchus* sp., apical part of anterior maxillary/dentary tooth (LPB [FGGUB] v.911) in **(M)**, labial, **(N)**, lateral, and (O) lingual views. *Allodaposuchus* sp., posterior maxillary/dentary tooth (LPB [FGGUB] v.913) in **(P)**, labial, **(Q)**, lateral, and **(R)** lingual views. *Acynodon* sp., anterior maxillary/dentary tooth (LPB [FGGUB] v.885) in **(S)**, labial, **(T)**, lateral, **(U)**, lingual, and (V) apical views; arrow in S, U and V point to the lateral platform (pf) replacing the carina basally. *Acynodon* sp., middle-posterior maxillary/dentary tooth (LPB [FGGUB] v.906) in **(X)**, labial, **(Y)**, lateral, and **(Z)**, lingual, views. *Acynodon* sp., posterior maxillary/dentary tooth (LPB [FGGUB] v.903) in (AA), labial/lingual, and (AB) apical views. Indeterminate crocodyliform morphotype (?*Acynodon*, or perhaps?juvenile *Allodaposuchus* – see text for details), probably middle-posterior maxillary/dentary tooth (LPB [FGGUB] v.920) in (AC), labial, (AD), lateral, and (AE), lingual views.

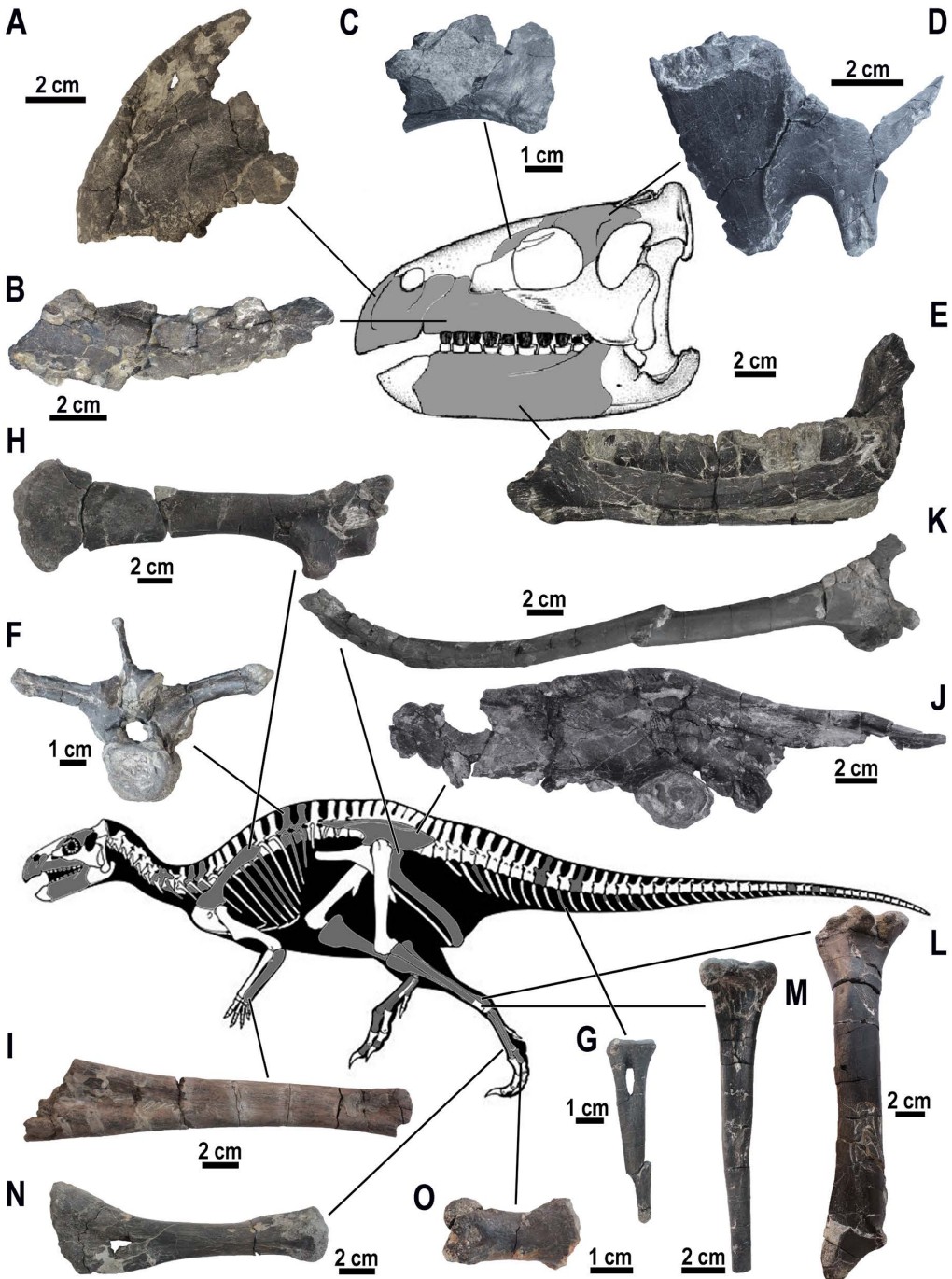

**Fig 10. Generalized cranial and skeletal reconstruction of the 'K2 rhabdodontid', Vălioara (Densuş–Ciula Formation). (A)** Associated premaxillae, LPB (FGGUB) R.2769 in left lateral view. **(B)** Right maxilla, LPB (FGGUB) R.2770, in lateral view. **(C)** Left prefrontal, LPB (FGGUB) R.2772, in dorsal view. **(D)** Articulated right frontal-postorbital, LPB (FGGUB) R.2774, in dorsal view. **(E)** Right dentary, LPB (FGGUB) R.2778, in medial view. **(F)** Middle dorsal vertebra, LPB (FGGUB) R.2795, in posterior view. **(G)** Chevron, LPB (FGGUB) R.2805, in anterior view. **(H)** Right scapula, LPB (FGGUB) R.2806, in lateral view. **(I)** Left ulna, LPB (FGGUB) R.2807, in medial view. **(J)** Right ilium, LPB (FGGUB) R.2809, in lateral view. **(K)** Left ischium, LPB (FGGUB) R.2810, in lateral view. **(L)** Left tibia, LPB (FGGUB) R.2812 in lateral views. **(M)** Right fibula, LPB (FGGUB) R.2814, in medial view. **(N)** Third metatarsal, LPB (FGGUB) R.2816, in lateral view. **(O)** Pedal phalanx, LPB (FGGUB) R.2822, in dorsal view. Shaded skeletal elements indicate bones represented in the K2 assemblage (adapted from [12]).

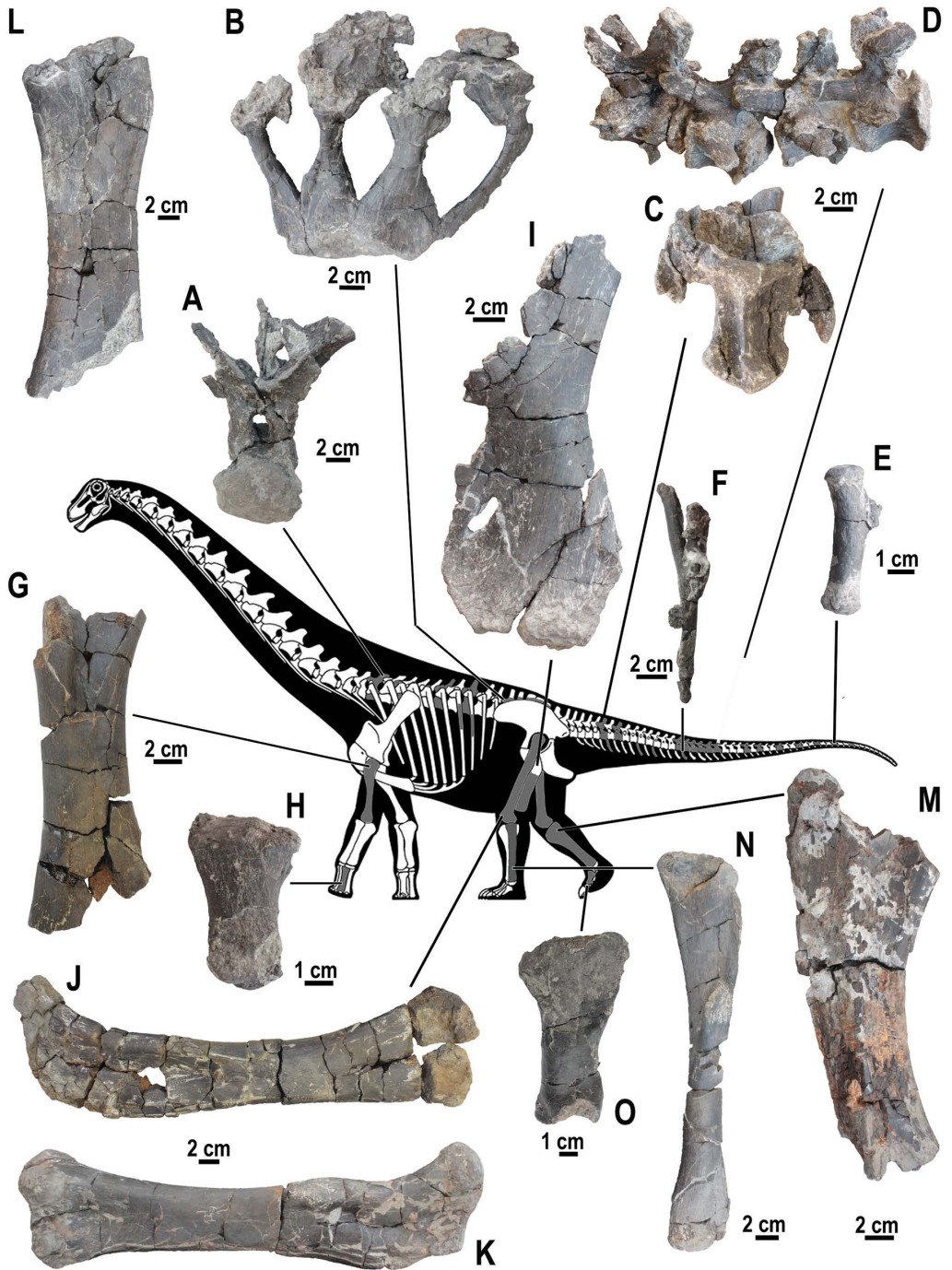

**Fig 11. Generalized skeletal reconstruction of the 'K2 titanosaur', Vălioara (Densuș–Ciula Formation). (A)** Dorsal vertebra LPB (FGGUB) R.2896, in anterior view. **(B)** Partial sacral ribs LPB (FGGUB) R.2897, in dorsal view. **(C)** Anterior caudal vertebra LPB (FGGUB) R.2715, in ventral view. **(D)** Articulated anterior mid caudal vertebrae LPB (FGGUB) R.2715, in left lateral view. **(E)** Posterior caudal vertebra LPB (FGGUB) R.2898, in left lateral view. **(F)** Chevron LPB (FGGUB) R.2899, in anterior view. **(G)** Right humerus fragment LPB (FGGUB) R.2900, in anterior view. **(H)** Metapodial element LPB (FGGUB) R.2091, in lateral view. **(I)** Right pubis fragment LPB (FGGUB) R.2902, in lateral view. **(J)** Fragmentary left femur LPB (FGGUB) R.2903, in posterior view. **(K)** Right femur LPB (FGGUB) R.2904, in posterior view. **(L)** Fragmentary and crushed right femur LPB (FGGUB) R.2905, in posterior view. **(M)** Right??tibia fragment LPB (FGGUB) R.2906, in lateral? view. **(N)** Right fibula LPB (FGGUB) R.2907, in lateral view. **(O)** Pedal phalanx LPB (FGGUB) R.2908, in dorsal view. Shaded skeletal elements indicate bones represented in the K2 assemblage (based on the generalized skeletal reconstruction of a Transylvaninan titanosaur from [5]).

**Testudines:** Turtle remains are among the most frequently recovered vertebrate fossils in the K2 collection, being represented by smaller or larger shell fragments, although cranial, axial and appendicular elements are yet to be identified. A relatively large and almost complete, but crushed turtle shell (LPB [FGGUB] R.2710) was also collected from the site, with both the plastron and carapace preserved; the carapace displays an ornamentation pattern that consist of tiny and low vermiculations and tubercles and is reminiscent of *Kallokibotion bajazidi* [86,87], suggesting that the K2 specimen most probably also belongs (or is closely related) to this common latest Cretaceous Transylvanian basal testudine. Turtle plate fragments showing the same ornamentation pattern are the most commonly occurring specimens in the K2 assemblage, showing that *Kallokibotion* sp. may have been abundant in the area. Meanwhile, a few turtle plates (e.g., LPB [FGGUB] R.2707 and R.2709) show slightly different ornamentation (fine, closely spaced, parallel longitudinal ridges, as well as wide and deep sulci) which may indicate that beside *Kallokibotion*, remains of another (probably dortokid) turtle taxon are also present in the material [11]; nonetheless, this second taxon appears to have been very subordinate in the local assemblage (see below, Taphonomy).

**Lizards:** A well-preserved maxilla (LPB (FGGUB) v.902; Fig 8 D-E), as well as isolated teeth and vertebra fragments of squamates were discovered at site K2 during screen-washing. From this material, only the maxilla shows diagnostic characters that allow a more precise taxonomic identification. According to [83], the morphological features of the maxilla are reminiscent of those of the endemic barbatteiids *Barbatteius* and *Oardasaurus*, also known from the Maastrichtian of the Transylvanian area [88,89]. However, the details of its heterodont dentition – characterized by a combination of unicuspid anterior teeth and tricuspid posterior ones – differ from those of the latter taxa, as the teeth of *Oardasaurus* were described as exclusively bicuspid [89], while in the case of *Barbatteius*, bi- and tricuspid teeth make up the dentition [88]. Based on these preliminary comparisons, the K2 maxilla may represent at least a new species of one of the above-mentioned genera or possibly even a new genus within Barbatteiidae [83]; its detailed study is pending.

**Crocodyliforms:** Crocodyliforms are particularly well represented in the K2 assemblage, albeit exclusively by isolated teeth, a commonly occurring phenomenon in many uppermost Cretaceous vertebrate accumulations from Hațeg Basin [17] as well as from the wider Transylvanian area [90–92]; in addition, crocodyliform teeth make up by far the largest part (about 80%) of the entire sample of isolated teeth recorded at this locality. Several different morphotypes can be identified within the crocodyliform tooth assemblage, suggesting both the presence of several different taxa (indicated by widely divergent and sometimes diagnostic teeth morphologies, as already hinted at by [11]) as well as that of potentially heterodont taxa (documented by morphological differences that could be accounted for as changes along the tooth series), again replicating situations reported previously from elsewhere all across the Hațeg Island area [8,15,17,18,90,92–96].

One of these crocodyliform tooth morphotypes is characterized by triangular, clearly latero-medially compressed blade-like teeth with erect, symmetrically placed and slightly lingually curved apex, a basal constriction, as well as coarsely serrated anterior and posterior carinae displaying a true ziphodont denticulation pattern (e.g., LPB (FGGUB) v.910; Fig 9A-B); such teeth had been first reported from Hațeg Basin by [97] and were referred to the ziphodont mesoeucrocodylian *Doratodon* [98]. Morphologically, these teeth are most similar to the third *Doratodon* tooth morphotype described by [99] from the Santonian of Hungary, and thus they probably represent elements of the middle to posterior dentition of a *Doratodon*-like crocodyliform. A second, related morphotype is exemplified by several specimens (e.g., LPB (FGGUB) v.907; Fig 9 C-E) that share with the former type the presence of serrated anterior and posterior carinae (although the individual denticles here appear to be less well developed and separated) as well as the roughly symmetrical triangular crown, but they are lower and more bulbous, less labio-lingually compressed (especially broadly bulging labially), with a worn tip. The enamel of these specimens is also more ornamented, with a large number of irregular, variably long apico-basal wrinkles. These specimens are reminiscent of the fourth *Doratodon* morphotype separated by [99], and thus they probably typify the posteriormost elements of the tooth row in the same *Doratodon*-like taxon.

These laterally compressed and ziphodont crocodyliform dental morphotypes, usually identified as *Doratodon* sp. teeth, are commonly reported from across Hațeg Basin [8,15,17,18,94,95,97] as well as other areas of Hațeg Island such as

Rusca Montană Basin [100] and the southwestern Transylvanian Basin [90,92,96]. In addition, we point out here a previously misidentified record of this morphotype, by noting that the tooth specimen UBB NgTh 1 reported from Rusca Montană Basin by ([101]: Fig 31.4.B) as belonging to a 'troodontid-like' theropod most likely also represents the *Doratodon* morphotype, given the marked differences between this specimen and typical serrated troodontid teeth [102–104], including: (1) upright, centrally placed apex versus a strongly recurved one; (2) sub-equal denticles on the anterior and posterior carinae versus smooth or at most very finely denticulated anterior carina associated with a coarsely serrated posterior one; and (3) relatively closely packed and subrectangular posterior denticles versus largely separated, distally pointed and hooked ones.

A third crocodyliform tooth morphotype recognized at K2 includes high, conical and relatively laterally compressed teeth, with their apex curved lingually but also somewhat distally relative to the midline of the crown. As a result, the anterior edge of the crown is longer and more convex than the posterior one, although the degree of these curvatures (and especially that of the anterior/posterior asymmetry) varies between the different specimens (e.g., LPB (FGGUB) v.915, v.916, v.917, v.925; Fig 9F-H). The labial side of the crown is usually more convex than the lingual one, but the relative development of the two sides varies among the specimens. Most of these specimens are fairly complete, some even retaining the root as well (e.g., LPB (FGGUB) v.915; Fig 9F-H). They are relatively small, reaching about 5–6 mm in apicobasal height and 2.5–3 mm in anteroposterior basal length in some cases, although certain specimens (e.g., LPB (FGGUB) v.925) are even smaller, at 3.1 mm height and 1.4 mm basal length. The anterior and posterior carinae are relatively poorly developed and sometimes bear weakly marked pseudoziphodont crenulations. The enamel is largely smooth in some of the specimens, but variably apico-basally wrinkled in others. The size and overall morphology of the teeth representing this morphotype are reminiscent of the anterior caniniform teeth of 'Theriosuchus-like' basal neosuchian taxa reported previously from Hațeg Basin, such as *Sabresuchus* ('Theriosuchus') *sympiestodon* [105,106] and *Aprosuchus ghirai* [107]. Accordingly, this morphotype is thus classified here provisionally as 'Theriosuchus-like', without a more precise taxonomic identification, given the presence of at least two roughly contemporaneous taxa with largely similar dentition in Hațeg Basin, as well as the still ambiguous phylogenetic position of these taxa [107,108]. Teeth of this general morphology occur commonly in Hațeg Basin [17, 106], and were also reported from Rusca Montană Basin [91] and the Transylvanian Basin [96]. A morphologically and dimensionally similar isolated crocodyliform tooth had also been reported by ([15]: pl. III.10) from the Budurone MvBB south of Vălioara, located somewhat higher up-section than K2 within the lower part of the Densuș-Ciula Formation Middle Member [20], and was referred to the basal eusuchian *Allodaposuchus precedens*. However, its minute size (height 2.2 mm, antero-posterior basal length about 1 mm), highly conical shape with a slightly distally recurved apex, and the presence of weak and smooth apicobasal carinae argue more convincingly for its 'Theriosuchus-like' affinity, and thus extend the record of this morphotype to this site as well.

Specimen LPB (FGGUB) v.917 from the previously described sample marks a transition towards a largely similar fourth crocodyliform dental morphotype from site K2, represented by specimens such as LPB (FGGUB) v.923 and v.924 (Fig 9 IJ). These are of roughly the same size and overall morphology as those identified above as 'Theriosuchus-like' caniniforms, but differ from these in that they have better developed, prominent and blade-like anterior and posterior carinae that are further emphasized by the presence of apico-basally elongated furrows bordering them towards the midline of the crown; these furrows are better developed, deeper and wider on the lingual side. The presence of these carinae also further accentuates the basal constriction at the crown-root junction. The enamel is sparsely ornamented with apico-basally extending but discontinuous wrinkles, some of which extend onto the carinae and develop into weakly expressed preudoziphodont denticulations. In LPB (FGGUB) v.924 (Fig 9 IJ), the apical part of the labial face is also marked by the presence of coarser, apicobasally extending rounded ridges, with better developed, more prominent and widely spaced ridges being intercalated with several finer and somewhat discontinuous ones; the base of what appear to be similar ridges are also visible in the apically less completely preserved LPB (FGGUB) v.923. Whether these labial ridges extend to the tip of the crown or not remains unclear as v.924 is also incomplete apically. Despite their slightly different morphology, these teeth are also identified here as representing a 'Theriosuchus-like' morphotype; certain caniniforms of *Sabresuchus*

*sympiestodon* such as the hypertrophied fourth maxillary tooth [105,106] do present somewhat better developed marginal carinae, although apparently not to the extent seen in these K2 specimens.

A fifth crocodyliform tooth morphotype recognized in the K2 sample is represented by moderately labio-lingually compressed teeth with a high triangular profile and slightly lingually recurved apex (e.g., specimens LPB (FGGUB) v.918, v.926, and v.931, among others; Fig 9 K-L); their height-to-basal length ratio is somewhat lower than those of the two previously discussed morphotypes (closer to 1.5 than to 2), but otherwise display several features reminiscent of those, including overall small size (3–5 mm height, at most 3.5 mm basal length), presence of a basal constriction, as well as a biconvex cross-section with a more bulging labial side and a slightly convex lingual side that is excavated marginally by longitudinal troughs bordering the marginal carinae. The carinae are smooth to very slightly crenulated, in a pseudoziphodont manner, as occasionally some of the mainly apicobasally aligned, discontinuous wrinkles that ornament the enamel extend onto the carinae themselves. Teeth displaying similar morphology had not been reported yet *in situ* associated with one or the other of the Transylvanian '*Theriosuchus*-like' taxa, but are here tentatively also identified as representing this taxonomic category, and probably a more posterior tooth position than the caniniforms described previously.

Remarkably, however, the K2 crocodyliform assemblage did not yielded any specimen comparable with the very characteristic leaf-shaped – i.e., low-crowned, labiolingually compressed, triangular and pseudoziphodont – posterior teeth of the Transylvanian '*Theriosuchus*-like' taxa, and which are reported preserved *in situ* in the cases of both *S. sympiestodon* and *A. ghirai* [105–107]. This is at odds with previous reports according to which such teeth are frequently preserved isolated in different uppermost Cretaceous localities of Haţeg Basin [17,106]) as well as in other areas of Haţeg Island such as Rusca Montană Basin [17]). Notably, a large number of isolated crocodyliform teeth from the Oarda de Jos A (ODAN) MvBB, in the southwestern Transylvanian Basin [60,109,110] have been described briefly and figured by [90] (pl. XI 7–17), and these were referred by this author to *Doratodon.* The corresponding teeth can, nonetheless, actually be reassigned as '*Theriosuchus*-like', as they lack the definitively ziphodont, coarsely denticulated carinae that characterize the dentition of the former taxon [98,99]. Instead, several of these specimens display the highly conical morphology of the anterior caniniform morphotype (morphotypes 3 and 4), while others correspond to the highly triangular morphotype 5, both of these being reported from site K2 (see above), meanwhile a large part of the ODAN sample represents the typical low-crowned and leaf-shaped distal dental morphotype of these basal neosuchians, clearly indicating the presence of '*Theriosuchus*-like' crocodyliforms at Oarda de Jos.

A sixth crocodyliform dental morphotype is exemplified by usually larger-sized (sometimes up to over 1.5 cm high, despite being only partially preserved), conical teeth with a pointed apex that is deflected lingually and only very slightly distally. None of these specimens (e.g., LPB (FGGUB) v.911, v.912; Fig 9M-O) are complete, but nevertheless they preserve enough of their morphology to show that they are almost uncompressed labio-lingually and are lined by low, unserrated anterior and posterior carinae; their sides, and especially their lingual surface, can be ornamented by well-marked, rounded apicobasal ridges (see also [11]: fig 11C). Based on their relatively large size compared to most other crocodyliform teeth from K2, as well as their morphology, these can be most probably identified as anterior teeth of *Allodaposuchus* [111]), a basal eusuchian taxon whose original lectotype material was also discovered in the Vălioara area, and most probably even somewhere close to site K2 both geographically and stratigraphically [11,112,113]. Teeth corresponding to this morphotype are routinely found isolated in the different uppermost Cretaceous vertebrate localities across the former Haţeg Island [11,17,90,94,96,114]) and represent one of the most commonly encountered type of fossil remains in these deposits [4].

A seventh crocodyliform dental morphotype identified at K2 is represented by LPB (FGGUB) v.913 (Fig 9 P-R), another relatively large, completely preserved tooth (10 mm high, 9 mm basal length). Unlike the previous morphotype, however, it is a low and stout, somewhat bulbous tooth with a rounded apex, quasi-equally convex labial and lingual sides, and weak, albeit crenulated marginal carinae; the enamel is not fluted, as in the previously discussed morphotype, but is instead ornamented by apicobasally oriented wrinkles, some of which converge on the carinae producing pseudoziphodont denticulations. Based on their large size and overall shape, such globidont teeth, also commonly encountered in uppermost

Cretaceous vertebrate fossil sites across Haţeg Island, are usually identified as the posteriormost teeth of *Allodaposuchus* sp. (e.g., [114]), albeit none of these is preserved *in situ* in either the formerly recognized lectotype from Vălioara [111] (Buscalioni et al., 2001), or in the designated neotype from Oarda de Jos [115]. We agree that this appears to be the most reasonable identification for LPB (FGGUB) v.913 as well.

Finally, three further crocodyliform dental morphotypes identified in the K2 assemblage share the combination between relatively small dimensions and low, rather bulbous crowns, although the degree of apico-basal depression, respectively labio-lingual bulging changes progressively between them. The first of these morphotypes, already reported from site K2 by ([11]: fig 11D), includes teeth that are moderately low-crowned with roughly equal apico-basal height and antero-posterior basal length (2.6 mm x 2.5 mm in LPB (FGGUB) v.885; Fig 9 S-V) and a spatulated morphology. Their labial side is markedly bulging whereas the lingual side is convex (although to a lesser degree than the labial one) for most of its extent, but becomes shallowly concave towards and below the widely rounded and slightly lingually leaning tip; the tip itself shows a small wear facet apically. The crown presents well-developed, smooth marginal carinae that are emphasized by apicobasally extending basal troughs separating these from the central, more bulging parts of the labial/lingual surface. While one of the carinae runs uninterrupted from the apical region to the slightly constricted base of the crown, the other is replaced above mid-height by a flattened vertical platform that extends both labially and lingually (Fig 9 S,U,V -pf). This morphotype had been first reported from Haţeg Basin by [97] from the Fântânele MvBB, a site located north of Vălioara and placed stratigraphically somewhat higher than site K2 in the lower part of the Densuş-Ciula Formation Middle Member [20], and was identified as part of the anterior dentition of an indeterminate species of the basal eusuchian genus *Acynodon* [116,117]. Subsequently, spatulated teeth reminiscent of this morphotype had been also reported from the ODAN MvBB from Oarda de Jos, Transylvanian Basin, by [90], although these appear to lack the flattened marginal platform along one of the carinae. In accordance with these previous reports, and also since these K2 teeth are largely similar to those described *in situ* in different nominal species of *Acynodon* from France [118] and Italy [119], we identify them as premaxillary and/or anterior maxillary teeth of *Acynodon* (or 'Acynodon-type').

The second morphotype from this series is represented by specimen LPB (FGGUB) v.906 (Fig 9 XZ), a very low-crowned tooth, with a subunitary apicobasal height-to-anteroposterior basal length ratio (0.68) and extremely bulbous, globidont overall appearance (due to the sub-equal labiolingual width and anteroposterior basal length). Both the labial and the lingual sides are markedly convex, to an almost identical degree, and are covered by a network of prominent and rounded, apico-basally extending parallel ridges that converge onto the apical margin; these are more prominent on the lingual face. The apical part of the crown is a widely convex arch, with a minute subcentrally placed inflexion point as tip; despite its more markedly bulging nature, the lingual face still preserves a slightly concave area right below the apical arch. This arch merges downward into the anterior and posterior carina, respectively, each of which are still emphasized lingually by moderately deep depressions separating it from the central bulging region of the lingual face. The presence of the carinae accentuates the basal constriction of the crown at the crown-root junction. The many similarities that exist between LPB (FGGUB) v.906 and the specimens identified above as anterior teeth of *Acynodon*, as well as their commensurate size, suggest that all these specimens most probably belong to the same taxon, whereas the differences noted indicate that the currently discussed morphotype typifies the middle-distal part of the tooth row.

The third morphotype from the series, exemplified by three specimens (LPB (FGGUB) v.903-v.905), represents the culmination of the morphological modification trends noted for the previous two morphotypes. The crowns are extremely low and wide, with an occlusal surface that is sub-quadrangular-to-oval in outline, equally well bulging both labially and lingually (Fig 9 AA-AB). In specimen v.904, an antero-posterior apical ridge crossing the occlusal surface is still present, although it is largely removed by wear and/or erosion, but the anterior and posterior carinae are no longer obvious. The labial and lingual surfaces are both widely convex and covered by several prominent, apico-basally extending ridges. In the other two specimens, the crown is entirely button-like, tribodont in overall morphology, with the widely

sub-quadrangular to oval occlusal surface shaped as a well-developed horizontal platform. A number of coarse ridges cresting all the sides converge onto the margin of the horizontal occlusal plate; the latter is excavated in its central part by an antero-posteriorly elongated depression marking the position of the lost/worn apical crest. Between the two specimens, the occlusal surface of v.905 is still relatively wider and less elongated, with a shorter oval contour, whereas that of v.903 is already relatively more elongated and sub-quadrangular. Similar coarsely ribbed tribodont teeth have been reported previously from several microvertebrate localities across Haţeg Basin, both near Vălioara [120] and at Pui [94,114], as well as in the southwestern Transylvanian Basin, at Oarda de Jos [90], and were convergently interpreted as elements of the posterior dentition of *Acynodon* sp. based on similarities with *in situ* posterior crushing teeth found in *Acynodon iberocci-tanus* [118], an identification with which we also concur in the case of the K2 morphotype discussed here. Furthermore, the detailed differences noted between the three specimens representing the tribodont morphotype suggest that v.904 is probably the anteriormost element of these, whereas v.903 (Fig 9 AA-AB) is the posteriormost one, based on comparisons with the morphological progression reported in *A. iberoccitanus* by [118].

Finally, a number of crocodyliform teeth (LPB (FGGUB) v.920-v.922; Fig 9AC-AE) from site K2 display a morphology that is difficult to univocally allocate to one of the major taxonomic categories identified previously (i.e., *Doratodon*, '*Theriosuchus*-like', *Allodaposuchus*, and *Acynodon*). They are generally very weakly labio-lingually compressed, robust triangular-conical teeth, with a centrally placed, blunt apex and anterior/posterior carinae that are usually better developed, more trenchant basally but diminish or even disappear towards the tip; the enamel is relatively coarsely ornamented with apico-basally extending lines (v.920, v.921) or small ribs (v.922). Nonetheless, there are also certain dimensional or morphological differences between these specimens that can be arranged theoretically into a serial progression. Specimen v.920 is the smallest in absolute terms, slightly taller than long (4 mm apicobasal height to 3.6 mm anteroposterior basal length; Fig 9 AC-AE), with a somewhat more pointed (albeit still blunt) and lingually leaning apex underlain by a depressed area so that the lingual face has a somewhat sinusoidal aspect in anterior/posterior view, and a labial face that is clearly more convex than the lingual one. Specimen v.921 is somewhat larger, but also relatively lower (apicobasal height and anteroposterior basal length are about equal, at 4.7 mm), the apex is blunter and less lingually tilted but still undercut by a weak concavity, and the lingual face is almost as bulging in anterior/posterior view as the labial one. Finally, v.922 is the largest but also the relatively lowest one in the series (5.3 mm high, 6.8 mm long basally), with a widely arched and almost erect tip, and quasi-equally convex labial and lingual sides.

The absence of sharp, marked carinae bearing clearly defined, ziphodont denticulations, as well as the relatively robust, bulbous shape of these teeth preclude their referral to *Doratodon*. The combination between a moderately tall to low triangular shape and a great labio-lingual width, together with the absence of any clearly marked pseudoziphodont crenulations on the carinae suggest that they probably do not correspond to any of the '*Theriosuchus*-like' morphotypes, either; meanwhile, their relatively small size and low, robust shape shows they are not anterior *Allodaposuchus* teeth. They (and especially the lowest-crowned specimen v.922) are similar in some respects to the posterior, bulbous teeth of the *Allodaposuchus* morphotype, but are still smaller than those, and the somewhat wavy, sinusoidal aspect of the lingual face seen in this morphotype is not present in teeth assigned to *Allodaposuchus*. Finally, they also show some resemblances to the previously described spatulated to globidont morphotypes of *Acynodon*, but for their larger size and the relatively poorer development of the anterior/posterior carinae; furthermore, they are relatively thicker labio-lingually compared to the spatulated morphotype, but taller than the otherwise equally labio-lingually robust globidont morphotype. Morphologically, they are somewhat more reminiscent of the middle maxillary teeth of *Acynodon adriaticus* [121], although appear to lack the finely but densely wrinkled enamel ornamentation of the latter. It is conceivable that these teeth represent either members of the serial dental variation present in one or another of the previously separated morpho-taxa (excepting most probably *Doratodon,* though) or juvenile teeth of the larger-sized *Allodaposuchus*, but it is equally possible that they represent a further, distinct morpho-taxon present at this locality, one that may be most closely related to *Acynodon* (systematically and/or paleoecologically).

**Rhabdodontid ornithopod dinosaurs:** Associated and partly articulated cranial and postcranial elements of rhabdodontids recovered from site K2 (Fig 10), originating from at least two individuals of the same taxon, were already described in detail by [12]. The K2 rhabdodontid shares several similarities with specimens referred to *Zalmoxes shqiperorum* in earlier publications [122,123], such as the wide buccal shelf of the dentary, the dorsoventrally strongly expanded scapular blade, and the dorsoventrally low, but anteroposteriorly elongated postacetabular process of the ilium. However, some of the skeletal elements also bear features that are reminiscent of the other previously separated *Zalmoxes* species, *Z. robustus*, including the narrow acetabulum and the slightly laterally bowed tibia [122]. Furthermore, the K2 material exhibit features that differentiate it from both the above-mentioned species, such as the very slight lateral expansion of the dorsal edge on the ilium. Finally, the K2 material also shows diagnostic differences from the recently described third Transylvanian rhabdodontid, *Transylvanosaurus platycephalus* [124], as the K2 frontals – the only overlapping element shared with this later taxon – display differentiated nasal and prefrontal articular surfaces, and do not present a strong transversal crest at the posterior edge of these facets. Accordingly, the K2 rhabdodontid material appears to be distinct from other previously known Transylvanian rhabdodontids, at least at a species level, and thus – pending the results of ongoing investigations concerning other important Transylvanian rhabdodontid specimens – the taxon represented by this material is currently referred to only as Rhabdodontidae indet.

**Hadrosauroid ornithopod dinosaurs:** Besides two isolated teeth (LPB [FGGUB] R.2888, R.2889), only a small dentary fragment (LPB [FGGUB] R.2887; Fig 8J) referable to hadrosauroids was recovered from site K2. The teeth are either heavily eroded or else attached to another element representing a different taxon (Fig 8K), and therefore their detailed observation is not possible. These specimens could be identified as hadrosauroid teeth based on their distinctive diamond-shape morphology and the presence of denticles along the mesial and distal edges.

The right dentary fragment preserves the tall, dorsally expanded coronoid process and a small, posteriormost part of the apparently straight tooth row with six tall, vertical alveoli, albeit without *in situ* preserved teeth (Fig 8J). The alveoli are separated by straight but slightly posteriorly oriented, narrow sheet-like and parallel septa.

The specimen is rather poorly preserved. The coronoid process is laterally offset from the tooth row; in lateral view, its basal part bends posteriorly, but is then redirected anteriorly around mid-length. On the medial side there is a dorsoventrally oriented groove, and posteriorly the process borders the large mandibular adductor fossa. The articulation area with the surangular reaches the dorsal part of the coronoid process. The K2 specimen shares these features both with the Transylvanian hadrosauroid *Telmatosaurus transsylvanicus* [125,126] and with the potentially new taxon from the slightly younger Fântânele 3 site [20], north of Vălioara [127]. Nonetheless, it also shows certain differences, at least compared with *T. transsylvanicus,* as the dorsal tip of the coronoid process is more anteroposteriorly expanded and its dorsal edge is straight; this part is not preserved in the Fântânele 3 hadrosauroid. Even though the K2 specimen is poorly preserved, and can only be referred to as Hadrosauroidea indet., is worth noting that the posterior part of the coronoid process dorsal edge appear to display a posterodorsal point, reminiscent of another dentary fragment from the Vălioara region, specimen SZTFH v.13527 [128]. A straight dorsal edge of the coronoid process with a caudodorsal point or a triangular caudal apex, is characteristic for individuals representing different derived hadrosaurids such as some adults of certain brachylophosaurins (at least *Maiasaura* and *Brachylophosaurus,* [129]), as well as select individuals of *Parasaurolophus*, *Tlatolophus* and *Edmontosaurus* from North America, and *Arenysaurus* from Europe [130,131]. Furthermore, a similar posterodorsal process-like structure, albeit associated with a rounded dorsal edge, also occurs in the western European basal hadrosauroid *Fylax thyrakolasus* [132], to which these two specimens from Vălioara may be more closely related on paleogeographic grounds; however, testing such a hypothesis would require the discovery of more complete remains referable to these taxa, both in Romania and in Spain.

**Titanosaurian sauropod dinosaurs:**   Aside the turtle plate fragments, titanosaur remains are the most common elements in the K2 assemblage. The more than one hundred titanosaur skeletal remains collected from the site include dorsal and caudal vertebrae, a part of the right side of the sacral region, ribs, one pubis fragment, two humeri (from the

right and left side), three femora, one fibula, and two metatarsals (Fig 11). The well-preserved titanosaur remains are dominated by the dorsal and caudal vertebrae, several of which were preserved associated or even in articulated position [11]. The presence of three similar sized femora indicates that several individuals are represented in the assemblage, and based on its taphonomical features (as discussed below) we suggest that this material indeed represents the associated incomplete skeletons of at least two titanosaur. Earlier, the K2 titanosaur material was tentatively assigned to a *Paludititan*-like taxon based on the presence of amphiplatyan mid-caudal vertebrae inserted within the procoelous series [11], a feature reported to be autapomorphic for this taxon described previously from Nălaţ-Vad in Haţeg Basin [133]. However, more detailed comparisons focusing on the morphology of the neural arches [134], the discovery of further titanosaur remains at site K2, as well as the detailed revision of the entire titanosaur material from Haţeg Basin [5], question the validity of this assignment, and based on the currently accepted diagnoses of the Romanian titanosaurs it can be suggested that the K2 material is more reminiscent of *Magyarosaurus*. A detailed study of the K2 titanosaur material, currently underway, is of particular importance, as it represents the most complete and well-documented associated skeletal assemblage for this group known from the entire basin, and as such, it can help to better understand the affinities and phylogenetic relationships of the Transylvanian titanosauris.

**Theropod dinosaurs:** Theropod remains are extremely rare in the K2 assemblage, as only very few isolated tooth fragments indicate reliably their presence at the site. Among the collected teeth, one of the better preserved specimens (LPB (FGGUB) R.2884; Fig 8H-I), albeit represented only the apical part of the crown, is a small tooth fragment about 2.5 mm in height. It is morphologically similar to the small-sized *incertae sedis* coelurosaur *Richardoestesia* [102], a taxon that was reported previously from several sites of Haţeg Basin (e.g., [8,17,135,136]) as well as other areas of Haţeg Island [17]. Similarities with *Richardoestesia* include a nearly straight crown shape, slightly flattened labio-lingually, and only serrated on the posterior carina, while the anterior edge is transversely rounded; its morphology suggests a rather tall, apico-basally elongated and antero-posteriorly narrow crown when complete, with both the labial and lingual sides evenly convex so that the cross-section was a widely rounded oval. The denticles are relatively small, straight and rectangle-shaped, apico-basally longer than tall, and terminally mildly convex. There is no significant variation in size among the serrations, with about 5 denticles per 0.5 mm, although these do become slightly lower, more rounded and somewhat more widely separated, progressively attenuating in size towards the tip which is not reached by the denticulation. Another, more basal tooth fragment (LPB (FGGUB) R.2886) is reminiscent of the former specimen in its overall cross-section, with both lateral sides evenly rounded, as well as in the presence of evenly sized and spaced denticles only on the posterior edge. These denticles are, nonetheless, somewhat smaller than those seen on R.2884, with about 7.5 denticles per 0.5 mm, possibly indicating – if these specimens do indeed represent the same taxon, which seems likely – a reduction in denticle size towards the base of the crown, similarly to the trend reported towards the apex.

Besides the *Richardoestesia*-like theropod teeth, another, morphologically markedly different and larger (preserved height 5 mm, antero-posterior basal length 5.2 mm) theropod tooth specimen was also collected from the site. This tooth fragment, coming from the basal part of the crown (LPB (FGGUB) R.2885; Fig 8F-G), is strongly laterally compressed and shows serrated carinae on both the anterior and the posterior edges. The somewhat flattened labial and lingual surfaces bear a shallow midline longitudinal depression that extends in the apicobasal direction, giving the crown a mildly 8-shaped cross-section. The serrations of the anterior edge are restricted to the apical part of the preserved fragment, replaced more basally by a widely rounded anterior edge lacking a carina, while the more prominent posterior carina extends farther basally and is serrated throughout. The denticles of the posterior carina are much more pronounced, larger (about 3.5 denticles per 0.5 mm) and especially taller than those on the anterior carina (about 5 denticles per 0.5 mm). Although the crown is incomplete, the anterior edge seems to present a significantly more important curvature along its length than the posterior edge, suggestive of an overall posteriorly recurved crown outline. All these features suggest that this tooth fragment is most probably related to a velociraptorine dromaeosaurid, a tooth morphotype frequently reported in the past both from across Haţeg Basin (e.g., [8,17,135–137]) and from the wider Transylvanian area as well (e.g., [90,94,138]).

Accordingly, based on the different tooth morphologies recognized, at least two small-sized theropod taxa can be identified in the K2 assemblage. Besides the teeth reported here, further potential theropod material collected from the site includes at least three limb bone fragments (a tentative identification based on their thin bone walls), although these are too poorly preserved for a detailed identification, and one anteroposteriorly elongated amphyplatian posterior caudal vertebral centrum (LPB [FGGUB] R.2890) without a neural spine. It is octagonal in cross section and has a longitudinal canal on the ventral side, making this specimen very similar to caudals of other paravian theropods [139].

**Pterosaurs:** Among the specimens with possible pterosaurian origins there are a few crushed, elongated limb bone fragments with very thin bone walls (e.g., LPB (FGGUB) R.2891; Fig 8K), as well as a small, fragmentary fossil (LPB [FGGUB] R.2892) with a complex but somewhat rectangular morphology. This latter specimen has two asymmetrically placed concave articular surfaces on its (?) proximal side and a slightly roughened surface on its (?)distal side. Because of this irregular and asymmetrical structure with rugosities, it is most reminiscent to a carpal (maybe a proximal carpal) of a derived pterosaur [140].

**Multituberculate mammals:** Only one isolated multituberculate tooth was identified among the microvertebrate material recovered from site K2 (LPB (FGGUB) M.1710; Fig 8L-M); nonetheless, this specimen makes site K2 the tenth multituberculate occurrence known from the Densuş-Ciula Formation (of which, notably, 7 are located in the immediate neighbourhood of Vălioara), and the 25th such site from Haţeg Basin overall (see [19,141,142]). It is a relatively small (1.3 mm x 0.84 mm), bicuspid tooth specimen showing the complete, well-preserved and pristine crown but only the base of the anterior root. Its morphology and extremely reduced cusp formula strongly indicate that it most probably represents a left first upper premolar (P1; see [14]). In occlusal view, it has a mesio-distally elongated oval outline, with a slightly convex labial and a more convex lingual margin. The mesial margin is rounded, but marked with a lip-like mesial extension in its labial half; the distal margin is more elongated, roundedly pointed and asymmetrical. The two conical, slightly mesially leaning cusps form a transversely aligned row, with the labial one being slightly smaller than its lingual counterpart. Mesial to the transverse cusp row, the occlusal surface of the crown shows a flattened platform extending towards the mesial margin; distally, the cusps are bordered by a similar, but more extensive, elongated platform that is deeply excavated on its distal half, suggesting that the succeeding P2 had a rather well-developed mesial heel overlapping this area, as often reported in the Transylvanian latest Cretaceous kogaionids (e.g., [143,144]). The enamel of the cusps is ornamented with a small number of divergent radial ridges; the cusps themselves are complete, showing at most a very minor amount of wear on their tips.

The general morphology of this P1, and especially its very reduced cusp formula, suggests that it belongs to a kogaionid, the only multituberculate clade reported so far from the uppermost Cretaceous of the Transylvanian area [19]. As currently known, kogaionids are most often characterized by a tricuspid P1 (e.g., [143–147], although in a very few cases, bicuspid P1s were also reported previously, both from the uppermost Cretaceous of Transylvania [14,19,138] and from the Paleocene of western Europe [148,149]; unfortunately, however, none of these specimens is known preserved *in situ* in the maxilla. Remarkably, several latest Cretaceous examples of bicuspid P1s were reported from areas (and sometimes stratigraphic levels) placed close to those of site K2, especially in the case of the specimens reported by [14] from the Fântânele microvertebrate bonebed north of Vălioara, and referred by them to *Hainina* sp. A. Nevertheless, most of the bicuspid P1s on record, including those from Fântânele, depart from the K2 specimen in that their crowns have a quasi-circular outline in occlusal view, unlike the elongated oval outline of LPB (FGGUB) M.1710.

In this respect, it is most similar to specimen UBB Ng1–02 from Negoiu in Rusca Montană Basin, reported by [138] and referred by them to the small-sized kogaionid *Barbatodon oardaensis;* this specimen also closely matches the K2 P1 in size, although it appears to have somewhat less well developed mesial and especially distal platforms. The type material of *B. oardaensis*, however, includes only one tricuspid P1, indicating that the bicuspid specimens from the Rusca Montană and Haţeg areas may not actually belong to this taxon, despite the fact that [138] suggested the occurence of cusp formula polymorphism due to intraspecific variation in this taxon to account for the different P1 cusp numbers documented in

the different assemblages. Due to these uncertainties, as well as to the presence of some minor differences between the K2 P1 and that from Negoiu, we refrain for the moment to consider these specimens to be conspecific.

Nonetheless, in the same time we emphasize once again the clear differences in size, occlusal face contour and/or cusp formula between the K2 specimen and other kogaionid P1s reported from the Vălioara area previously [14,16]; while the bicuspid ones from the Fântânele MvBB are smaller and have a roughly circular outline, other P1 specimens that are of comparable size and shape are tricuspid instead, both at the Fântânele and the Fântânele 2 MvBBs. Accordingly, it appears probable that at least locally, in the basal part of the Densuş-Ciula Formation cropping out in the Vălioara area, the K2 kogaionid differs from other kogaionids whose presence was reported from here previously, although more material from this and other sites would be needed to substantiate such an assessment.

## 4.5. Taphonomy

During the three-year main excavation period, approximately 4.75 m$^2$ area of the bonebed was excavated and investigated in detail at site K2, with the excavations resulting in 872 recovered vertebrate remains (S2 Table), all of which were collected from a single well-defined sandstone bonebed horizon. Of these, 441 bones and teeth can be identified taxonomically, representing at least 17 different lower-level taxa (Tables 2 and 3).

### 4.5.1. Number of identified specimens (NISP).

**Method:**   The NISP value includes all vertebrate remains that were taxonomically identified at least at ordinal level; thus, the unidentified bone pebbles were excluded from this calculation (following [150]).

**Results:**   As already noted, from the recovered vertebrate remains 441 specimens were identified osteologically and taxonomically (NISP = 441; Fig 12A-B; Tables 2 and 3). The taxonomic abundances based on NISP show that remains of titanosaurs (32%) and of the basal turtle *Kallokibotion* (28%) represent the most commonly identified elements in the K2 assemblage (Table 3). Dinosaur remains make up more than half of the collection (52% of the NISP; Fig 12A), dominated by those of titanosaurs (62% of all dinosaur remains) and rhabdodontids (27%) (Fig 12B). Turtles represent the second (29%), and crocodyliforms, the third most abundant group (15%), whereas the amounts of fossil specimens assigned to other taxa (fishes, amphibians, squamates, pterosaurs, other dinosaurs, and mammals) are small and approximately equal (about 1% of NISP) (Fig 12A). Remarkably, the vertebrate assemblage of site K2 is dominated by terrestrial faunal elements (especially dinosaur remains), while other, presumably aquatic or semi-aquatic taxa (e.g., anurans, albanerpetontids, the crocodyliforms *Allodaposuchus* and *Acynodon*; representing about 5% in total, according to the NISP count) are only subordinately present.

**Comments:**   The low frequency of the remains of semi-aquatic – aquatic groups compared to terrestrial ones may indicate that the depositional environment of site K2 was a spatially localised and probably short-lived fluvio-lacustrine paleoenvironment where a diverse and well-established aquatic paleocommunity could not develop, and the remains of aquatic forms either accumulated at a slow pace attritionally or were transported into the site by periodic floods from more distal areas. On the other hand, it should be mentioned that several of the better-known uppermost Cretaceous local vertebrate assemblages from Haţeg Basin (and the Transylvanian area, overall) show a predominance of terrestrial groups (e.g., [4,8]), although such a trend is less well expressed in the case of the microvertebrate accumulations (e.g., [17,90]).

Turtle plate elements are usually among the, if not the, most common elements in fossil vertebrate sites of Haţeg Basin, due to the high preservation potential and easily diagnosable nature of the specimens, and this is also the case at K2; their large numbers are not necessarily indicative of their original super-high abundance within the local biota. However, the significant difference in abundance between dortokid-like (NISP = 2) and *Kallokibotion* (NISP = 124) shell fragments most probably does reveal certain paleoecological preferences and/or taphonomic biases between these two turtles (Table 3). The preservation mode (associated, and even articulated shell parts) and high frequency (about 28%, according to the NISP) of the *Kallokibotion* remains suggest that these were accumulated through taphonomical processes similar to those concentrating the fossils of terrestrial dinosaurs. Meanwhile, dortokid remains – similarly to other

**Table 2. Skeletal parts and taxon representation in the site K2 vertebrate assemblage.**

| | Actinopterygii | Albanerpetontidae | Anura | Squamata | Testudinata | Crocodyliformes | Ornithopoda indet. | Rhabdodontidae | Hadrosauroidea | Titanosauria | Theropoda | Pterosauria | Multituberculata | Indet. | Total |
|---|---|---|---|---|---|---|---|---|---|---|---|---|---|---|---|
| Skull elements | 0 | 0 | 0 | 0 | 0 | 3 | 2 | 8 | 0 | 3 | 0 | 0 | 0 | 5 | 21 |
| Maxilla | 0 | 0 | 0 | 1 | 0 | 0 | 0 | 1 | 0 | 0 | 0 | 0 | 0 | 0 | 2 |
| Dentaries | 0 | 1 | 0 | 0 | 0 | 0 | 0 | 3 | 1 | 0 | 0 | 0 | 0 | 1 | 6 |
| Isolated teeth | 0 | 0 | – | 1 | – | 57 | 0 | 6 | 5 | 0 | 3 | 0 | 1 | 0 | 73 |
| Vertebrae | 0 | 0 | 0 | 1 | 0 | 2 | 1 | 13 | 0 | 97 | 1 | 0 | 0 | 25 | 140 |
| Ribs | 0 | 0 | ? | ? | – | ? | 0 | 14 | 0 | 16 | 0 | 0 | 0 | 60 | 90 |
| Pectoral girdle elements | 0 | 0 | 0 | 0 | 0 | 0 | 0 | 1 | 0 | 2 | 0 | 0 | 0 | 0 | 3 |
| Pelvic girdle elements | 0 | 0 | 0 | 0 | 0 | 0 | 0 | 3 | 0 | 2 | 0 | 0 | 0 | 0 | 5 |
| Forelimb bones | 0 | 0 | 1 | 0 | 0 | 1 | 1 | 2 | 0 | 2 | 0 | 0 | 0 | 0 | 7 |
| Hind limb bones | 0 | 0 | 0 | 0 | 0 | 0 | 1 | 4 | 0 | 4 | 1 | 0 | 0 | 0 | 10 |
| Limb bone fragments | 0 | 0 | 0 | 0 | 1 | 2 | 3 | 1 | 0 | 0 | 1 | 1 | 0 | 16 | 25 |
| Metapodia | 0 | 0 | 0 | 0 | 0 | 0 | 4 | 7 | 0 | 2 | 0 | 1 | 0 | 1 | 15 |
| Osteoderms/Plates | 0 | – | – | 0 | 125 | 2 | – | – | – | 0 | – | 0 | 0 | 0 | 127 |
| Ganoid scales | 4 | – | – | – | – | – | – | – | – | – | – | – | 0 | 0 | 4 |
| Unidentified bones | 0 | 0 | 0 | 0 | 0 | 1 | 0 | 0 | 0 | 15 | 1 | 3 | 0 | – | 20 |
| **NISP** | 4 | 2 | 1 | 3 | 126 | 68 | 12 | 63 | 6 | 143 | 7 | 5 | 1 | – | 441 |
| **MNI** | 1 | 1 | 1 | 1 | 3 | 4 | – | 2 | 1 | 2 | 2 | 1 | 1 | – | 20 |
| **Number of taxa** | 1 | 1 | 1 | 1 | 2 | 4 | – | 1 | 1 | 1 | 2 | 1 | 1 | – | 17 |
| **Taxa represented** | Lepisosteiformes indet. | Albanerpetontidae indet. | Anura indet. | Barbatteiidae indet. | Kallokibotion sp.; Dortokidae indet. | Allodaposuchus sp; Acynodon sp.; Theriosuchus-like sp.; Doratodon sp. | Rhabdodontidae indet. AND/ OR Hadrosauroidea indet. | Rhabdodontidae indet. | Hadrosauroidea indet. | Titanosauria indet. | Richardoestesia sp.; Velocireptorine dromeosaurid indet. | Pterosauria indet. | Kogaionidae indet. | | |
| **References** | This study | [83] | This study | [83] | [11] | [11] and this study | This study | [12] | This study | [5,11] and this study | This study | This study | This study | | |

**Table 3. NISP and MNI values as well as fragmentation rate associated with taxa in the K2 assemblage.**

| Taxon | NISP | NISP% | MNI | MNI% | Fragmentation rate |
|---|---|---|---|---|---|
| Lepisosteiformes indet. | 4 | 0.9 | 1 | 5 | 4 |
| Albanerpetontidae indet. | 2 | 0.5 | 1 | 5 | 2 |
| Anura | 1 | 0.2 | 1 | 5 | 1 |
| *Kallokibotion* sp. | 124 | 28.1 | 2 | 10 | 62 |
| Dortokidae indet | 2 | 0.5 | 1 | 5 | 2 |
| Squamata | 3 | 0.7 | 1 | 5 | 3 |
| Crocodyliformes | 3 | 0.7 | | | |
| *Allodaposuchus* sp. | 28 | 6.3 | 1 | 5 | 28 |
| *Theriosuchus*-like sp. | 21 | 4.8 | 1 | 5 | 21 |
| *Acynodon* sp. | 5 | 1.1 | 1 | 5 | 5 |
| *Doratodon* sp. | 11 | 2.5 | 1 | 5 | 11 |
| Ornithopoda indet. | 12 | 2.7 | 0 | 0 | |
| Rhabdodontidae indet. | 63 | 14.3 | 2 | 10 | 32 |
| Hadrosauroidea indet. | 6 | 1.4 | 1 | 5 | 6 |
| Titanosauria indet | 143 | 32.4 | 2 | 10 | 72 |
| *Richardoestesia* sp. | 2 | 0.5 | 1 | 5 | 2 |
| Velociraptorinae, Dromaeosauridae | 1 | 0.2 | 1 | 5 | 1 |
| Theropoda | 4 | 0.9 | 0 | 0 | |
| Pterosauria | 5 | 1.1 | 1 | 5 | 5 |
| Kogaionidae indet. | 1 | 0.2 | 1 | 5 | 1 |
| ***Total*** | ***441*** | ***100*** | ***20*** | ***100*** | ***257*** |

presumably aquatic or semi-aquatic taxa – are only subordinately represented (about 0.5%, according to the NISP) and occur mainly isolated.

The high abundance of titanosaur and rhabdodontid remains in the K2 material is consistent with the pattern reported for the entire Hațeg Basin by [4], where representatives of these two dinosaur groups usually yield the most frequently recovered elements from the different vertebrate sites, while the hadrosauroid and theropod material (not to mention that of ankylosaurs) is far less common (but see also below, Discussions).

### 4.5.2. Minimum number of individuals (MNI).

**Methods:**

During the calculation of MNI, following the methodology of [8,43,151,152], we first determined the minimum number of elements (MNE) for each taxon, and then the greatest MNE value was defined as the MNI value for that taxon.

**Results:**

At least 20 different individuals were identified in the vertebrate assemblage from site K2 (Table 3 and Fig 12C). The taxonomic abundances estimated based on MNI show that dinosaurs are still the most abundant group in the assemblage (about 35% of MNI), seconded by crocodyliforms (20%), while the turtles represent the third most common group (15%) (Fig 12C). All other higher-level taxa (fishes, amphibians, squamates, pterosaurs and mammals) are possibly represented by as few as only one individual each, based on the MNI calculations (Table 3).

**Comments:**

The distribution of MNI-derived taxonomic abundances shows a very similar distribution to the one indicated by NISP, as in both cases dinosaurs, turtles and crocodyliforms are the dominant groups in the assemblage. Unfortunately, however, these MNI values are not suitable for a more detailed characterisation of the former diversity, as all other taxa except rhabdodontid and titanosaur dinosaurs as well as *Kallokibotion* turtles are potentially represented by only one individual.

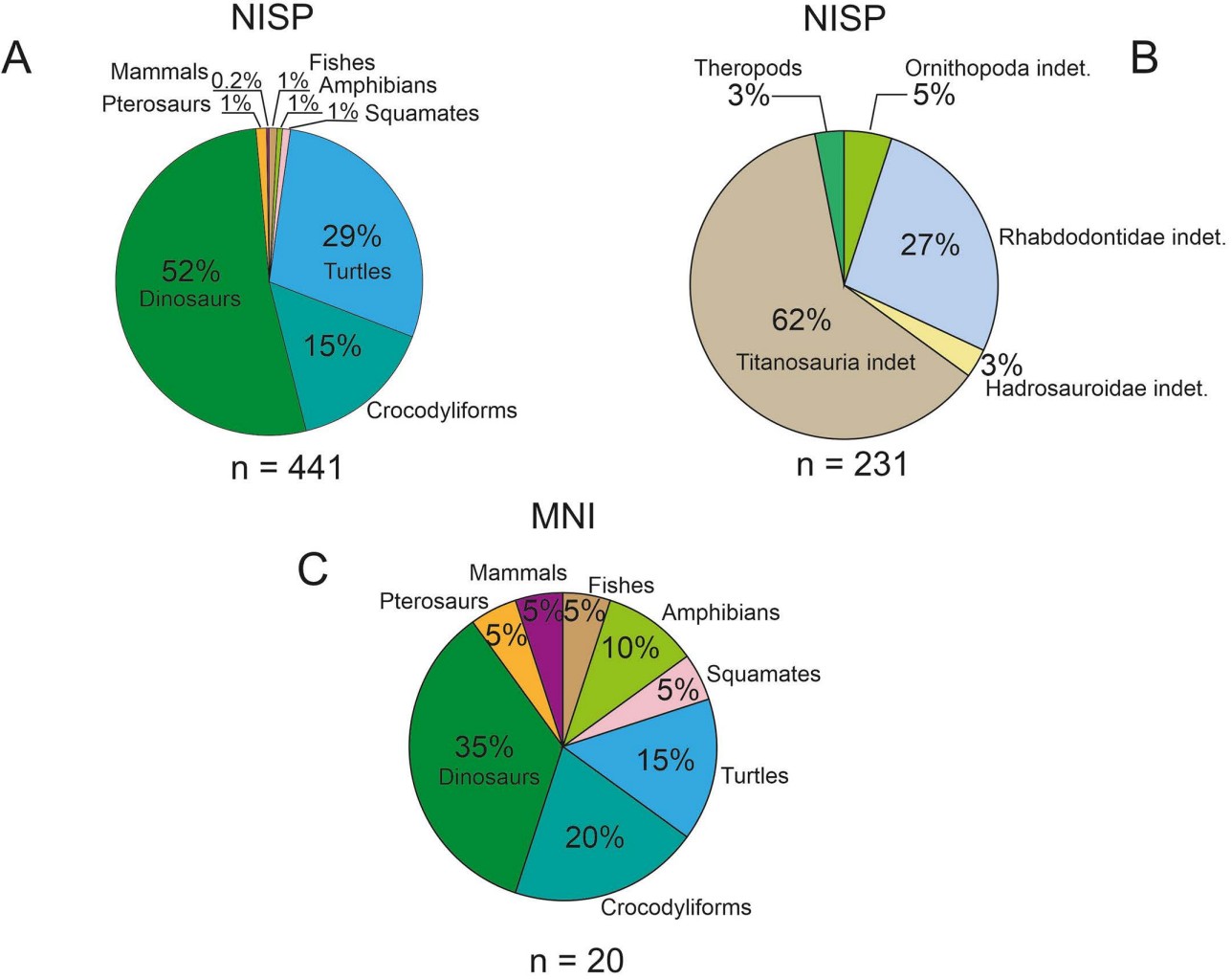

**Fig 12. Relative abundances of the vertebrate taxa in site K2, Vălioara (Densuş–Ciula Formation). (A-B)** NISP and **(C)** MNI distributions.

The very low MNI values may indicate that at least in part site K2 actually represents a random, accidental bone assembly formed during a short time interval, rather than the result of continuous, attritional accumulation where remains of several different individuals may have been gathered over a longer period of time.

### 4.5.3. Skeletal representation and preservation mode.

Most of the vertebrate material from site K2 was collected as isolated remains, although well–preserved, associated (e.g., rhabdodontid and titanosaur material) or, more rarely, articulated (e.g., rhabdodontid cranial bones, titanosaur vertebrae, and *Kallokibotion* shell) skeletal parts were also discovered. The large number of isolated elements in our count is certainly increased by the fact that any specimen was considered isolated as long as there was no definitive evidence for the probability of its association with other vertebrate remains. For example, the largest part of the K2 rhabdodontid material is represented by specimens that display similar preservation states and are of commensurate size, and often represent elements from opposite sides; thus, these specific elements of the rhabdodontid assemblage can be reasonably considered as an associated skeletal material (e.g., [12]). However, based on the MNI calculation at least two rhabdodontid individuals are represented in the K2 assemblage, and therefore it cannot

be excluded that isolated bones from other individuals may be also present besides the clearly associated skeletal material. Based on this reasoning, the rhabdodontid assemblage collected from site K2 cannot be strictly considered as associated material in its entirety without a detailed taphonomic analysis, since elements of this assemblage were found mixed with those of other vertebrates (see below). Accordingly, these remains are conservatively treated in our taphonomic survey as 'isolated elements', although highly likely most of these did originate from a small number of incomplete skeletons.

Stemming from these observations and rationale, the part of the K2 assemblage made up of isolated bones can be divided into three subsets indicating different taphonomic histories. The first subset includes the highly abraded, small-sized, rounded bone pebbles (n = 290; about 30% of the entire available vertebrate material) that are unidentified taxonomically and/or anatomically. These are theoretically hydraulically equivalent with the dominant size of the sediment grains, and thus they could have been transported from the background area for a long time together with the bed-load sediment of ancient streams discharging into the lake, or maybe even underwent limited intraformational reworking. The second subset groups about 53% of the K2 collection, containing most of the identifiable isolated bones (n = 452). These elements have a clearly different taphonomical history from those of the first subset, and represent the most intriguing subject for our taphonomic investigations (see below). The third group contains the isolated teeth (n = 70; about 9% of the entire sample), considered from a taphonomical point of view separately from the remainder of the isolated bone assemblage, because their particular dispersal potential and resistance against abrasion are markedly different [153,154].

The disarticulated but clearly associated skeletal elements are significantly rarer than the isolated bones, and are limited to those of rhabdodontid and titanosaur dinosaurs. These elements were discovered separated from each other and slightly dispersed, but nevertheless they clearly belong to the same skeleton as they were excavated in close spatial proximity, and their dimensions display appropriate proportions for them to belong to a single individual. The clearest evidence for such association of rhabdodontid skeletal material is represented by the example of the ilium LPB (FGGUB) R.2809 found alongside the closely spaced pair of left and right ischia (LPB (FGGUB) R.2810, R.2811; Fig 2G). Furthermore, field observations made during the excavation identified only slightly disturbed anatomical connections between certain rhabdodontid postcranial elements, such as dorsal vertebrae (e.g., LPB (FGGUB) R.2795 and R.2796) and closely spaced rib fragments (e.g., LPB (FGGUB) R.2787 and R.2791) (see also [12]).

Meanwhile, articulated skeletal parts are extremely rare in the K2 assemblage. Except for one articulated and almost complete *Kallokibotion* sp. shell preserving both the plastron and the carapace, all other articulated remains also belong to the rhabdodontid and titanosaur dinosaurs, as in the case of the associated ones. The titanosaur skeletal elements from this category are represented by four caudal vertebrae with their vertebral arches still in articulation, retaining their original anatomical position, and thus this partial skeleton is regarded as articulated (LPB (FGGUB) R.2715; Fig 11D). The similarly preserved rhabdodontid material includes mainly articulated cranial elements found in their original life-time position, such as the right and left premaxilla LPB (FGGUB) R.2769 (Fig 10A) or the frontal-postorbital complex LPB (FGGUB) R.2774 (Fig 10D).

Elements of the axial skeleton are the most abundant remains (representing about 59%) in the K2 assemblage, followed by plate-like turtle shell fragments (27%) and by appendicular skeletal elements (14%) (Fig 13A). All Voorhies groups are represented, with elements belonging to Group I (76%; ribs and vertebrae) being the most abundant, while those of Group II (14%; limb bones) and Group III (10%; skull and mandible elements) are present in roughly the same proportion in the assemblage (Fig 13B).

All Voorhies groups are under-represented compared to their expected frequencies, but the proportions of this bias are different (Table 4). The limb bones and the vertebrae have the same recovery rate (representing about 12%), while the recovered element percentage of skull and mandible elements is only about 4% in the K2 assemblages (Table 4). However, this distribution is notably different in the case of the rhabdodontid material, where the limb bones as well as the skull

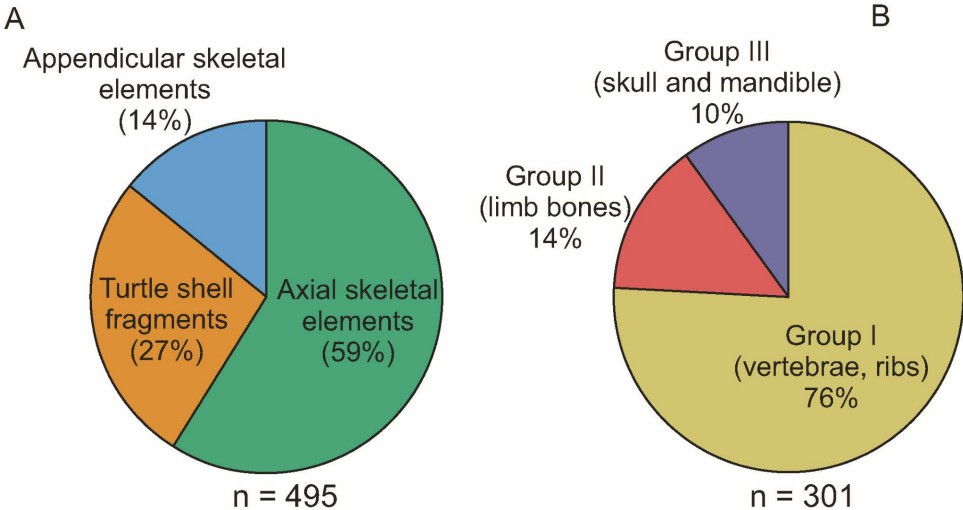

**Fig 13. Relative abundances of the skeletal elements in the K2 vertebrate assemblage. (A)** Different skeletal parts and **(B)** Voorhies Groups.

**Table 4. Recovery rate of the different skeletal elements in the K2 assemblage.**

| | | Testudinata | Crocodyliformes | Rhabdodontidae | Hadrosauroidea | Titanosauria | Theropoda | Pterosauria | Total |
|---|---|---|---|---|---|---|---|---|---|
| MNI | | 3 | 4 | 2 | 1 | 2 | 1 | 1 | 14 |
| Expected number of elements | Vertebrae | 126 | 288 | 156 | 87 | 164 | 73 | 15 | 909 |
| | Limb bones | 36 | 48 | 24 | 12 | 24 | 12 | 12 | 168 |
| | Skull elements | 93 | 124 | 62 | 31 | 62 | 31 | 31 | 434 |
| Actual number of element | Vertebrae | 0 | 2 | 13 | 0 | 97 | 0 | 0 | 112 |
| | Limb bones | 0 | 3 | 7 | 0 | 6 | 3 | 1 | 20 |
| | Skull elements | 0 | 3 | 12 | 1 | 3 | 0 | 0 | 19 |
| **Recovered element percentage of vertebrae** | | | 1 | 8 | 0 | 59 | 0 | 0 | 12 |
| **Recovered element percentage of limb bones** | | | 6 | 29 | 0 | 25 | 25 | 8 | 12 |
| **Recovered element percentage of skull and mandible elements** | | | 2 | 19 | 3 | 5 | 0 | 0 | 4 |

and mandible elements have much higher recovery rates (29% and 19% respectively; Table 4) than the vertebrae (8%), indicating a slightly different taphonomical history compared to the rest of the local assemblage (see below).

### 4.5.4. Skeletal completeness.

**Methods:** Calculation of total skeletal completeness (%TC) was conducted using the following equation [8,43,151,155,156]:

$$\%TC = \left(\sum At \times 100\right) / \left(\sum Et \times MNI\right), \text{ where}$$

%TCt: percentage of total skeletal completeness of taxon t;

∑ At: the actual number of skeletal elements from K2 referred to taxon t;

∑ Et: the expected number of elements in a complete skeleton of taxon t;

MNI: the minimum number of individuals.

These calculations concern only seven different groups (turtles, crocodyliforms, dinosaurs and pterosaurs), whereas the fishes, amphibians, squamates and mammals were excluded from this study.

*Results* The total skeletal completeness of the investigated seven groups range from 0.5% to 12.7%, with the rhabdodontids (11.5%) and titanosaurs (12.7%) showing the highest recorded values (Table 5).

*Comments*: The significant differences in calculated skeletal completeness data suggest that the sorting effect was not uniform for the taxa recovered at site K2. The relatively high skeletal completeness of the rhabdodontid (11.5%) and titanosaur (12.7%) bone assemblages suggests that large proportions of more or less complete skeletons of these animals were deposited together in the K2 thanatocoenosis, incidentally further supporting higher probabilities of association for the isolated elements of these dinosaurs than those for other groups. At least two possible scenarios can account for the observed skeletal completeness pattern. On the one hand, different values of relative skeletal completeness were suggested to distinguish between taxa that were members of a paleo-community living in the proximity to the site of deposition (and therefore are expected to have died often there) and taxa living in more distal communities [151,155]. According to this approach, rhabdodontid and titanosaur dinosaurs would represent autochthonous taxa that populated a habitat close to the lacustrine setting represented at site K2, whereas the skeletal material of other taxa identified at the site were transported here by fluvial action from more distal areas, concomitantly with a more advanced dispersion of the skeletons before final burial. However, an alternative possibility (albeit not necessarily excluding the first one) is that the skeletal material of the rhabdodontid and titanosaur dinosaurs was transported by currents into the site as floating carcasses, with soft tissue and ligaments still holding the body parts together, and subsequently, following the settling of their elements these were mixed with attritionally gathered remains of the other taxa represented in the assemblage (see below; Discussions).

### 4.5.5. Size and volume distributions.

**Methods:** The volume of skeletal elements was calculated from the measurements along their three axes. These were then classified in the following categories (following [42]): very small (<1 cm$^3$); small (1–16 cm$^3$); medium (16–126 cm$^3$); large (126–1000 cm$^3$); and very large (>1000 cm$^3$). Furthermore, the bones and bone fragments from the sample were also divided into four size categories (very small: <5 cm; small: 5–10 cm; medium: 10–50 cm; and large: 50–100 cm) according to their maximum dimension. We also estimated a pre-breakage maximum length for the incomplete elements and grouped these, too, into the same size categories as the complete ones.

*Results*: Maximum bone dimension in the K2 assemblage varies between 0.6 cm and 50.1 cm. About 45% of the sample, however, is represented by elements smaller than 1 cm$^3$, and almost 50% is smaller than 5 cm in maximum dimension (Fig 14A and B). Half of the elements were relatively small (between 5–10 cm long) even in their original, pre-breakage state (Fig 14C). Except for the rhabdodontid and titanosaur material, fossils of all other taxonomic groups are dominated by remains smaller than 1 cm$^3$ (Fig 14A).

**Comments:**  The maximum size of both completely preserved and of reconstructed skeletal elements is predominantly less than 10 cm, suggesting that the dominance of small-sized vertebrate remains in the K2 assemblage is not solely the result of fragmentation and destruction of (previously) larger bones; instead, it shows that even initially relatively small skeletal elements (<10 cm) were buried in greater numbers than larger ones.

Predominance of relatively large-sized elements (16–1000 cm$^3$) is observable in the case of the turtle, rhabdodontid and titanosaur material, while those smaller than 1 cm$^3$ are the dominant elements for the other taxa (Fig 14A). These distributional differences in calculated bone volumes further confirm that the K2 sub-samples belonging to the rhabdodontid and titanosaur dinosaurs experienced different taphonomical histories than those of other vertebrate groups represented in the K2 assemblage. Indeed, fossils of crocodyliforms, small theropods and even hadrosauroids consist of small-sized elements that could be easily transported by even a low-energy current. This, however, is definitively not the case of the large bones present in the rhabdodontid and titanosaur samples that could not to have been transported as isolated elements by fluvial traction, as they are clearly not in hydraulic equilibrium with the low-energy current that deposited the relatively

**Table 5. Skeletal completeness in the K2 assemblage: %TC – represents the percentage of total skeletal completeness; A – the actual number of skeletal elements recorded from site K2; E– the expected number of elements in a complete skeleton; %R – recovered element percentage of a certain skeletal element.**

| | Testudinata | | | Crocodyliformes | | | Rhabdodontidae | | | Titanosauria | | | Hadrosauroidea | | | Theropoda | | | Pterosauria | | |
|---|---|---|---|---|---|---|---|---|---|---|---|---|---|---|---|---|---|---|---|---|---|
| MNI | 3 | | | 4 | | | 2 | | | 2 | | | 1 | | | 2 | | | 1 | | |
| Elements | A | E | %R | A | E | %R | A | E | %R | A | E | %R | A | E | %R | A | E | %R | A | E | %R |
| premaxilla | 0 | 2 | 0 | 0 | 2 | 0 | 3 | 2 | 75 | 0 | 2 | 0 | 0 | 2 | 0 | 0 | 2 | 0 | 0 | 2 | 0 |
| maxilla | 0 | 2 | 0 | 0 | 2 | 0 | 1 | 2 | 25 | 0 | 2 | 0 | 0 | 2 | 0 | 0 | 2 | 0 | 0 | 2 | 0 |
| nasal | 0 | 2 | 0 | 0 | 2 | 0 | 0 | 2 | 0 | 0 | 2 | 0 | 0 | 2 | 0 | 0 | 2 | 0 | 0 | 2 | 0 |
| lacrimal | 0 | 0 | 0 | 0 | 2 | 0 | 1 | 2 | 25 | 0 | 2 | 0 | 0 | 2 | 0 | 0 | 2 | 0 | 0 | 2 | 0 |
| prefrontal | 0 | 2 | 0 | 1 | 2 | 13 | 1 | 2 | 25 | 0 | 2 | 0 | 0 | 2 | 0 | 0 | 2 | 0 | 0 | 2 | 0 |
| frontal | 0 | 2 | 0 | 1 | 2 | 13 | 2 | 2 | 50 | 0 | 2 | 0 | 0 | 2 | 0 | 0 | 2 | 0 | 0 | 2 | 0 |
| parietal | 0 | 2 | 0 | 1 | 2 | 13 | 0 | 2 | 0 | 0 | 2 | 0 | 0 | 2 | 0 | 0 | 2 | 0 | 0 | 2 | 0 |
| postorbital | 0 | 2 | 0 | 0 | 2 | 0 | 2 | 2 | 50 | 0 | 2 | 0 | 0 | 2 | 0 | 0 | 2 | 0 | 0 | 2 | 0 |
| jugal | 0 | 2 | 0 | 0 | 2 | 0 | 0 | 2 | 0 | 0 | 2 | 0 | 0 | 2 | 0 | 0 | 2 | 0 | 0 | 2 | 0 |
| quadrate | 0 | 2 | 0 | 0 | 2 | 0 | 0 | 2 | 0 | 0 | 2 | 0 | 0 | 2 | 0 | 0 | 2 | 0 | 0 | 2 | 0 |
| squamosal | 0 | 2 | 0 | 0 | 2 | 0 | 0 | 2 | 0 | 0 | 2 | 0 | 0 | 2 | 0 | 0 | 2 | 0 | 0 | 2 | 0 |
| predentary | 0 | 1 | 0 | 0 | 1 | 0 | 0 | 1 | 0 | 0 | 1 | 0 | 0 | 1 | 0 | 0 | 1 | 0 | 0 | 1 | 0 |
| dentary | 0 | 2 | 0 | 0 | 2 | 0 | 3 | 2 | 75 | 0 | 2 | 0 | 1 | 2 | 50 | 0 | 2 | 0 | 0 | 2 | 0 |
| surangular | 0 | 2 | 0 | 0 | 2 | 0 | 0 | 2 | 0 | 0 | 2 | 0 | 0 | 2 | 0 | 0 | 2 | 0 | 0 | 2 | 0 |
| angular | 0 | 2 | 0 | 0 | 2 | 0 | 0 | 2 | 0 | 0 | 2 | 0 | 0 | 2 | 0 | 0 | 2 | 0 | 0 | 2 | 0 |
| exoccipital | 0 | 2 | 0 | 0 | 2 | 0 | 0 | 2 | 0 | 0 | 2 | 0 | 0 | 2 | 0 | 0 | 2 | 0 | 0 | 2 | 0 |
| Vertebrae | | | | | | | | | | | | | | | | | | | | | |
| cervical | 0 | 7 | 0 | | | | 1 | 11 | 5 | 0 | 12 | 0 | 0 | 13 | 0 | 0 | 10 | 0 | 0 | 7 | 0 |
| dorsal | 0 | 10 | 0 | | | | 4 | 17 | 12 | 3 | 12 | 13 | 0 | 16 | 0 | 0 | 13 | 0 | 0 | 7 | 0 |
| caudal | 0 | 24 | 0 | 2 | 72 | 1 | 5 | 44 | 6 | 36 | 53 | 34 | 0 | 49 | 0 | 0 | 49 | 0 | 0 | | |
| sacrum | 0 | 1 | 0 | | | | 0 | 6 | 0 | 1 | 5 | 10 | 0 | 9 | 0 | 0 | 1 | 0 | 0 | 1 | 0 |
| Girdle and limb bones | | | | | | | | | | | | | | | | | | | | | |
| scapula | 0 | 2 | 0 | 0 | 2 | 0 | 1 | 2 | 25 | 2 | 2 | 50 | 0 | 2 | 0 | 0 | 2 | 0 | 0 | 2 | 0 |
| coracoid | 0 | 2 | 0 | 0 | 2 | 0 | 0 | 2 | 0 | 0 | 2 | 0 | 0 | 2 | 0 | 0 | 2 | 0 | 0 | 2 | 0 |
| humerus | 0 | 2 | 0 | 1 | 2 | 13 | 0 | 2 | 0 | 2 | 2 | 50 | 0 | 2 | 0 | 0 | 2 | 0 | 0 | 2 | 0 |
| ulna | 0 | 2 | 0 | 0 | 2 | 0 | 1 | 2 | 25 | 0 | 2 | 0 | 0 | 2 | 0 | 0 | 2 | 0 | 0 | 2 | 0 |
| radius | 0 | 2 | 0 | 0 | 2 | 0 | 1 | 2 | 25 | 0 | 2 | 0 | 0 | 2 | 0 | 0 | 2 | 0 | 0 | 2 | 0 |
| metacarpal or metatarsal | 0 | 20 | 0 | 0 | 20 | 0 | 4 | 14 | 14 | 1 | 20 | 3 | 0 | 8 | 0 | 0 | 16 | 0 | 1 | 16 | 6 |
| phalanges | 0 | 56 | 0 | 0 | 52 | 0 | 1 | 22 | 2 | 0 | 36 | 0 | 0 | 48 | 0 | 0 | 36 | 0 | 0 | 8 | 0 |
| ilium | 0 | 2 | 0 | 0 | 2 | 0 | 1 | 2 | 25 | 0 | 2 | 0 | 0 | 2 | 0 | 0 | 2 | 0 | 0 | 2 | 0 |
| ischium | 0 | 2 | 0 | 0 | 2 | 0 | 2 | 2 | 50 | 0 | 2 | 0 | 0 | 2 | 0 | 0 | 2 | 0 | 0 | 2 | 0 |
| femur | 0 | 2 | 0 | 2 | 2 | 25 | 0 | 2 | 0 | 2 | 2 | 50 | 0 | 2 | 0 | 1 | 2 | 25 | 1 | 2 | 50 |
| tibia | 0 | 2 | 0 | 0 | 2 | 0 | 3 | 2 | 75 | 0 | 2 | 0 | 0 | 2 | 0 | 1 | 2 | 25 | 0 | 2 | 0 |
| fibula | 0 | 2 | 0 | 0 | 2 | 0 | 1 | 2 | 25 | 1 | 2 | 25 | 0 | 2 | 0 | 0 | 2 | 0 | 0 | 2 | 0 |
| Complete shell (only for turtles) | 1 | 1 | 33 | | | | | | | | | | | | | | | | | | |
| Plate fragments (only for turtles) | 58 | 58 | 33 | | | | | | | | | | | | | | | | | | |
| ∑A | 59 | | | 8 | | | 38 | | | 48 | | | 1 | | | 2 | | | 2 | | |
| ∑ E | | 226 | | | 195 | | | 165 | | | 189 | | | 194 | | | 176 | | | 90 | |
| %TC | | | 8,7 | | | 1,0 | | | 11,5 | | | 12,7 | | | 0,5 | | | 0,6 | | | 2,2 |

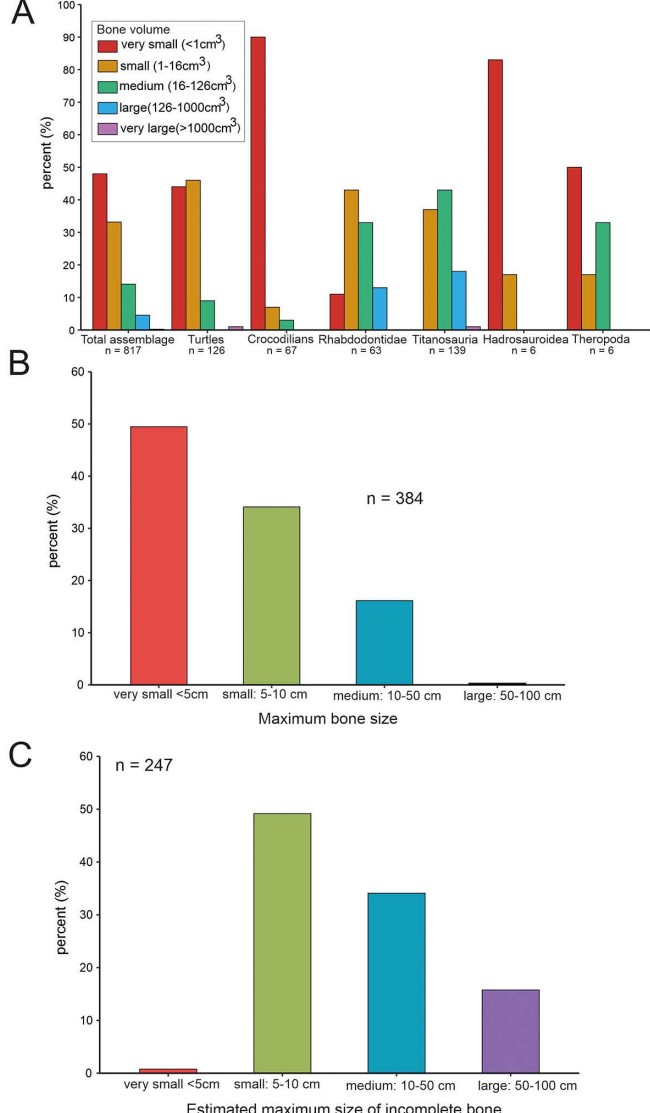

**Fig 14. Bone representation by size in the K2 assemblage.** (A) Size distribution of bones based on (A) bone volume (B) maximum dimension bones and bone fragments and (C) complete and reconstructed skeletal elements.

fine-grained sediments of the K2 bonebed. Lack of hydraulic equivalence between these large skeletal elements and the encasing sediment makes it rather more likely that, unlike the large number of other small vertebrate remains, the fossils belonging to individuals of these two dinosaur groups were transported to, and accumulated within the site as floating carcasses.

### 4.5.6. Abrasion and weathering.

**Methods:** Abrasion occurs through the interaction between sediment particles and bones, resulting in the grinding and polishing of the edges and/or surfaces of vertebrate fossils [41,157,158]. We examined the K2 material for these types of modifications and divided the fossils into two main categories: 1) unabraded bone, and 2) abraded bone. Weathering,

reflective of the period of surface exposure of bones before their burial [159], was also observed in the K2 assemblage that, again, can be divided into two main categories: 1) unweathered bones, and 2) weathered ones.

*Results*: The largest part of the assemblage is unabraded, with only 37% of the investigated bone material showing any evidence of abrasion (Fig 15A). Furthermore, only 21% of taxonomically identified elements (NISP) were abraded (Fig 15B), indicating that a disproportionately large part of the abraded bones in the K2 assemblage belongs to the category of unidentified bone pebbles. The skeletal material referrable to rhabdodontids shows the lowest overall abrasion rate (only 13% of this material is abraded to some extent; Fig 15B). Finally, the vast majority of the bones present no evidence of weathering, as only as few as 6% of the specimens display some degree of flaking associated with cracks due to weathering (Fig 15C).

**Comments:** The generally low abrasion rate detected in the K2 assemblage indicates that the vast majority of the fossils were not exposed to prolonged interaction with transported sediment particles, suggesting that piece-by-piece fluvial transport of the bones into the site was not substantial. The extreme low rate of weathering further shows that most of the bones in the K2 assemblage were buried relatively rapidly after death (probably within 1–2 years; [159]), and even before that were not exposed to the action of significant destructive agents and processes on the surface or within the soil zone. Altogether, the low rates of both abrasion (21%) and weathering (6%) in the taxonomically identified vertebrate assemblage (based on NISP) suggest that the death assemblage at site K2 was probably buried within/very close to its

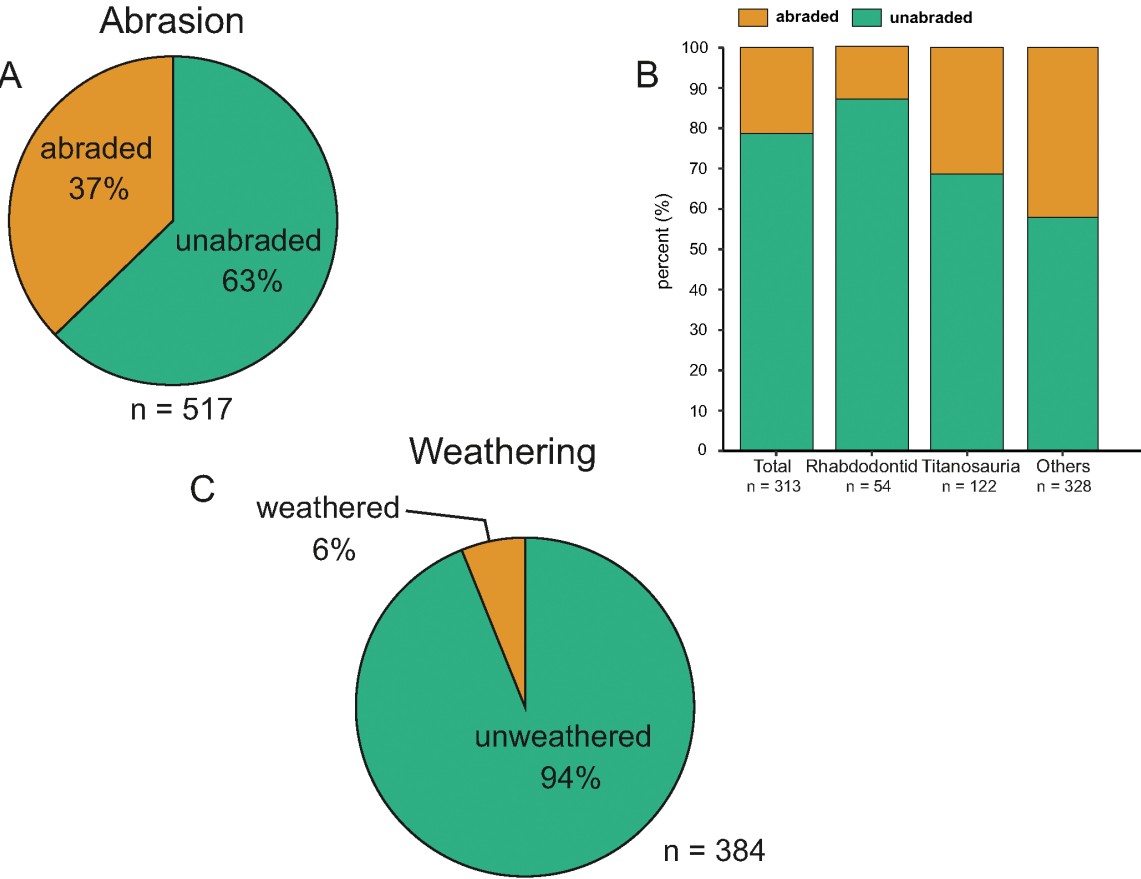

**Fig 15. Bone surface modifications recorded.** Distribution of abrasion based on total (A) and NISP assemblage **(B)**. Distribution of weathered bones in the site K2 assemblage based on total assemblage.

living environment, with a very limited post-mortem transportation and taphonomic processing, and thus it represents a largely (par)autochthonous assemblage (e.g., [41,160]).

### 4.5.7. Fracture patterns of skeletal elements.

**Methods:** Breakage pattern provides information about whether the observed bone was "fresh" (i.e., collagen, water and organic matter was still present in the bone structure) or already "fossilized" (meaning bone already devoid of collagen and water components) when it got broken. Several breakage types were distinguished within the K2 fossil material, of which oblique (2), longitudinal (3) and spiral fractures (4) are typical "pre-fossilization" breakage patterns, whereas smooth transversal breakage type (5) occurs during the fossildiagenetic phase [42,161,162].

**Results:** Remarkably, 88% of the K2 bone material is broken (Fig 16A), and this ratio remains rather high (74%) even within the assemblage of taxonomically identifiable elements (NISP) (Fig 16B). While 32% of the surveyed specimens show one or another of the "pre-fossilization" fracture types, about 40% of fractures have indeterminate origin; this latter group contains elements that suffered/underwent breakage during excavation, and for which the original breakage state

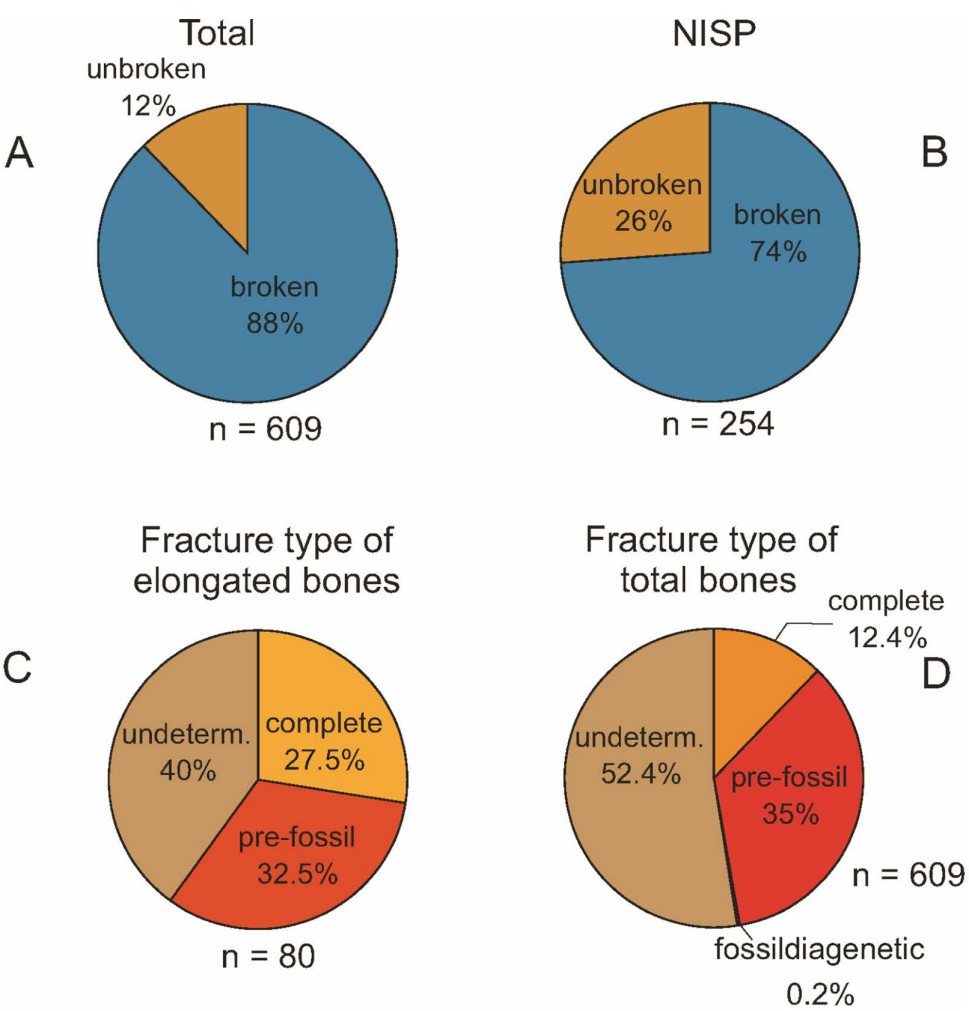

**Fig 16. Fracture patterns of skeletal elements.** Bone breakage distribution based on (A) total and **(B)** NISP assemblage. Distribution of breakage types (C) in the total and (D) the NISP assemblage.

could no longer be reconstructed reliably. Interestingly, no examples displaying a fracture type that originated in the fossildiagenetic phase was documented in the assemblage.

**Comments:** We note that the high fragmentation rate observed in the K2 assemblage was probably created mainly by, and during, the excavation procedure, as most of the recorded fracture types were of indeterminate origin. The complete absence of the smooth transversal breakage type in the K2 material suggests that 1) reworking of already fossilized bones from some earlier accumulations, if present at all, was definitively not extensive; and 2) post-burial disturbance at the site was not significant.

## 5. Discussion

In parallel with the excavations at site K2, a detailed geological mapping was conducted around Vălioara Valley over an area covering almost 10 km$^2$, in order to provide a more accurate picture of the geological, stratigraphical and paleoenvironmental context of the different vertebrate sites from this part of Haţeg Basin [20]. This investigation has shown that the important environmental shift from marine to continental deposition in Haţeg Basin occurred significantly earlier (by the middle late Campanian) than hitherto considered, and that the lower Densuş-Ciula Formation – previously thought to be restricted to the Maastrichtian – covers a good part of the upper Campanian as well. Most significantly in this respect, several recently acquired zircon U-Pb radiometric ages derived from primary andesitic-dacitic clasts coming from volcaniclastic deposits that belong to the Lower Member of the Densuş-Ciula Formation [35] as well as from detritic zircon samples that originate from tuffitic intercalations in the basalmost part of the continental Densuş-Ciula succession [20] fall into the latest early to earliest late Campanian time interval (roughly 81–78 Ma), in good accordance with different marine micropaleontological and palynological biostratigraphic constraints currently available from both the continental and the underlying marine beds [11,20,32,52,59]. These new age constraints regarding the starting point of terrestrial sedimentation in Haţeg Basin also shifted the depositional datum of several vertebrate sites from the Middle Member of the Densuş-Ciula Formation towards older ages than previously assumed [20]. According to the dataset presented in this former study, and convergently also supported by the palynological investigations reported here, an earliest early Maastrichtian age (close to Campanian/Maastrichtian boundary) was determined for the K2 bonebed (Fig 3A), and thus this site yields the remains of the oldest well-diversified latest Cretaceous vertebrate assemblage known from the entire Haţeg Basin [20].

### 5.1. Depositional environment of site K2

The studied lacustrine sequence at site K2 is underlain by red, poorly sorted floodplain muds (Fig 3B). As fluvial channel deposits (e.g., gravel and sands) are lacking at the base of the succession, the lake is unlikely to represent an oxbow lake formed due to channel abandonment. Rather it was most likely formed either by water being trapped in a local depression on the floodplain during/after a flooding event or by the subsidence of the low-lying area of the floodplain below the groundwater table, or else by the combination of these two factors (e.g., [44,45]) (Fig 17; stage 1). This scenario is supported by the identification of a relatively raise water table throughout the larger study area at this stage of basin evolution as suggested by the presence of grey floodplain mudstones, contrasting with the dominant red colour of the under- and overlying fine-grained sediments, as well as by the dominance of fluvial deposits and the lack of proximal alluvial fan gravels [20]. Elevated groundwater levels are probably linked to a minor sea-level increase at that time [20] compared to other periods of latest Cretaceous basin evolution as well as to overall higher subsidence rates within the basin.

   In addition to the sedimentological characteristics of the site, the palynological investigations also point towards a fluvio-lacustrine depositional environment, as the local palynomorph assemblage is dominated by hygrophytic fern spores such as *Polypodiaceoisporites*, *Deltoidospora* and *Laevigatosporites,* derived from plant communities that usually grow along river banks or around lakes [163], under rather warm climatic conditions [56]. The dominance of these fern spores, combined with the presence of *Proteacidites* and *Inaperturopollenites* (*Taxodium*) pollen, and that of rare phytoplankton specimens referable to freshwater algae, is suggestive of perennially to temporarily wet, low-elevation fluvio-deltaic

systems such as wetlands, ponds, swamp or temporary lakes [37,57,58]. The presence of freshwater gastropods and bivalves, suggesting steady or slowly running water, and that of the terrestrial snails indicative of humid forests further supports these sedimentological and palynological results.

The three coarser, silty-sandy horizons identified in the K2 local succession represent three distinct flooding events that were capable of transporting coarse material along with occasional fossils into the lake, potentially in the form of discharging ephemeral but high-energy fluvial channels (Fig 17; stage 2). It is already documented that sedimentation in the basin took place under a subhumid, subtropical climate characterized by strong seasonality [164], a climate that favours the occurrence of high-intensity rainfall events leading to episodes of strongly seasonal discharge and flooding events characterised by rapid rise, sharp and peak [48,165,166]. From this it also follows that the lake itself, despite being a transient feature of the floodplain (as documented by the spatially and temporally restricted sedimentary succession it produced), persisted for a long enough time interval to overlap with at least three distinct episodes of major river floods, most probably representing three distinct seasons with exceedingly high precipitation rates. Indeed, the existence of important time gaps between these three main flooding events is clearly supported by the presence of the 1 to 1.5 m thick lacustrine mud intervals between the coarser flood beds. These interbeds show that finer-grained material could have been introduced into the lake more continuously by smaller-magnitude flooding events that were, however, not capable of transporting coarser sediments. In this respect, it is worth noting that thicker deltaic bodies (e.g., [44,45,167]) are missing from the K2 outcrop, although it is still possible that site K2 represents a more distal part of the lake, beyond the area of delta front deposits. Furthermore, the strong climate seasonality [164] is largely incompatible with the formation of sustained deltas (e.g., [168]), and the alluvial facies of the basin fill in the Vălioara area also indicates the activity of mainly ephemeral streams [20] – altogether weakening support for the development of sustained deltas in the study area. Finally, the absence of desiccation and pedogenic features, of root traces, as well as the general composition of the palynomorph assemblage show that the depositional area of site K2 remained submerged during the lifetime of the lake, in spite of the assessed presence of drier seasons with limited rainfall [164] – most likely as the result of generally high groundwater levels, accompanied by high subsidence rates and/or repeated water supply during floods, throughout the time of lacustrine deposition.

Given the spatial limitations of the outcrop along with the absence of lake margin deposits (e.g., coal seams, root traces [167]) within the K2 local succession, the lateral extent of the lake cannot be precisely assessed. The nearest outcrops of the hosting depositional unit (the grey 'Vălioara' beds), located ca. 350 m to the SE of site K2, only reveal fluvial channel and floodplain deposits, while the supposed basin bounding fault is located about 2 km to the N from the site. These two distances circumscribe the maximal areal extent of the lake, which means that it was a local feature and did not cover the entire basin, and show that the lacustrine deposits terminate somewhere between the K2 and NVS sites. In the latter outcropping succession, along the Ogradiilor Valley (see [11]), there is no sign of any major erosional surface that could have potentially removed the once-existing lacustrine deposits, indicating that the lacustrine bodypinched out before reaching this area.

The local sedimentary sequence along the Neagului Valley represented by the lacustrine deposits that host site K2 is sealed by clean brown mudstone, and then overlain by red, poorly sorted floodplain mudstones (Fig 3B). Their occurrence indicates that the lake eventually dried up due to a possible drop in groundwater level (Fig 17; stage 3), leading to the oxidation of its uppermost clay bed, and finally to the return to oxidative, better-drained floodplain conditions.

## 5.2. Discussion and interpretation of the taphonomic features recorded at site K2

Based on the bonebed classification scheme of [169], the K2 accumulation can be characterized as a high-diversity multitaxic bonebed, because it has yielded remains of 17 vertebrate taxa, which represent ~40% of the known formational diversity [3,8]. The bonebed is multidominant in terms of relative taxonomic abundance, because the NISP assemblage is dominated by three different taxa, with none of these exceeding 50% (Titanosauria: 32%; *Kallokibotion*: 28%; Rhabdodontidae: 14%; Table 3). Based on the dimensions of the recovered elements, the K2 material represents a mixed

(macrofossil and microfossil) assemblage because the percentage contributions of bones larger (50.5%) and smaller (49.5%) than 5 cm both exceed 25% of the NISP (see [169]).

Altogether, 872 vertebrate remains were collected from an approximately 4.75 m² area of the bonebed horizon at site K2 (S2 Table), indicating an extremely high bone density (~183 remains/m²) compared to other vertebrate sites of Haţeg Basin [3,8]. Nonetheless, only about half of the collected specimens (n = 441) can be taxonomically identified as representing at least 17 different taxa, emphasizing that the unidentified bone fragments form an important part of the investigated assemblage. The abrasion rate is predominantly low in the assemblage (only 21% of the NISP were abraded; Fig 15A-B), suggesting that fluvial transport of that particular subsample of the K2 bone material was not significant. Furthermore, the reduced overall weathering state of the fossil material (only 6% of the NISP were weathered; Fig 15C) indicates that the largest part of the identifiable bone assemblage was buried rather rapidly and without significant subaerial exposure after death. Relatively high values of skeletal completeness were only detected in the case of the fossil material referable to rhabdodontid and titanosaur dinosaurs, while this value barely reaches 1% for the other vertebrate groups represented (Table 5). This feature, together with the important proportions of associated and articulated material noted for the same sets of dinosaur remains, but not for the other taxa represented, indicate that more than one taphonomic pathway contributed to the genesis of the K2 taphocoenosis (see below).

The detailed sedimentological investigation carried out at site K2 reveals that the depositional environment of the bonebed was a small lacustrine delta, where ephemeral fluvial channels terminated in the lake, and where the fluvially transported bones and/or carcasses could be preferentially gathered due to a sudden drop in transport energy. This type of sedimentary environment is favourable for the formation of significant bone accumulations (e.g., [170–173]), as the two most important factors for bonebed genesis (i.e., sudden energy depletion and rapid burial of the remains) are clearly present and active. Another significant feature that characterizes such environments is that vertebrate remains with different taphonomical histories can be collected within the same bonebed, due to the co-occurring action of different bone accumulation processes [170,172]; such taphonomically distinct sets of remains can thus include: 1) autochthonous faunal elements from the lacustrine environment itself; 2) parautochthonous faunal elements from within and around the channel system feeding the lacustrine delta and/or from the emergent area surrounding the lake; and 3) allochthonous (channel lag) faunal elements transported from more distal environments by the fluvial system draining into the lake. In the case of site K2, all of these taphonomic sets (and, accordingly, acting factors) can be detected, since the taphonomic analysis of the recovered fossil material has allowed identification of several different modes of bone accumulation. As discussed, three different sub-samples can be separated within the K2 vertebrate assemblage, each of which being characterized by a distinct taphonomical history: 1) unidentified bone pebbles; 2) identifiable isolated bones and teeth; and 3) associated and articulated elements.

The first taphonomical sub-set contains the small-sized, highly abraded, rounded bone fragments (n = 290; about 30% of the entire material), whose poor preservation does not allow their anatomical and/or taxonomic identification. Such type of material is generally common and often dominant in vertebrate fossil sites where fluvial transport has been involved in the accumulation of the remains (e.g., [4,8,156,172,174,175]). They could have been transported (and most of them, for a long time, given their advanced degree of abrasion) alongside the bed-load within fluvial channels from a background area as their size is theoretically hydraulically equivalent with the dominant size of the sediment grains. The relatively high proportion of this category in the K2 material suggests that bone supply from more distant environments, mainly in the form of isolated and abraded fragments, may have been significant at least during high-energy flood events. This taphonomic set separated at site K2 represents a markedly attritional allochthonous channel lag assemblage that got finally introduced into, and settled within the hydrodynamically quieter lake margin environment at the place of fluvial discharge; it is characterized both by very extensive spatial, as well as high time-averaging ($10^2$ to $10^5$ yrs; based on [175]).

The separation of isolated and associated skeletal parts represented in the K2 assemblage is a more difficult endeavour than that of the first, above-discussed taphonomical sub-set. In distinguishing between isolated and associated skeletal parts we follow the interpretation of [176] according to which as long as there is no unequivocal evidence for the

probability of association among vertebrate remains, all elements should be regarded as separated and isolated; using this very restrictive, conservative approach, with the exception of a few clearly associated or articulated elements, the entire identifiable fossil material from K2 should be categorised as isolated remains. However, the taphonomical investigation of the K2 assemblage has pointed out that the probability of association between skeletal elements is higher in the case of the rhabdodontid and titanosaur material than for all the other taxa detected at this site.

Indeed, among the over 800 vertebrate specimens recovered from site K2, only a few cranial elements of rhabdodontids (LPB (FGGUB) R.2769, R.2774), four titanosaur caudal vertebrae (LPB (FGGUB) R.2715), and one *Kallokibotion* shell (LPB [FGGUB] R.2710) were found properly articulated. Nevertheless, somewhat more common are the associated skeletal remains, all of these belonging exclusively to rhabdodontids and titanosaurs (e.g., LPB (FGGUB) R.2795, R.2796, R.2787, R.2791, R.2809, R.2810, R.2811). Assessing the validity of these associations is based on several concurring observations such as the spatial proximity of the specimens within the site, the relative sizes of the individual elements, as well as their good fit together along the joint surfaces. Furthermore, the quasi-exclusive presence of articulated skeletal parts among the rhabdodontid, respectively titanosaur material indicates that the probability of skeletal element association is most likely higher in the case of these two taxa, compared to the other ones found at site K2. Finally, we emphasize that the relatively high values of skeletal completeness recorded for the rhabdodontid (11.5%), and titanosaur (12.7%) bone assemblages also suggests that the probability of association between elements belonging to these groups, but which were otherwise found isolated within the bonebed, must be higher than for the other taxa represented at K2 (skeletal completeness 0.3–1%; see Table 5).

We also note that all of the rhabdodontid and titanosaur remains share the same colour as well as the same main preservational features, indicating a largely similar taphonomic history. Rhabdodontid skeletal elements originating from different regions of the body are largely commensurate in size, an observation that also suggests that these belong to skeleton-level associations (see [12]). Furthermore, it is worth emphasizing that except for the rhabdodontid and titanosaur skeletal material, all other vertebrate groups identified at K2 are characterized by elements that are almost always smaller than $1\,cm^3$, while relatively large bones ($1$–$1000\,cm^3$) form a major size category in the case of the two above-mentioned dinosaur groups (Fig 14A). Small-sized vertebrate remains could have accumulated piece by piece within the site in a variety of ways (e.g., long-distance transport through fluvial activity during flood periods, *in situ* attritional death, transportation into the site by surface water run-off from the contemporaneous ground surface), whereas similar mechanisms would not function in the accumulation of the larger bones as isolated remains. It is far more likely that these latter elements were transported into, and accumulated in, the site while the body parts they belonged to were still held together by soft tissue and ligaments (see below).

Last but not least, all of the rhabdodontid and titanosaur fossils were collected from a localized area of about 4.75 m², where the different disarticulated elements were found mixed together and piled on top of each other over this very small surface. Although elements representing the other groups identified at K2 were also collected from the same general area together with the rhabdodontid and titanosaur material, these were rarer, always isolated, and highly scattered across the bonebed, and never displayed any degree of anatomical connection with each other. Based on the above-mentioned taphonomical features, the disarticulated material collected at site K2 can be further subdivided into two different sets characterized by clearly different taphonomical histories.

1. Unassociated skeletal remains that include small-sized (mostly less than 5 cm in maximum dimension) bones and teeth from different aquatic, semiaquatic and terrestrial animals, and which were probably added to the K2 bone assemblage as individual organo-sedimentary 'clasts' collected and transported from different places and environments, thus representing an attritional material.

2. The presence of articulated and associated elements, shared sets of taphonomic modifications, as well as their relatively high skeletal completeness indices, indicate that the rhabdodontid and the titanosaur material recovered from site K2 can be considered as representing remains of the associated skeletons of at least two rhabdodontid and two

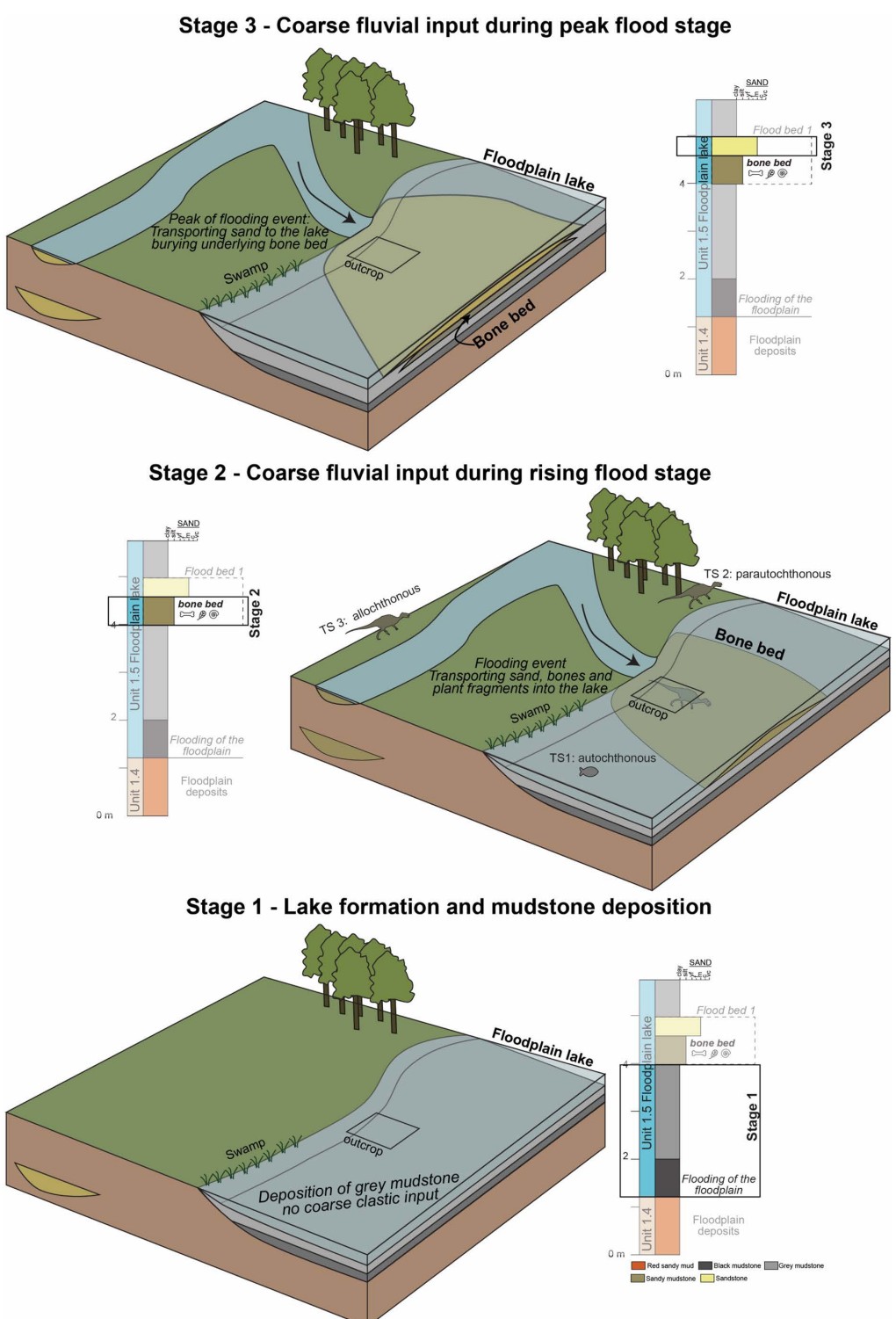

**Fig 17. Reconstruction of the depositional processes producing the studied fossiliferous sedimentary sequence at site K2, Vălioara (Densuş–Ciula Formation).** Stage 1: Formation of the lake and deposition of black/grey mudstone from suspension settling in a quiet environment. Stage 2: Fluvial flood transporting coarser material, plants and vertebrate remains in the dominantly muddy lake. TS. - taphonomic set (see text for details). Stage 3: Peak flooding event burying the fossil-bearing bed with sandstone. Subsequently, renewed suspension settling (see Stage 1) dominates deposition until the next flood.

titanosaur individuals, despite the fact that many of these skeletal elements were – strictly speaking, in accordance with the definition of [176]- recovered 'isolated'.

The well-preserved and taxonomically identifiable isolated skeletal elements form the most diverse part of the site K2 assemblage, containing mixed remains of aquatic (fish), semi-aquatic (amphibians, dortokid turtles, certain crocodyliforms) and purely terrestrial (other crocodyliforms, squamates, theropod and hadrosauroid dinosaurs, pterosaurs, mammals) animals representing at least 15 different taxa. This material includes mostly small-sized vertebrate remains and is dominated by teeth, but more rarely a few larger and relatively well-preserved elements are also present (e.g., theropod limb bones, turtle plates). Bones and teeth from this set are usually unabraded and unweathered, but are thoroughly disarticulated and dissociated to the extent that it is virtually impossible to confidently conclude that any two skeletal elements may have originated from the same individual. This sub-sample of the K2 material can be described as a proper microfossil assemblage, since it is clearly dominated by small teeth and the proportion of bones larger than 5 cm does not exceed 10%. Based on these taphonomical features, this part of the assemblage most probably represents an attritional assemblage concentrated in the lacustrine delta setting. The relatively good preservation of the larger elements from this subgroup (e.g., theropod and pterosaur limb bones) suggests that they were probably transported from a nearby location to the accumulation area, rather than that these elements represent the fauna of a more distant region. This diverse attritional microvertebrate assemblage was gathered and transported by the flood that deposited the flood-related sediments of the delta system. Since this material contains the remains of animals that died at different places and at different times, as and assemblage it can be characterized by a relatively lengthy time averaging, ranging from ≤ $10^2$ to > $10^3$ yrs (based on [175]).

The last taphonomical grouping identified within the K2 assemblage consists of associated and even articulated skeletal elements that represent remains of larger-sized, terrestrial animals – the herbivorous rhabdodontid and titanosaur dinosaurs as well as the turtle *Kallokibotion*. According to the sedimentological investigation of the K2 bonebed, these incomplete skeletons were accumulated within a delta intruding into the lake, an environment characterised by a sudden decrease in transport energy. A scenario of *in situ* death occurring strictly in, or nearby, the place of deposition is not conceivable, because it would entail that remains of at least four individuals belonging to two different relatively large-bodied (more than 2 m long) taxa were somehow crowded into a very small area (about 4.75 m²), and for such an extensive accumulation of carcasses to occur in a localized place, the action of some sort of concentration processes is required. Furthermore, the sedimentological investigation pointed out that the burial place of the carcasses was subaqueous and relatively far from the lake margin, also at odds with the possibility of *in situ* death of some terrestrial animals.

On the other hand, an alternative scenario according to which the associated skeletal subset identified at site K2 has been transported by fluvial action into the depositional environment piece-by-piece, and mainly in the form of an assemblage composed of isolated remains, can be also safely discarded, as the remains themselves are considered too large to be hydraulic equivalents of the encasing sediment, but also because the representatives of both dinosaur taxa exhibit high degrees of skeletal completeness. It is thus more likely that the skeletons representing this particular subset of the K2 assemblage were floated into the site as bloated carcasses sometime after the death of the animals – a transportation mode that is an often cited mechanism for this type of taphocoenoses (e.g., [43,177–181]).

According to this scenario, as the carcasses were drifting downstream on the river, the transport energy of the latter suddenly dropped in the delta built into the lacustrine area, allowing the carcasses to pile up on top of each other. After their accumulation, the carcasses were exposed to subaqueous destructive processes (e.g., decay, disarticulation by physical agents and/or aquatic vertebrate and invertebrate scavengers) during which different parts of the skeletons were separated from each other. Although estimating the time elapsed between death and the disarticulation of the carcasses is extremely difficult, a comparative literature survey suggests that this most probably represented weeks or months rather than years [154,182]. The relatively short duration of taphonomic processing of the carcasses is independently confirmed by sedimentological studies, which also indicate that the sediments of the bonebed accumulated over a period

of months. Once the decomposition of the carcasses reached an advanced stage, their different parts could be separated and scattered even more easily by weak water currents. The presence of jaw and skull elements as well as that of distal limb elements (tibia, fibula, phalanges) in the associated rhabdodontid assemblage [12] suggests that much of the decomposition and dispersion of the carcasses definitively occurred *in situ*, after their stranding on the delta, as these skeletal elements are easily separated from the body and removed during transport as soon as the decomposition of soft tissues allows it (e.g., [173,177,183,184]). However, the number of the recovered vertebrae in the K2 rhabdodontid assemblage is far below their predicted count (Table 4), suggesting some degree of removal of the less dense elements after disarticulation. The presence of both axial and appendicular skeletal parts in the K2 titanosaur assemblage also indicates that the skeletons were fairly complete, and that the different skeletal parts were still held together by ligaments when they arrived in the stranding area.

Our assessment according to which associated skeletons of at least four individuals representing two different larger-bodied dinosaur taxa were deposited and preserved over a very small, 4.75 m² area during a relatively brief time period (months) suggests interesting implications regarding their death. Indeed, the roughly simultaneous demise of these animals is supported by the following observations: 1) the deposition time of the bonebed was estimated to be weeks or at most months according to the sedimentological and taphonomical investigations; 2) all the associated skeletal material from K2 was recovered from the same, approximately 0.5 m thick level within Flood Bed 1 (Fig 3B), in which the bones were found in close association, almost jumbled together across a very small area; 3) remains from a number of associated skeletons are mixed together within this relatively thin layer, indicating that the carcasses were deposited and decomposed roughly in the same time; 4) this associated and articulated subset shows a different taphonomical signature from other sets of fossils (isolated bones and bone pebbles) from the same site; and 5) the associated skeletal material subsets from K2, regardless whether they represent rhabdodontids or titanosaurs, display broadly uniform taphonomic features (e.g., very limited abrasion, quasi-absence of weathering, often complete epiphyses of the long bones).

The accumulation of skeletons from multiple individuals belonging to different taxa within a small area suggests that the bonebed is the result of a series of mortality events occurring within a short period of time, which also raises the possibility that the deaths of the individuals represented by the associated skeletons may be attributed to a single main cause of death. Although known mass mortality-reflective dinosaur taphocoenoses are generally represented by monotaxic or at least monodominant assemblages (e.g., [154,162,173,178,185–190] and references therein), [191] pointed out that mass death events do not always produce monospecific bone assemblages and that the resulting diversity is often similar to that of the cumulative assemblage. For example, [192] reported a mass mortality assemblage from the Upper Cretaceous Jack's Birthday Site (Campanian Two Medicine Formation of Montana, USA) where the remains of at least 15 individuals representing three hadrosaurid genera, as well as four theropod individuals were preserved, whereas the drought-induced mass death assemblage from the Upper Jurassic Cleveland-Lloyd Dinosaur Quarry (Morrison Formation of Utah, USA) contains remains of dozens of individuals from a minimum of twelve genera [193], besides other examples of non-dinosaur mass death assemblages that include mixes of several taxa (e.g., [191,194] and references therein). According to these previous data, it cannot be excluded that the two rhabdodontid and two titanosaurian individuals known from the single bonebed of site K2 represent results of a single major mortality event.

However, according to its currently accepted definition, a true mass mortality assemblage contains remains of animals (eventually those of herds or other social groupings of animals) that died over a brief time span due to a single, non-repeating agency of death (e.g., [43,178,187,190,191,192,195,196]). From these requirements, that of a short period of mortality appears to be justified in the case of the associated skeletons from the K2 assemblage, as both sedimentological and taphonomic investigations document convincingly a brief time interval for the accumulations of the carcasses. Meanwhile, the second criterion of a single, non-repeating agency of death can be hardly defended based on our data collected at site K2. For example, during a prolonged period of drought, members of different taxa can congregate around

remaining water bodies (such as the lake reconstructed for the fossiliferous K2 local succession), in which case it is conceivable that several different species will be present in the resulting mass death assemblage [190,193,197]. However, drought as a possible cause of mass death can be securely discarded for site K2 because (1) there are no sedimentological evidences at K2 or in its hosting unit (unit 1.5 or 'grey Vălioara beds') for periodic severe drought (such as mud cracks, mud rip-up clasts or evaporites etc.); and (2) the overall sedimentology as well as the palynology-suggested vegetation convergently indicate a wetland habitat with elevated groundwater table in the wider area around, and during the genesis of, site K2 [11,20], in good accordance with the subtropical climate reconstructed for the broader Haţeg Basin area during the latest Cretaceous [34,164]. Furthermore, both the local faunal composition and certain taphonomic observations (i.e., most of the skeletons are preserved strongly disarticulated; extreme rarity to absence of definitively juvenile individuals) argues against the drought-related mass death hypothesis [186,187,193,197,198].

Alternative scenarios proposed previously to explain this type of taphocoenoses, such as fire-induced mass mortality (e.g., [199,200]), also remain poorly supported for site K2, given the lacustrine-marshy, water-logged environment reconstructed for it, as well as the absence of charcoal, and that of charred wood or bone. Finally, there is no taphonomic evidence, such as extensive bioturbation of the fossil-bearing sediments, widespread preservation of skeletal articulation, indications of *in situ* death, or upright skeleton poses, that would support a process of miring in soft ground [201,202], and such a scenario would be also difficult to reconcile with the relatively higher water depth reconstructed for the depositional setting of the fossiliferous bed at site K2. Disease or poisoning, other possible causes of mass-death events, are theoretically possible [192,195,203], but reliable detection of such agents is difficult.

However, it is more likely that the associated rhabdodontid and titanosaur material discovered at site K2 represents the remains of animals drowned slightly before or during a peaking flood stage. Indeed, several previous studies have documented that a fair number of specimen-rich bone accumulations of dinosaurs could have been the results of mass drowning events affecting large herds of herbivores, some of which occurred while the herds attempted to cross flooded rivers [43,154,162,178,179], while in other instances these were impacted by extensive coastal plain flooding events (e.g., [202,204]). Such a scenario would be potentially further supported by the observation that drowning is a relatively common cause of death in terrestrial animals (e.g., [154,162,178,179,205–207]), as well as by the fact that the K2 bonebed appears to be associated with sediments accumulated during severe flooding events.

On the other hand, mass death assemblages caused by mass drowning are only conceivable for terrestrial animals with a herding lifestyle because it would require that numerous individuals of the respective taxa to be in close proximity to each other right before their death (i.e., were members of herds or other types of social groupings). Although herding lifestyle has been confirmed separately both for different ornithopods [173,179,187,189,208–215] and for sauropods [183,188,201,216–218], we are unaware of any fossil assemblage that would document the fact that representatives of these two groups (or any two distinct higher-level taxa) lived in taxonomically mixed, common herds – this is why most dinosaur mass death assemblages attributed to mass drowning are mono-taxonomic or at least heavily mono-dominant, paucispecific [162,189,204]

Despite the fact that detailed taphonomical evidence for a herding lifestyle is absent for the different Haţeg Basin dinosaurian herbivores such as the rhabdodontids and the titanosaurs, sets of bones with similar taphonomic signatures belonging to titanosaurs and/or rhabdodontids were noticed recurrently in different bonebeds in the past (e.g., La Cărare site; Groapă site; Kadić's site I; NVS site; [3,4,5,11]), and in such cases these are derived from several individuals of different body sizes, and often show a distinctive taphonomic signature different from that of other sets of bones from the same site. Accordingly, [4] have suggested that the occurrence of these sets of remains suggests that several individuals of a given taxon experienced a common taphonomic history, lending some support to the idea that rhabdodontids and titanosaurs were, at least occasionally, gregarious. Nevertheless, none of these previously studied occurrences would suggest large-sized herds of hundreds to thousands of individuals congregated, as was proposed for other herbivorous dinosaurs [e.g., 185,187,204].

In conclusion, it cannot undoubtedly be confirmed that the associated material of rhabdodontid and titanosaurian dinosaurs from site K2 includes the remains of victims of a mass death event – in fact, it is more likely that these deaths occurred individually across the proximal wetland area and may had been spread over a period of weeks to a few months, immediately preceding, or during, the time of the severe flood event generating Flood Bed 1 (Fig 3B). Nevertheless, we stress again that the unusually high number of incomplete skeletons confined into the small area of the K2 bonebed – an occurrence as yet unmatched by any other currently known Hațeg Island vertebrate accumulation – together with their similar taphonomic characteristics, indicates that their accumulation must have occurred within a relatively short period of time. We thus interpret the K2 thanatocoenosis as a mass accumulation (as opposed to a mass death assemblage), where the exceptionally high frequency of the skeletal remains is determined not by a common cause of death acting in a catastrophic manner, but instead by a taphonomic agent/mechanism promoting increased chance of concentration of the skeletons during a brief time interval.

Our preferred taphonomic scenario thus state that roughly synchronously with the accumulation time of the sediments building up Flood Bed 1 of the K2 succession (Fig 3B), a relatively large number of rhabdodontid and titanosaurian dinosaurs died in the neighbourhood of the site. The cause(s) of these death events is(are) currently unknown, but since the accumulation of carcasses took place rather rapidly following the death of the animals – as supported by their low weathering/abrasion stages and partial association/articulation, suggestive of a floating carcass transport – and since the genesis of the main bonebed of site K2 is considered to be a short-term event related to a severe flooding episode (see discussions above), it is tempting to hypothesize that the environmental stresses induced by the flooding period may have heightened for a brief time interval the mortality rate in the affected wetlands above normal, background levels, albeit without necessarily contributing directly to it, since large and mobile herbivores – as reconstructed for rhabdodontids and titanosaurs – may easily avoid being drowned by moving away from the endangered areas. Furthermore, the fact that the flooding itself and its related environmental stresses did not cause directly a mass mortality event, as was suggested previously for the severe floods affecting the Campanian North American coastal plain wetlands (e.g., [204]), is also indicated by the apparently non-selective nature of the death agent(s). Indeed, both of the most affected groups (rhabdodontids and titanosaurs) are represented in the K2 thanatocoenosis by sub-adult to adult, but probably not senescent individuals ([12]; B. Vila, pers. comm. 2025), and thus neither by individuals considered to be more prone to occur with priority, nor by age classes that should be also represented, in catastrophic mass-death assemblages, i.e., juveniles and old adults (e.g., [12,192]). Subsequently, the carcasses lying around in the flood-affected wetlands were swept together and carried into the lake by the raising floodwaters.

The depositional environment detected at the bonebed of site K2 (a small flood induced lacustrine delta built at the discharge point of a river) can be considered as a suitable trapping place, where alluvial streams that transported the carcasses and sediment downstream during flooding events entered into the lake. This lead to a sudden decrease in the flow velocity and thus deposition of their sediment load potentially building deltaic bodies. Also within this environment, carcasses could have arrived and accumulated at slightly different times, while the river flood persisted, similarly driven by the sudden changes in flow conditions as the river entered the lake. The presence of carbonized tree limbs and trunks, frequently encountered around the bone concentrations in K2, could also point towards a lacustrine delta environment as these can accumulate significant amounts of plant debris, generated by the same processes that caused the accumulation of the sediments and the carcasses themselves. Indeed, although most of the tree trunks at site K2 are lying more or less parallel to the bedding plan, there are several examples where these stand oriented vertically, also consistent with the scenario of sudden drop in current energy (Fig 2C). Furthermore, these tree trunks themselves may have contributed to the genesis of the local vertebrate thanatocoenosis, since bloated carcasses drifting into this trunk-laden delta environment, could got stuck on the stacks of logs already accumulated as these either made the water depth to drop steeply in their proximity or even formed log jams (e.g., [154]).

However, it is worth emphasizing that the large-scale accumulation of skeletal material is only present in the main bonebed layer (corresponding to Flood Bed 1 in Fig 3B), and that other flood events recognized higher up in the section

(Flood beds 2 and 3, Fig 3B), albeit created through the same sedimentary processes acting within a similar depositional environment, did not resulted in the presence of significant amounts of vertebrate material, with only invertebrates recovered from these. This difference suggests that 1) there was an even higher than usual flood-time mortality rate prior to the accumulation of the K2 bonebed; or 2) the transport energy of the floods was no longer sufficient to accumulate significant amounts of vertebrate remains (as well as larger tree trunks) into the K2 depositional area, with the two options not being mutually exclusive. However, since mortality rates appear to not have been raised significantly during flooding events in the Vălioara area during the earliest Maastrichtian (as discussed above), we contend that the main factor contributing to the genesis of the K2 assemblage was the more efficient action of the agent/mechanism of accumulation during the first flood period recorded in the local section – hence, as we underscore once more, the K2 assemblage is more properly regarded the result of an exceptional mass accumulation event, instead of that of a catastrophic mass death event.

### 5.3. Fossil vertebrate assemblage of site K2 and its significance for the composition and evolution of the Haţeg Island vertebrate faunas

Detailed lithostratigraphic, magnetostratigraphic and biostratigraphic investigations of the different vertebrate sites from Haţeg Basin in recent decades, in some cases also supported by radiometric dating information, have led to a better understanding of the temporal distribution of the different faunal assemblages discovered here, as well as in the wider Transylvanian area (see S4 Table and Fig 18), revealing potential patterns of faunal changes, or else the presence of stasis, on Haţeg Island during the latest Cretaceous (e.g., [4,20,28,164,219–222]). Based on the available data, the Densuş vertebrate site is probably the oldest currently known uppermost Cretaceous vertebrate occurrence from Haţeg Basin (Fig 18), but it yielded only one very incomplete rhabdodontid specimen consisting of fragmentary jaws with *in situ* maxillary and dentary dentition [4]. The bone-bearing level represented by the Densuş site is located in the Lower Member of the Densuş-Ciula Formation, a succession that was dated by zircon U-Pb geochronometry to the 'middle' Campanian (= early part of late Campanian) [20,35]. Further up-section, site K1 reported by [11] also yielded disperse rhabdodontid remains, and this occurrence, together with the slightly younger [20] rhabdodontid material collected from site K2 (see also [12]) as well as the roughly similar-aged rhabdodontid remains discovered in the Scoabă section, within the Sibişel Valley succession of the Sînpetru Formation near Sânpetru (e.g., [123]), clearly indicate that rhabdodontid dinosaurs were common members of the earliest faunal assemblages documented from the Haţeg Basin area. Similarly, rhabdodontids currently represent the earliest-appearing continental vertebrates in the Campanian-to-Maastrichtian transitional succession known from Petreşti-Arini in the southwestern Transylvanian Basin, as well [223], with their lowest occurrence within the middle part of the upper Campanian [224]; they are also the only vertebrate group that is continuously recorded up until the lowermost Maastrichtian part of the local succession (e.g., [223,225]). Considered together, all available data indicate that rhabdodontids were one of the, if not precisely the oldest-appearing faunal elements of the latest Cretaceous Haţeg Island ecosystems. Moreover, it is worth emphasizing that the remains of rhabdodontids represent the third most frequently found fossils at site K2, after titanosaurs and *Kallokibotion* turtles, indicating that these herbivorous dinosaurs were also a common member in these earliest faunas.

Furthermore, rhabdodontids remained dominant members of the local paleofaunas until the late Maastrichtian, as documented by the common occurrence of their remains at the Nălaţ-Vad vertebrate locality (e.g., [3,123,226] see Fig 18). Accordingly, rhabdodontids most likely represent the longest-living dinosaur group of Haţeg Island, with a stratigraphic range extending from the middle part of the late Campanian to the late Maastrichtian, and thus indicating the continuous presence of this group for a timespan of about 7 to 7.5 Ma at the least.

Currently two rhabdodontid genera are known from Haţeg Basin: *Zalmoxes* [122], including two species (*Z. robustus* and *Z. shqiperorum*), and the more recently described *Transylvanosaurus platycephalus* [124]. The latter taxon was discovered, and is currently only known, from lower/upper Maastrichtian boundary beds near Pui (belonging to the informal 'Pui Beds' unit; see [3]), whereas *Zalmoxes* (formerly also called *Mochlodon* or *Rhabdodon*; e.g., [25,227]) was used

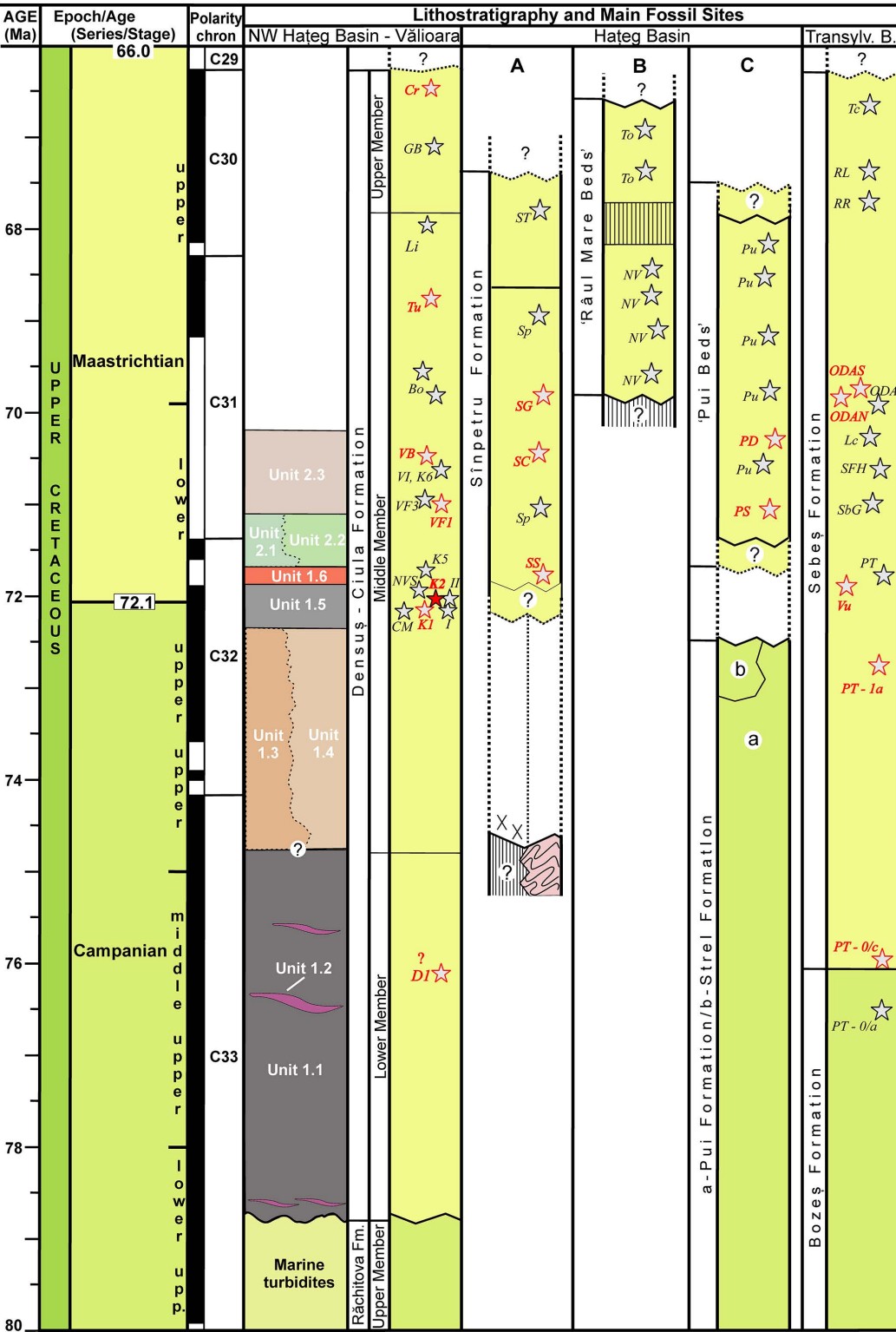

**Fig 18. Stratigraphic position of site K2 (Neagului Valley, Vălioara – red star), Middle Member of the Densuş-Ciula Formation, relative to other important uppermost Cretaceous Transylvanian vertebrate sites (grey stars), including those mentioned/discussed specifically in the text (stars outlined in red), from the Haţeg (A – Sibişel Valley section, B – Râul Mare section, and C – Pui section) and southwestern Transylvanian**

basins (updated and modified from [3,20]); note that, as already acknowledged by [3], the relative positions of the sites within the different lithostratigraphic units is reliable, albeit not always precisely established, whereas that between these units is largely tentative. For the definitions and lithological make-up of the sedimentary units (units 1.1 to 2.3) separated in the lower part of the Densuș-Ciula Formation around Vălioara, see [20]. Abbreviations of localities and sites: I, II, VI - Kadić's sites I, II and VI, Vălioara; Bo – Boița; CM – Ciula Mică; Cr – Crăguiș; D1 - Densuș; GB – General Berthelot; K1, K2, K5, K6 - sites K1, K2, K5 and K6, Vălioara; Lc – Lancrăm; Li – Livezi; NV – Nălaț-Vad (several different sites); Nvs – New Vertebrate Site, Vălioara; ODA – Oarda de Jos (several sites); ODAN – Oarda de Jos, microvertebrate lens; ODAS – Oarda de Jos, 'Stocks' accumulation; PD – Pui Depozit; PS – Pui Swamp; PT – Petrești (different other sites); PT – 0/a - Petrești, level 0/a; PT – 0/c - Petrești, level 0/c; PT – 1a - Petrești, level 1a; Pu – Pui (several different sites); RL – Râpa Lancrămului; RR – Râpa Roșie; SbG – Sebeș-Glod; SFH – Secaș Feții Hill; SC – Sânpetru Cărare; SG – Sânpetru Groapă; SS – Sânpetru Scoabă; ST – Sânpetru Terminus; Tc – Teleac; To – Totești-Baraj (several different sites); Tu – Tuștea-Oltoane; VB – Vălioara Budurone; VF1 – Vălioara Fântânele 1; VF3 – Vălioara Fântânele 3; Vu – Vurpăr (for references regarding these sites, see text); '?' marks tentative (i.e., not well constrained) position of sites/boundaries.

historically to include all rhabdodontid fossils collected from several different vertebrate sites across Hațeg Basin and the wider Transylvanian area (e.g., [8,26,110,112,122,123,223,226–231]), suggesting a potentially very wide distribution of this genus both stratigraphically and geographically. However, the latest discoveries together with reassessment of the older ones had demonstrated that the previous practice of uncritically referring mainly isolated rhabdodontid remains to *Zalmoxes* was erroneous, hindering a proper understanding of the once-existing rhabdodontid diversity on Hațeg Island (see discussions in [12,124,232]). A detailed taxonomic revision of the entire available Transylvanian rhabdodontid material is currently underway, and the present stage of this effort already indicates that the rhabdodontid remains classified previously under the name *Zalmoxes* most probably represent several different taxa (e.g., [12,124,232]). It is also clear even now that the rhabdodontid material from site K2 (despite not being formally separated as a distinct taxon) is different taxonomically both from *T. platycephalus* as well as from the two species of *Zalmoxes*, based on the currently accepted diagnoses of the latter [12], and may well represent a new genus within the clade.

It is currently unclear whether the distinction between *T. platycephalus* and the K2 rhabdodontid indicates temporal separation between these two taxa (given the earliest Maastrichtian age of the K2 material versus the late early Maastrichtian age of *T. platycephalus*), or rather a geographical and/or paleoecological one (given the well-drained floodplain environment reconstructed for the type locality of the Pui taxon, and the generally better-drained depositional environments that characterize the entire Bărbat River succession from Pui [3,124,164]), or else a combination of thereof. Meaningful distributional comparisons between the K2 rhabdodontid and the two *Zalmoxes* species are deferred here for the moment, pending their ongoing revisions. We note, nonetheless, that the type localities of both of these *Zalmoxes* species are located in the Sibișel Valley section of the Sînpetru Formation, south of Sânpetru, which may already indicate some degree of geographical/environmental separation between these taxa, and maybe even a temporal one, although reliable geochronological constraints are scarce for the Sînpetru Formation, as are the means of tight stratigraphic correlation between the continental uppermost Cretaceous successions cropping out in the Vălioara and, respectively, the Sânpetru areas.

Besides the rhabdodontids, the other two common herbivorous dinosaur groups of Hațeg Island (e.g., [1,3,4,26,114,]), the hadrosauroids and the titanosaurian sauropods, are also represented within the vertebrate material recovered from site K2. Of these, the sauropod remains are the most frequently collected fossils from the site, while those of the hadrosauroids are much less abundant, being represented only by few isolated teeth and a single dentary fragment (Table 2). The titanosaur remains from site K2 do not represent the oldest record of the group from the Densuș-Ciula Formation, since isolated titanosaur fossils had been recovered previously from stratigraphically lower levels within the Densuș and Ciula Mică areas (e.g., [5,11]). Remarkably, even associated titanosaur remains – a relatively rare occurrence in the Hațeg area, overall [4,5] – had been reported from near Ciula Mică, including here the type material of the dwarf *Magyarosaurus dacus* [5,9,11], from sites that are probably marginally older than K2. Nonetheless, their occurrence at site K2 does represent a novelty in two different aspects.

For one, it is the first instance when the association of the recovered titanosaur elements is documented in detail, based on first-hand field observations during excavation as well as on in-depth taphonomical investigations (see above), unlike the previously identified important titanosaur associations from the wider Vălioara area (i.e., assemblages A, K and M, as well as individuals C, E and S of [5]) for which skeletal association is mainly inferred based on overall similar preservational features, matching size and, only occasionally, indirect evidence from written accounts [5]. The K2 material was designated preliminarily as Individual V by [5], albeit based on the much more limited material and information available at that moment in the accounts of [11], respectively [134]. Currently, this material is known to include the associated remains of at least two (most probably conspecific) individuals (see above), and as such, it should be designated as Assemblage V instead. The study of this assemblage is currently underway, and although it was first tentatively assigned to a *Paludititan*-like taxon [11], subsequently – with the discovery of further remains at K2 – this assignment had been questioned [5,134], and its taxonomic identity remains undecided, pending our ongoing investigations.

Furthermore, regardless of the outcome of these investigations, the importance of the K2 titanosaur material is already acknowledged. Indeed, since recognizing skeletal association was shown recently to be quintessential for a reliable assessment of taxonomic identity of different, mainly isolated fossils in the Transylvanian continental vertebrate faunas (e.g., [4,5,12,124]), the associated K2 titanosaur material has the potential to contribute to a great extent to the understanding of the identity and affinities of the Hațeg Island titanosaurs.

Remarkably, titanosaur remains appear to be completely absent in deposits from the southwestern Transylvanian area (e.g., at Petrești and Vurpăr) that are assessed to be roughly synchronous with those from the Vălioara area (e.g., [3,223,224]), whereas they are known to also occur in the roughly contemporaneous Sânpetru-Scoabă local succession of the Sînpetru Formation along the Sibișel Valley, near Sânpetru ([3]; ZCsS, pers. obs.). Meanwhile, as already noted above, rhabdodontids are represented in all these local successions (see comments below, 5.4. Paleoecological significance).

Finally, it should be pointed out that, just as in the case of the rhabdodontids, titanosaurs are known to occur up to the upper end of the currently known fossil record range of Hațeg Island, well into the upper Maastrichtian. In the Transylvanian Basin, they are reported from several upper Maastrichtian (Tier 4 of [3]) localities [3,110,231]. Meanwhile, in Hațeg Basin, despite the current absence of clearly Tier 4-distributed titanosaur skeletal elements, they were identified as potential egg-layers of megaloolithid eggs reported in large numbers from the Tier 4 Totești-Baraj locality by [233]).

Albeit represented only by a few isolated remains at site K2, unlike the other two major herbivorous dinosaur groups, hadrosauroids were undoubtedly present in the oldest faunal assemblages of Hațeg Basin. Indeed, the fossils recovered from site K2 confirm earlier hypotheses according to which the main herbivorous dinosaur clades making up the typical titanosaurian – rhabdodontid – hadrosauroid Hațeg Island assemblages [3,26,112] were already present in the Hațeg Basin faunas from the earliest times, and they remained common faunal elements throughout the entire Maastrichtian of Hațeg Basin [3]. However, unlike the cases of rhabdodontids and titanosaurs discussed above, and whose remains are already recorded in older sites, site K2 marks the stratigraphically lowest confirmed occurrence of this group in the Vălioara area, and thus possibly across the entire Hațeg Basin, which puts their earliest apparition into the very early Maastrichtian, close to the Campanian/Maastrichtian boundary [20].

Meanwhile, although hadrosauroids were long considered to appear only sometimes during the early Maastrichtian in the Transylvanian Basin (e.g., [3,224]), new (albeit rare) discoveries document their oldest occurrence there already in the uppermost Campanian [234]. Accordingly, their oldest occurrence datum succeeds to that of the rhabdodontids, but apparently predates significantly that of the titanosaurs, whereas the reverse of this pattern appears to be the case in Hațeg Basin – titanosaurs seem to occur before the first hadrosauroids. Nonetheless, even if the current evidence suggests that hadrosauroids might have been present somewhat earlier in the lowland, coastal plain environments of southwestern Transylvania than in the more inland, alluvial fan settings of Hațeg Basin, they soon appear to have become established in these upland areas as well, as they are continuously present (albeit still being subordinate in abundance; see below, 5.4. Paleoecological significance) throughout the overlying succession of lower Maastrichtian sites from the Vălioara area [3,11,17,112,144].

Unlike the three common herbivorous dinosaur clades discussed above, the remains of ankylosaurs – the fourth group of dinosaurian herbivores documented in Haţeg Island – are completely absent from the K2 assemblage, and ankylosaurian fossils are also currently unknown from other vertebrate sites from the Vălioara area that are placed more or less at the same stratigraphic level with site K2 (e.g., Kadić's sites I through IV, as well as sites NVS and K1 to K4; see [11,235]). This absence seems to confirm previous assumptions according to which ankylosaurs were not members of the earliest faunal assemblages in Haţeg Basin, and that they were probably later-arriving immigrants from the Transylvanian Basin, where this dinosaur group was already present by the late Campanian [96,236]. The oldest Haţeg Basin ankylosaur remains are known from slightly younger beds of the Densuş-Ciula Formation, at the Fântânele microvertebrate bonebed [236], assessed to represent the early part of early Maastrichtian [20]. From then on, they are continuously reported throughout the Maastrichtian from the different uppermost Cretaceous outcropping areas (e.g., [236–238]), including in the uppermost part of the Densuş-Ciula Formation, from the uppermost Maastrichtian sediments of Crăguiş site that provided a single tooth fragment referred to this group [3]. Thus, based on the currently available fossil record, ankylosaurs were extremely rare in the entire Haţeg Basin area, but were missing entirely from the earliest Maastrichtian vertebrate communities.

Besides the rhabdodontids and titanosaurs, the basal testudine *Kallokibotion* turtles are also detected from the lowermost to the uppermost part of the Maastrichtian sedimentary succession of Haţeg Basin and are often the most abundantly represented taxon in the different vertebrate sites. The oldest-known important occurrence of *Kallokibotion* in Haţeg Basin is represented by the large number of isolated and articulated remains collected from site K2 (this taxon was reported, nonetheless, from the slightly older site K1 as well; [11]), and the abundant material by which it is represented here also suggests that this turtle was already a common member of the earliest continental faunal assemblages in Haţeg Basin. Further up-section in the uppermost Cretaceous continental beds, all taxonomically diverse multitaxic fossil accumulations known from Haţeg Basin remain dominated (in number of identifiable specimens) by terrestrial organisms, and the dominant taxa are usually *Kallokibotion* turtles besides one or more of the three major herbivorous dinosaur groups (e.g., [3,4,11]) – a paleofaunal composition feature that is already well detectable in the lowermost Maastrichtian, at site K2.

Meanwhile, although the presence of *Kallokibotion* (or at least of *Kallokibotion*-like turtles) is reported in most known uppermost Cretaceous fossiliferous localities of the southwestern Transylvanian Basin (e.g., [86,109,110,231,239]), these appear to be absent in the uppermost Campanian to lowermost Maastrichtian succession cropping out at Petreşti [223], in deposits that are slightly older to roughly contemporaneous with those sampled at site K2 (see S4 Table and Fig 18). Their absence there may indicate either that *Kallokibotion* appeared diachronously, i.e., somewhat later, in the southwestern Transylvanian Basin compared to Haţeg Basin, or else that some degree of selective habitat preference may have prevented *Kallokibotion* to live in the low coastal plain settings represented within the Petreşti succession, as this taxon is known to be a land-dwelling form adapted to drier environments, unlike the more aquatic dortokids (e.g., [86,240]) which, on their turn, were reported to be present at Petreşti [223]. Regardless of the reason behind this diachronism in distribution, it is worth emphasizing here again that the most common Haţeg Island turtle – the terrestrial *Kallokibotion* – was already abundant at the beginning of the Maastrichtian in Haţeg Basin, while being absent (Petreşti) to moderately common (Vurpăr) in largely contemporaneous beds of the southwestern Transylvanian Basin, whereas the opposite trend seems to be the case for the aquatic dortokids.

Crocodyliforms, represented by four generalized morphogroups (most often identified as the ziphodont '*Doratodon*-type', the leaf-shaped toothed 'atoposaurid' or '*Theriosuchus*-type'*,* the somewhat durophagous '*Acynodon*-type', and the largest-bodied, and most abundantly represented '*Allodaposuchus*-type'), are chronostratigraphically widely distributed in Haţeg Basin [3,97]; the same loosely defined groups are also reported throughout the southwestern Transylvanian Basin (e.g., [231]) (S4 Table), as well as – albeit less diversified, and with a more restricted fossil record – in the more westerly Rusca Montană Basin [3,100,101]. All of the crocodyliform groups previously identified from

Haţeg Basin are also detected at site K2, suggesting that taxonomically and ecologically diverse crocodyliform assemblages were already a constant component of the Haţeg Island faunas from the earliest Maastrichtian.

An almost identical pattern can be identified in the case of the theropods. According to a recent overview of their fossil record in Haţeg Basin, they are represented here mostly by isolated teeth that can be referred to a small number of distinct morphotypes, most probably representing three different taxa: indeterminate velociraptorine dromaeosaurids, as well as the incertae sedis taxa *Richardoestesia* and *Euronychodon* [136]; of these, as discussed previously, two (velociraptorines and *Richardoestesia*) are also present in the K2 assemblage (Fig 8F-I). Previous reports concerning the occurrence of small theropods in Haţeg Basin show that these, albeit not very abundant, are commonly encountered in both microvertebrate (e.g., [17,135]) and macrovertebrate (e.g., [8,137]) localities. Furthermore, their remains are known to be spread widely spatially, from Ciula Mică in the west [16] to Pui in the east [17,94], and from Crăguiş in the north ([3]) to Sânpetru [3,137] and Toteşti [135] in the south. They are also wide-ranging temporally, from lowermost Maastrichtian occurrences (e.g., [15–17,120]) to sites probably representing the upper part of the upper Maastrichtian [3,135].

Although sometimes the three taxa are reported to co-occur within the same locality (e.g., [17,120,135]), *Richardoestesia* and, especially, velociraptorines appear to be somewhat more commonly and more widely encountered [3,8,17,94]. Outside of Haţeg Basin, these same two theropod taxa were also reported from the Rusca Montană [91,101] and southwestern Transylvanian [90] areas, with at least *Richardoestesia* possibly present in the latter region since as early as the latest Campanian [96]. In the Haţeg area, velociraptorines may had appeared earlier than other theropod taxa at Ciula Mică [3,16], but the age of this particular locality is poorly constrained, and yielded only a very small number of fossils, so the absence of other taxa may be a by-product of limited sample size instead of real faunal composition signal, especially since only slightly up-section velociraptorines are already associated with *Richardoestesia* (including at K2, see above) and even with *Euronychodon* (e.g., [17,120]). Overall, the Transylvanian Landmass theropod occurrences show that the velociraptorine-*Richardoestesia* association became established very early in the faunal composition, as also witnessed by site K2, that small theropods (especially the velociraptorines and *Richardoestesia*) were geographically widespread, and that these persisted up until the end of the known local fossil record.

Several other taxa from the K2 assemblage are too poorly represented, and in most cases too undiagnostic, as well, as to allow identification of finer-grained faunal trends; their occurrence suggests, however, that these taxa were already present in the earliest known paleofaunas of Haţeg Basin. Four well-preserved ganoid scales collected from site K2 indicate that lepisosteiform fish were present in the local community (Fig 8A). However, despite the relatively extensive screen-washing, no other fish remains were identified from the site, implying that this group was rare in the K2 paleoenvironment. Extreme rarity of fish remains is a feature that is already well documented in the fossiliferous localities of Haţeg Basin as a whole (e.g., [4]), even if several of the known fossil occurrences originate from sites that indicate water-rich environments. In this regard, the Haţeg Basin local assemblages depart from those documented in the southwestern Transylvanian areas where, although fish remains are also relatively rare overall, these appear to be nevertheless better represented, somewhat more abundant at least in some localities/fossiliferous sites such as at Petreşti (e.g., [96,231]) or Oarda de Jos (e.g., [90,109,230,241,242]).

Although amphibian remains are also very rare, both commonly detected amphibian groups (Anura and Albanerpetontidae) are present in the K2 assemblage. Discovery of a well-preserved albanerpetontid dentary is paleontologically important (Fig 8B-C), as it represents the oldest albanerpetontid fossil from Haţeg Basin [83]. Based on preliminary investigations, the K2 specimen appears to be similar to either *Anoualerpeton* [84] or to *Shirerpeton* [85], both taxa known from the Lower Cretaceous, and thus this fossil possibly indicates a previously undetected new species of Albanerpetontidae from Haţeg Basin [83]; a more detailed investigation of this specimen is currently underway.

The squamates are represented at site K2 by a maxilla (Fig 8D-E), as well as by isolated teeth and vertebral fragments, As the best preserved and most diagnostic specimen, the maxilla tentatively indicates the presence of a taxon belonging to Barbatteiidae [83], a clade of teiioid lizards thus far known to be endemic to the Transylvanian Landmass

[89]. Barbatteiid lizards were previously reported from Haţeg Basin deposits representing the lower part of the upper Maastrichtian (Pui Beds; [88]) and, respectively, in the Transylvanian Basin in the upper part of the lower Maastrichtian at Oarda de Jos [60,89]. Accordingly, if this identification is upheld by future, more detailed investigations, the K2 occurrence represents the oldest record of this squamate family in Haţeg Island.

The pterosaur remains recovered at K2 are too poorly preserved and/or relatively non-diagnostic (Fig 8K) as to allow detailed assessment of their significance for reconstructing local faunal evolution. Overall, the pterosaur fossil record itself is also rather poor across the Transylvanian landmass, although the group (represented solely by azhdarchoids, according to our current knowledge) appears to have been present throughout the existence of the Haţeg Island faunas, both in time and across territory (e.g.,[3,90,110,223]). We note, nonetheless, that the fossils from site K2 appear to indicate a small-moderate sized individual, whereas previously only very large azhdarchoids were reported from the wider Vălioara area (e.g., [10]), which suggests that the group may had been more diverse in this particular area and time-slice than previously acknowledged.

Finally, mammals are represented at K2 by a very meagre fossil record, amounting to just one isolated tooth of a kogaionid multituberculate (Fig 8L-M); nonetheless, it adds to the growing list of multituberculate-bearing sites documented throughout the Transylvanian Landmass, and especially in Haţeg Basin [19]. More importantly, it also accounts as one of the oldest kogaionid occurrences within Haţeg Basin, only marginally younger than that from site K1, and roughly coeval with that from site K3 – both reported previously by [11] from the same west-Vălioara area; kogaionids are otherwise known to occur up-section up to some of the youngest known fossiliferous sites, including within the Densuş-Ciula Formation [19]. Together, all these early occurrences from western Vălioara indicate the presence of multituberculates in the Haţeg area as early as the Campanian/Maastrichtian boundary, although still slightly postdating their earliest occurrence documented in the southwestern Transylvanian area, by the later part of the late Campanian [19,96,224,231].

Furthermore, this specimen, despite being only an isolated P1, is also remarkable in that its dimensions (about 1.3 mm long) suggest the presence of a small-sized kogaionid individual (and taxon, see above, Section 4.4), according to the methodology of rough body-size estimation used by [19]. Given its very basal position within the Densuş-Ciula Formation (and the uppermost Cretaceous of Haţeg Basin overall), its relatively small size conforms to the body size-related stratigraphic distribution pattern identified previously by [19], adding further support to the scenario according to which small kogaionids precede their larger-bodied relatives in the Haţeg Island faunal assemblages. Whether the K2 kogaionid belongs to one or the other previously reported taxa from the Transylvanian area or represents a previously unknown one, and thus its significance to tracking taxon-specific spatio-temporal distribution patterns (e.g., [19,138]), remains to be established once more abundant and diagnostic kogaionid material may become available from site K2 and/or other roughly time-correlative sites from the basal part of Unit 1.5 (grey Vălioara beds) of [20].

In summary, based on its currently recognized faunal composition, site K2 – as the oldest high-diversity uppermost Cretaceous vertebrate site of Haţeg Basin – points to a remarkable large-scale faunal stability in these parts of Haţeg Island during the Maastrichtian. According to our data, the dominant elements of the Haţeg Basin paleofaunas were already present here by the earliest Maastrichtian, and no significant differences in overall faunal composition can be detected between the oldest and the youngest vertebrate assemblages of Haţeg Basin, at least not at the level of higher taxa. Furthermore, it appears that not only the overall faunal composition, but also the dominance/relative abundance of the different taxa remained largely similar, as the faunal assemblage of site K2 is almost identical to those of many younger multitaxic sites from Haţeg Basin in that: 1) the assemblage is dominated by remains of terrestrial animals; 2) the most frequently encountered elements belong to herbivorous dinosaurs and *Kallokibotion* turtles; 3) the dinosaur material is dominated by rhabdodontid and titanosaur remains; 4) all three commonly found dinosaur taxa (rhabdodontids, titanosaurs and hadrosauroids) are represented in the assemblage; 5) ankylosaurs are either absent or extremely rare; and 6) a high-diversity crocodilyform fauna is present, including four distinct taxonomic/eco-morphologic groups (*Doratodon*-like

ziphodont predators, *Theriosuchus*-like terrestrial forms, larger-sized *Allodaposuchus*-like predators, and durophagous *Acynodon*-like forms), all of which are detected at site K2 as well.

Based on the available data, there is no evidence that any major clade disappeared during the Maastrichtian, as all of these are recorded continuously from the earliest to the late Maastrichtian; nevertheless, some regional differences across Haţeg Island may have existed in the stratigraphic distribution of certain taxa such as the titanosaurs or the ankylosaurs (e.g., [223,224,236]). This overall faunal conservatism suggests that there were no higher-level clade replacements or disappearances on Haţeg Island such as those reported in the western Ibero-Armorican Landmass for the same time interval (the Maastrichtian Dinosaur Turnover of [243–246]), and that the decline of the typical latest Cretaceous continental faunas in this particular region did not happen gradually, prolonged over millions of years before the Cretaceous-Paleogene boundary, at least when considered at clade level.

Meanwhile, it appears that this clade-level faunal stability was nevertheless associated with lower-level (generic and/or species-level) changes both stratigraphically and geographically, as was already hinted at by previous results of [5] for titanosaurs, by [124] and [12] for rhabdodontids, by [127] for hadrosauroids, and by [19] for kogaionids. All these previous studies have underscored the critical importance of well-preserved, associated and diagnostic material in recognizing and tracking such lower-level taxonomic differences. In this respect, the ongoing detailed taxonomic investigation of the associated and/or articulated skeletal parts of different dinosaurs such as titanosaurs and rhabdodontids (see [12]) collected from site K2 may shed further light on the presence (or absence) of such previously hidden changes in faunal diversity across Haţeg Island, both through time and through space.

## 5.4. Paleoecological significance of the K2 vertebrate assemblage

As discussed in the previous sections, one of the most outstanding characteristics of site K2 is represented by its sedimentary setting, reconstructed in detail by thorough sedimentological investigations, supplanted by the examination of the local mollusc and palynomorph assemblages, as well as by palynofacial and organic geochemistry observations. The sedimentary sequence that forms the fossiliferous part of the investigated Pârâul Neagului succession is represented by the deposits of a relatively short-lived and spatially restricted, but still perennial lake, a sedimentary environment yet to be convincingly identified and documented in other parts of Haţeg Basin (see below) despite these deposits often being cited loosely as 'fluvio-lacustrine' (e.g., [31,247]) or even as entirely 'lacustrine' in some areas [30]. Meanwhile, the interbedded main fossil-bearing unit at site K2 in particular corresponds to a small lacustrine deltas built at a fluvial discharge point into the lake during and after more or less intense peak flood periods – another particular sedimentary environment that is at present only recognized and reconstructed at this site across the entire former Haţeg Island – making site K2 overall a unique fossiliferous occurrence in the uppermost Cretaceous of the Transylvanian area.

Indeed, of the relatively large number of previously identified uppermost Cretaceous fossil sites from the area of Haţeg Island (e.g., [3,4,19,20,94,109,110,223,231]), only a select few have been investigated in more detail and reported as to their specific depositional environments and/or taphonomic characteristics in addition to their detailed fossil content (e.g., [8,15,90,94,240,248,249]). From the list of the sites surveyed in more details, several had been identified as being deposited in standing or at most very slowly flowing water bodies, instead of being formed in either purely fluvial (e.g., [240]) or emerged floodplain (e.g., [8]) environments. These include sites interpreted as representing swampy ponds/oxbow lakes at Pui (Pui-Swamp MvBB; [3,95,250]; and Pui Deposit; [94,248]); ponds set in a closed-vegetation setting at Vălioara (Budurone MvBB; [15]), both of these localities in Haţeg Basin; as well as in abandoned channels (ODAN MvBB; [90,248,251]) and probably oxbow lakes (ODA Stocks; [249]) at Oarda de Jos, in the Transylvanian Basin. In neither of these cases, however, did sediment deposition and fossil accumulation resulted from the interaction between a larger and more perennial standing water body and a river discharging into it, as it is reconstructed here for site K2 (Fig 17). Under these circumstances, it is thus to be expected that the particular sedimentological context of site K2 also impacted in unique ways the taphonomic pathways leading to, and thus the taphonomic history, of the K2 fossil vertebrate

 

assemblage, compared to these other 'standing-water' fossil sites, by producing a distinctive taphonomic signature and, correspondingly, introducing special biases that affect its composition. The differences resulting from these distinct sedimentary settings (as well as potential similarities), with implications for the paleoecological and paleoenvironmental significance of site K2, will be noted and discussed in some detail below.

Firstly, although the K2 assemblage can be characterized as a mixed, micro- and macro-vertebrate bonebed, it, and especially its taxonomically identifiable sub-sample, is dominated by macrovertebrate remains, especially when not the absolute size of the recovered fossils themselves but that of the organisms (original body weight larger than 5 kg; [41]) the remains originate from is considered (Figs 12 and 14C). This pattern of dominance is counter-intuitive, as small-sized taxa normally represent the more abundant and diverse component of any (paleo)ecosystems (e.g., [252]), and thus must be the result of biases connected to the particular taphonomic history of site K2 (see also more comments below).

Furthermore, given the definitively subaqueous, lacustrine depositional environment inferred for site K2 (Fig 17), the rarity (both in abundance and in diversity) of semi-aquatic and, especially, aquatic taxa is striking (Fig 12). For one, fish remains are extremely rare in the K2 material, with only four ganoid lepisosteiform scales recovered that indicate the presence of these purely aquatic organisms at this site, despite the sedimentological investigations suggesting accumulation within a lake at its confluence with a flowing river course, an environment that would provide optimal conditions for thriving fish communities. Nonetheless, both potentially autochthonous (living in the lake environment itself) and allochthonous (living in the river channel) fish remains are almost completely absent.

It should be noted, nevertheless, that fish remains are generally very rare to completely absent in the different uppermost Cretaceous vertebrate fossil sites of Haţeg Basin [4,17] as well as the Transylvanian area in general (e.g., [3,242,231]), in stark contrast with the fact that many of these (including site K2) preserve sediments deposited in aquatic environments where a greater presence of fish remains would be expected. The rarity of fish fossils seems nonetheless to be real in most of these sites, as taphonomic or preservational factors cannot easily explain the presence of often large numbers of fragile but well preserved cranial and postcranial amphibian remains in these accumulations alongside the much rarer, albeit otherwise very resistant ganoid scales or isolated fish teeth, if these latter remains are present at all (e.g., [3,17]). The extreme rarity or even absence of fish remains is clearly indicative of some paleoenvironmental and/or paleoecological factor at work, not only at site K2 but in the entire Haţeg Basin area.

It might be assumed that the local fluvial systems developing on the edges of debris cones, as was reconstructed for the Vălioara area by [20], not to mention possible small lakes within the same setting, may have been too ephemeral, short-lived and/or highly localised to allow significant colonization by fish faunas. However, even this scenario is largely undermined by the facts that: 1) it is well known that fishes colonise all wetlands easily and quickly [253]; and 2) all these sites, including K2 as well, are known to contain predominantly or exclusively remains of predatory, piscivorous fish such as lepisosteiforms (including *Lepisosteus* sp. and *Atractosteus* sp.; e.g., [15,17,94]), which in itself indicates that a more diverse fish fauna must have existed in the area in order to complement the local aquatic food web and support the presence of gars.

Remarkably, a partially different distributional pattern of fish remains can be noted in the southwestern Transylvanian Basin. Although here, too, fish remains are often very rare to absent in the different fossil accumulations (e.g., [231,242]), at least in three cases (PT L0/c MvBB from Petreşti, respectively ODA Stocks and ODAN MvBBs from Oarda de Jos) these are not only present but even well represented [96,231,242,249]. While the depositional setting of site PT L0/c is not yet documented in detail, the Oarda de Jos sites both represent standing-water deposition, roughly comparable to, and possibly even less perennial and spatially extensive than, that identified for site K2. It thus appears that the general rarity of uppermost Cretaceous fish remains in Haţeg Basin has less to do with the type, size or permanence of the aquatic setting, but rather with some type of paleogeographic exclusion, with the more lowland environments of the Transylvanian Basin being somewhat more favourable to their presence and distribution compared to the more inland, intermountain settings that characterized Haţeg Basin. In conclusion, although the current data do not allow a definitive answer in this

regard, the most plausible explanation for the extreme rarity of fishes in the case of site K2 seems to be some combination between factors such as the short-lived lake environment, an original, paleogeographically selective distribution, as well as a particular local taphonomic history (i.e., selectivity in the transport and accumulation of organic remains by the river flowing into the lake; see more details regarding this aspect, below).

Notably, amphibian fossils are also extremely rare in the K2 vertebrate assemblage, although, admittedly, the small number of these fossils may be due in part to the somewhat limited extent of the screen-washing done at the site. It should be pointed out, nevertheless, that processing of roughly comparable quantities of fossiliferous matrix from other uppermost Cretaceous localities/sites has usually yielded relatively high numbers (and especially high percentages) of amphibian fossils when compared, e.g., with those of crocodyliforms that are generally represented by isolated teeth recovered through similar micropaleontological techniques; these numbers indicate that both frogs and albanerpetontids were common elements of the Haţeg paleofauna(s) (e.g., [3,15,17,18,88,90,231,242,248,254]). As a side note in this respect, it should be emphasized that while albanerpetontids appear to have favoured upland settings and karstic environments, albeit still moist and vegetated ones, during Cenozoic times [255], in the Mesozoic they are very often associated with definitively aquatic deposits accumulated in well-vegetated swampy areas (e.g., [85,256]), and their remains occur occasionally even in wetland-linked lacustrine deposits (e.g., [257]) – the type of environment reconstructed here for site K2 and its surroundings. Accordingly, although albanerpetontid remains are rather widely distributed in the uppermost Cretaceous deposits of the Transylvanian area, having been found in sites representing a wide range of paleoenvironments (e.g., [17]), they probably preferentially inhabited precisely those habitats that were widely available in the ambient paleo-landscape of site K2.

Furthermore, amphibian remains are always present and most often abundant in the different local faunal assemblages that contain microvertebrates, regardless whether fish are present or not, or else whether these assemblages derive from deposits representing more poorly drained or better-drained environments (e.g., [17]). This pattern suggests that their remains were not subjected to the same (paleoenvironmental and/or selective geographic distribution-driven) biases that induced the commonly observed pattern of quasi-general rarity of fish remains (see above). Considering all these circumstances, as well as that the fossiliferous sediments excavated at K2 were clearly accumulated in sub-aquatic environments, it is even more striking to see the extremely low values of NISP for these two groups of aquatic/semi-aquatic taxa (Table 3). In this regard, the K2 assemblage differs radically from those recovered from other sites representing similar/comparable depositional environments, at Pui [94,95,248], Oarda de Jos [90,248,250], or even in other sites from the Vălioara area [15,17,250].

Interestingly, the same patterns noted above regarding the representation of aquatic-semiaquatic taxa are replicated to an extent in the case of the turtles as well. Remains of *Kallokibotion* sp. (mostly as shell fragments, but also in the form of an incomplete, but articulated shell) are among the most abundant remains in the K2 assemblage, while those belonging to a probable dortokid are far less common (Table 3). Given the sub-aquatic sedimentary environments reconstructed for K2, such a relative abundance of the two turtle fossil samples is somewhat counterintuitive, as a (semi)terrestrial lifestyle was suggested for *Kallokibotion* by several authors based on their anatomy (e.g., highly domed carapace, dorsoventrally expanded skull, curved femur and humerus; e.g., [86]) and taphonomical preservation mode, with several examples of (par)autochthonous articulated shell material discovered in definitively terrestrial, more drier floodplain environments (e.g., [8,240]), while dortokids (currently only represented by *Dortoka vremiri* in the uppermost Cretaceous of Romania) are assessed to have had a more aquatic lifestyle (see details in [86,240]).

It is true that similar distribution patterns are documented in the entire Haţeg Basin, and throughout the Transylvanian area, as most sites are dominated by (or at least yielded large samples of) fossil remains belonging to *Kallokibotion* (e.g., [3,4,86]), while remains of other turtle taxa (such as the dortokids) are usually very rare (e.g., [86,240,258]); most probably this overarching relative abundance pattern does indicate a latest Cretaceous Transylvanian turtle fauna dominated by *Kallokibotion*, with dortokids only as minor components. Thus, it is conceivable that the abundance of *Kallokibotion*

remains that was also detected in the K2 material would only conform to this generalized pattern, although it also indicates that *Kallokibotion* came to dominate turtle assemblages inhabiting the fluvio-lacustrine settings of Haţeg Basin already from the inception of the Haţeg Island faunas.

Nonetheless, despite this overarching pattern of *Kallokibotion* dominance, dortokids still appear to be somewhat better represented in certain local assemblages. At Petreşti (southwestern Transylvanian Basin), in a transitional-to-continental sequence lying at the base of the purely continental uppermost Cretaceous succession of Haţeg Island, dortokids are reported to be present, whereas no *Kallokibotion* remains had been identified thus far [223]. More generally, dortokids were apparently more common, or at least their remains occur with higher frequency and are more widespread, in the southwestern Transylvanian Basin (e.g., [86,110,231,258–260]) compared to their fossil record in the Haţeg and Rusca Montană basins (e.g., [86,101,240,260]). Last, but not least, it is worth emphasizing that at least some of the already discussed 'standing-water' fossil accumulations contain dominantly or even exclusively dortokid remains, be those from Haţeg Basin (Pui-Depozit; [94,248]) or from the southwestern Transylvanian Basin (ODAN; [90,248]; ODA Stocks; [251]) – an otherwise rare pattern of relative turtle abundance across the former Haţeg Island. In this respect, the very low value of the dortokid-to-*Kallokibotion* ratio at K2 does stand out, and further underscores the observed general rarity of the aquatic-semiaquatic organisms in the local assemblage.

Although the K2 crocodyliform assemblage is largely similar in its higher-level taxonomic composition to those reported in most other multitaxic accumulations across Haţeg Island, it also displays a more peculiar feature – the relatively important presence of teeth assigned to the *Acynodon* morphotype, despite these forming the least numerous sub-set of the local crocodyliform sample. Indeed, *Acynodon* is probably the least commonly occurring crocodyliform tooth morphotype category across the entire Transylvanian Landmass, not encountered at all in several important (micro)vertebrate accumulations such as the Budurone MvBB [15] in Haţeg Basin, or the Fărcădeana MvBB [17] in Rusca Montană Basin. Meanwhile, it was reported to occur in several of the known (or potentially) 'standing-water' assemblages discussed before, such as those from Pui-Swamp [95] and Pui-Depozit [94], as well as from the Fântânele MvBB [97] and a second, closely spaced and smaller MvBB, Fântânele 2 [121] near Vălioara, being especially well represented in the ODAN MvBB at Oarda de Jos [90].

The specialized durophagous *Acynodon*-type basal eusuchians are usually interpreted as semi-aquatic forms feeding on hard-shelled food such as molluscs and arthropods in heavily vegetated and turbid waters (e.g., [121]), as it was also suggested for their Haţeg representatives by [97]. Such a lifestyle and feeding habits are in line with the presence of generally wetland-type paleoenvironments reconstructed for the sedimentary subunit containing site K2 [20], as well as specifically for the site itself (this study). Together with *Allodaposuchus*, a larger-bodied semi-aquatic generalist predator [97,105], and also the most commonly encountered crocodyliform on Haţeg Island (e.g., [26,90,101,112,133]), crocodyliforms ecologically related to aquatic environments thus make up about 50% of the K2 crocodyliform sample (Table 3). This figure is, however, notably less than what is usually identified in different Transylvanian uppermost Cretaceous fossil vertebrate assemblages (and especially the well-sampled and specimen-rich ones), in which semi-aquatic crocodyliforms often dominate by far, representing at least 75–80% of the local crocodyliform sample (e.g., [17,90]). The case of crocodyliforms thus mirrors, and further reinforces, the overall conclusion derived from the survey of the other aquatic and semi-aquatic taxa at K2 – that most aquatic and semi-aquatic taxa are overall poorly represented at this site. Nonetheless, the K2 crocodyliform sample stands out among currently known Haţeg Island vertebrate assemblages in the rather unusual combination between an overall relatively low abundance of semi-aquatic taxa combined with the presence (and even relatively good percentage representation) of the small-sized semi-aquatic *Acynodon*.

The resulting pattern of low number/abundance of the remains of aquatic and semi-aquatic organisms stands in stark contrast with that recorded in the case of the terrestrial ones, the remains of which (in terms of both NISP and MNI) dominate the K2 local assemblage (Table 3; Fig 12). Furthermore, it is noteworthy that the associated (even articulated) preservation mode is recorded at K2 only for rhabdodontid and titanosaur dinosaurs as well as for *Kallokibotion*, all definitively terrestrial organisms. The case of turtles, represented at K2 by both semi-aquatic and terrestrial taxa (see above)

with roughly similar anatomy and preservation potential, is especially revealing. The preservation mode (including articulated shells) and the high frequency of *Kallokibotion* remains suggest that these were accumulated under similar conditions with the fossils of the purely terrestrial dinosaurs that show comparable taphonomical characteristics (see above), while the small absolute dimensions, rarity (about 1% of the NISP), and isolated nature of the dortokid remains are all reminiscent of the taphonomic attributes displayed by the remains of other aquatic or semi-aquatic taxa. Also telling in this respect is the relatively high abundance values shown by the remains of crocodyliform taxa that are assessed to have been (mostly or entirely) terrestrial – the small-bodied predatory *Doratodon*, respectively the similarly sized omnivorous '*Theriosuchus*-type'form (e.g., [97,99,261]) –, despite the fact that these are usually less well represented than their semi-aquatic relatives in the different Haţeg Island fossil occurrences (e.g., [17,90]).

These peculiar, and remarkable, compositional and preservational features of the K2 assemblage, especially when contrasted with other assemblages recovered from sites that originated in largely similar 'standing-water' depositional environments (such as those from Pui, Oarda de Jos or Vălioara, mentioned in the previous comparisons), are noteworthy and require an explanation. We consider it highly likely that this unique set of taphonomical characteristics is linked to, and is a by-product of, the specific depositional setting and taphonomic history of site K2 which are as yet similarly unique in Haţeg Basin, and the Transylvanian area overall: a small lacustrine delta built by rivers discharging their load into the floodplain lake especially during peak flood events, alongside with the organic remains they swept together from across the floodplain and transported as bedload and/or floating carcasses (see above, 5.2). Given this particular scenario of sediment (and fossil) accumulation, is can be assumed that:

(1) the aquatic ecosystems (especially the hosting lacustrine one) were sampled at site K2 in an attritional manner, with the remnants of the autochthonous aquatic and semi-aquatic organisms being buried after relatively prolonged periods of within-environment taphonomic processing (hence their exclusively isolated nature, high degree of fragmentation, and general rarity, keeping pace with the natural rate of skeletal attrition), whereas

(2) the ecosystems of the surrounding terrestrial, albeit probably still wetland-type (see [20]) habitats were sampled in a more complex manner by the fluvial network, bringing together both remains of organisms that were lying around on the floodplain or even reached the channel itself, but resulted from normal, attritional mortality as well, and underwent previous taphonomic processing of variable, but usually rather intense, degrees, alongside those of organisms that succumbed close to, or even during, the peaking flood periods and some of which were still in the very early stages of their taphonomic history.

Under such circumstances, the remains belonging to the former 'terrestrial' category are expected to display taphonomic features similar to those from the strictly autochthonous aquatic ecosystems (except for the specific results produced by distinct subaqueously, respectively subaerially active taphonomic processes), such as high degrees of dissociation and fragmentation, resulting in generally small dimensions of the remains and lack of their skeletal association, let alone articulation, as well as more advanced stages of taphonomic processing (weathering, abrasion etc.). Meanwhile, carcasses that were becoming available shortly before the flood event (or maybe even were connected to this event; see above, 5.2) would undergo fluvial transport into the site of final burial as still articulated 'floating carcasses', thus producing sets of associated (potentially even articulated) remains with significantly lower degrees of taphonomic imprinting. Furthermore, given the good preservation state and high degree of element representation noticed within these incomplete skeletons (Table 5), combined with the presence of still articulated or quasi-articulated skeletal parts, it appears likely that this fluvial transport was neither intensive nor long-distance, suggesting that both rhabdodontids and titanosaurs were inhabitants of the paleoenvironments lining closely the lake margins and river banks.

This multi-pathway genetic scenario also accounts for the previously discussed and seemingly counterintuitive combination between (1) a low frequency of poorly preserved aquatic and sub-aquatic organism remains as observed at site K2 (coupled with a similarly low frequency of that of certain terrestrial organisms such as lizards, some dinosaurs, and

multituberculates) – all of these being the results of time-averaged, long-term attritional mortality, and (2) a remarkably high frequency of better-preserved and more complete remains of other terrestrial taxa, the results of, if not a strictly catastrophic, at the least a short-term and higher-intensity mortality period brought by the stressful flooding season, combined with an inherently significantly shorter time-averaging (see further comments below) and the increased intensity of the concentrating agent activity. That the intensity of mortality and/or carcass accumulation was probably highly dependent on the gravity of pluvial extremes during the potential wet season(s) is further suggested by the fact that other lacustrine deltaic intercalations identified at site K2 but which indicate less severe flooding episodes (flood beds 2 and 3, see above, Sedimentology) are completely devoid of associated fossil material representing terrestrial vertebrates (Fig 3B and S3 File). Furthermore, these other intercalations contain only remains of gastropods, while are also devoid of the remains of aquatic vertebrates (S3 Files), showing that the rate of normal, attritional vertebrate material production within the aquatic ecosystem(s) reaching the burial stage was low enough as to not allow preservation of any significant amount of such remains in the time interval elapsed between the subsequent flooding events.

Turning to the terrestrial subsample of the K2 assemblage, it is worth pointing out that at first glance, the dominance of the titanosaur and rhabdodontid remains in this assemblage could also be consistent with, and may appear to simply reflect, their common nature and high abundance reported for the entire Haţeg Basin by [4], with a similar pattern also being suggested to generally hold for other parts of the Transylvanian area, both by data available for the southwestern Transylvanian Basin (e.g., [90,109,110,231]), and even by the more meagre fossil record currently known from Rusca Montană Basin [17,101]. Indeed, these two dinosaur groups are almost always reported to yield the most frequently encountered dinosaurian skeletal elements, compared to those of the hadrosauroids and theropods, not to mention the extremely rare ankylosaurs.

Nevertheless, local exceptions to this overarching abundance patterns were also reported, including some where one or another of these dinosaur clades is either extremely under-represented or completely absent, such as in the case of the hadrosauroids and titanosaurs in the basal part of the Sebeş Formation at Petreşti [223,224,234], or, conversely, is over-represented, such as the struthiosaurids at Vurpăr, also in the basal Sebeş Formation (e.g., [109,110,238]). In both of these instances, such outlier situations were explained by details of local faunal evolution (i.e., later arrival of certain taxa) and/or differential geographic distribution (e.g., [223–225,234,236]). However, another case of divergence from this generalized pattern of rhabdodontid-titanosaur dominance is more intriguing, as it concerns a fossil locality – the Tuştea-Oltoane nesting site (e.g., [6,8,13,]) – that also belongs to the Densuş-Ciula Formation, albeit positioned at a higher stratigraphic level within the Middle Member of this unit compared to site K2 (e.g., [3]), and representing a markedly different paleoenvironmental setting (well-drained and pedogenetically modified distal floodplains). The vertebrate assemblage of Tuştea is dominated by hadrosauroid remains together with, and seconding those, of the rhabdodontids, whereas the titanosaur material accounts for only 3% of the NISP [8]. The unusually high frequency of hadrosauroid remains at Tuştea was explained by two interrelated factors, one reflecting the habitat preference and selectivity of the Transylvanian hadrosauroids for relatively drier, better drained floodplains, as being more favourable for their most probably group nesting behaviour, while the other, the occurrence of a relatively high number of hadrosauroid hatchling remains at Tuştea, a result of their reproductive behaviour pattern that seemingly included a combination between recurrent colonial nesting and the phylopatry (i.e., tendency to remain in the place of birth) of their altricial early hatchlings in this apparently favourable environmental setting (e.g., [8]).

Given the noted large-scale pattern of abundance ranking between the different dinosaurian megaherbivores – i.e., generally rarer hadrosauroid remains, compared to rhabdodontids and titanosaurs -, it is not unexpected to find that the latter are more abundant than hadrosauroids at site K2 as well, as a simple local reflection of this pattern. However, our sedimentological and taphonomical survey at this site suggests that this is not necessarily the case, and contributes new data regarding the possible habitat preference(s) and paleoenvironmental segregation of the hadrosauroids. Indeed, despite the fact that in their basin-wide survey regarding the distribution of the different dinosaur remains, [4] failed to

document significant differences between the large-scale habitat preferences of the major megaherbivores of Hațeg Island (i.e., rhabdodontids, titanosaurs and hadrosauroids), at K2 we noted important preservational and taphonomical differences between the specimen sub-samples representing the three taxa. Whereas the remains of the rhabdodontids and titanosaurs share features such as high abundance, a high degree of completeness and association of their remains, representing different parts of a number of incomplete skeletons, and displaying very low to at most moderate degrees of taphonomical modifications, those of the hadrosauroids are isolated, significantly rarer (representing only about 1% of the NISP) and small-sized, show advanced degrees of fragmentation, and are mainly restricted to very resistant isolated teeth. The hadrosauroid fossils thus display features that characterize the subsample of terrestrially derived but attritionally produced vertebrate remains from K2, brought by fluvial action from the neighbouring areas into the final depositional setting already as individual elements that underwent a significant amount of taphonomical processing before transport and final burial (see above).

In accordance with the previously presented multi-pathway scenario for the genesis of the K2 thanatocoenosis (see 5.2), such a taphonomical distinction between the rhabdodontids and titanosaurs, on one part, and the hadrosauroids, on the other, would suggest either that (1) hadrosauroids were members of more distal paleobiocoenoses with respect to the lake margins/river banks, instead of being strictly sympatric there with the rhabdodontids/titanosaurs, and their already scattered isolated remains were collected in more remote areas by the floodwaters as they swept across the floodplain, or that (2) hadrosauroids, although sharing the same lake-margin paleoenvironments with the rhabdodontids/titanosaurs, were less exposed to the perilous effects of the more important flooding events (such as those that produced the main K2 vertebrate accumulation). In the latter case, the lower susceptibility to damage of the hadrosauroids may be due to some particular paleobiological characteristics that improved their ability to cope with such challenges (e.g., better floating or swimming capacities), or else they were temporarily absent from these more humid wetland environments during peak flood periods (e.g., seasonal migration habits, either those of hadrosauroids out of the proximal wetland environments, or else those of the rhabdodontids and titanosaurs into these wetland settings, during the rainy season). Unfortunately, the currently existing observational data are not goodenough to definitively decide between these competing hypotheses, and further carefully excavated and studied sites are required to substantiate support for one or the other. New discoveries, currently under study, indeed show that hadrosauroids definitively did not avoided completely the less well drained floodplain settings (e.g., [127]), although the paleoenvironments discussed by these authors as hosting hadrosauroid remains were apparently less water-logged than those reconstructed for the surroundings of site K2 ([20]; this study).

Nevertheless, the definite abundance rank differences detected in the case of dinosaurian megaherbivores between site K2 and Tuștea locality might, however, lend some support to the idea that titanosaurs and possibly also rhabdodontids preferred (albeit without being exclusively restricted to) the more proximal wetland environments (as those reconstructed for and around site K2), while the hadrosauroids favoured, at least occasionally, more distal, better drained floodplain settings (such as the depositional environment identified at Tuștea locality), regardless whether the underlying cause for this was specific nesting behaviour, genuine habitat segregation, higher taxon-specific mobility during everyday activity, and/or seasonal migrations. We note, nonetheless, that any potential migratory behaviour, whose existence is currently supported by stable isotope geochemistry at least for certain hadrosauroids (e.g., [262]) and sauropods (e.g., [263]), travelling must have been rather short-distance, as previous stable isotope analyses excluded the occurrence of large-scale migratory movements for the two Hațeg Basin ornithopod groups (rhabdodontids and hadrosauroids) [34]. Meanwhile, the same analyses also suggested the absence of large-scale habitat partitioning between these two groups of herbivores, indicating that any habitat segregation involving rhabdodontids and hadrosauroids, if existed at all, must have been at a very small spatial scale.

The theropod subsample of the K2 assemblage differs by the exactly same preservational and taphonomic characteristics from the subsamples corresponding to the rhabdodontids and, respectively, titanosaurs, as does the hadrosauroid one, i.e., small sample size (low NISP – about 3%; Fig 12), exclusively isolated remains with high degree of

fragmentation, and represented mainly by isolated teeth; such coherent differences might be invoked to suggest similar paleoecological implications – i.e., that theropods may have been members of more distal paleocommunities. Nevertheless, we caution against such a simplistic interpretation of the taphonomic data, as the relatively robust and resistant skeletal remains of the larger-bodied hadrosauroids most probably had markedly different preservational responses to the different taphonomic processes operating at site K2 than the more gracile and hollowed-out, pneumatic skeletal elements of the smaller-sized theropods.

On the other hand, a general taphonomical equivalence most probably does exist between the theropod and crocodyliform subsamples, as these are both made up almost exclusively by isolated teeth of roughly similar sizes and overall shapes, allowing for comparable biases affecting them; thus, the differences noted between these subsamples probably do carry paleoecological significance. In this respect, it is worth noting that the crocodyliform teeth preserved at site K2, even if restricted only to the assumedly more terrestrial *Doratodon* and '*Theriosuchus*-type' morphologies, are significantly more numerous than those of the theropods, suggesting that the former may have been more common members of the neighbouring terrestrial ecosystems than the latter. Furthermore, given the terrestrial predator role generally accepted for the ziphodont *Doratodon* (e.g., [97,99,105]), its significantly higher abundance at K2 indicates that it probably was the common predator in these wetland ecosystems instead of the small theropods. On the other hand, among the local predators the small theropods appear to have been more taxonomically diverse (although, admittedly, taxonomic diversity is difficult to assess in poorly known heterodont crocodyliforms such as *Doratodon* that are represented only by isolated teeth in the Transylvanian area, with two different taxa represented. Intriguingly, one of these – *Richardoestesia* – was sometimes considered to be piscivorous on the account of its peculiar dental morphology (e.g., [103]), a feeding habit probably well suited for the wetland environments reconstructed for the neighbourhood of site K2 ([20]; this study).

At last, it should be noted that the absence, thus far, of ankylosaurians from site K2 may not bear any particular paleoenvironmental/paleoecological significance such as habitat partitioning among the dinosaur taxa, as was proposed, e.g., by [25] and [26]. Instead, it most probably conforms to the wider pattern of paleogeographic distribution of these animals across Haţeg Island, as they appear to have colonized first the lowland-type settings of the southwestern Transylvanian Basin, before making their appearance in the more upland, intermountain areas of Haţeg Basin itself and turning up in localities/sites that are somewhat younger than site K2 [236].

To conclude, the paleontological and taphonomical investigation of the vertebrate material from site K2 indicates that the recovered assemblage contains mixed remains that may represent as many as three different paleoenvironmental settings and their respective paleocommunities: 1) a local, aquatic to semi-aquatic community connected directly to the lake body and maybe the rivers discharging into it; 2) a proximal terrestrial community that inhabited the nearshore, richly vegetated wetlands; and 3) possibly a more distal terrestrial one, albeit probably still representing the wetland-type environments widespread in the sedimentary subunit hosting site K2. Superposition of the different taphonomic histories sampling these different paleoenvironments, as reconstructed for the K2 fossil vertebrate assemblage, produced one of its most outstanding features – the clear-cut dominance of terrestrial organisms in the assemblage over the aquatic-semi-aquatic ones, both in terms of absolute abundance, as well as in those of skeletal and individual representation, despite the sub-aquatic environment of sediment (and fossil) accumulation that existed at this site. In this regard, the K2 assemblage departs sharply in its overall make-up and taphonomic characteristics from previously reported vertebrate assemblages, from Haţeg Basin as well as from other parts of the latest Cretaceous Transylvanian Landmass, even from those that derive from largely comparable depositional settings dominated by 'standing water' sediment accumulation such as floodplain ponds, oxbow lakes and small swamps. Indeed, in virtually all of these latter sites the remains of semi-aquatic to aquatic taxa are well to clearly over-represented, whereas those of the terrestrial vertebrates, if present, are rare, mainly isolated and often fragmentary (especially in the case of the larger-bodied taxa), and frequently display other signs of advanced pre-burial taphonomical processing (abrasion, weathering, bioerosion etc.) – in clear contrast with the preservational features identified at site K2.

We interpret the dominant presence of the associated, even articulated, and relatively well-preserved remains of three definitively terrestrial taxa (the turtle *Kallokibotion*, respectively the rhabdodontid and titanosaur megaherbivores) at site K2 as the most probable result of their introduction into the site following a relatively short-distance transport of still partly complete animals ('floating carcasses') during peak flood events. On its turn, we regard this scenario as a reliable indication that these organisms were members of the stable proximal wetland paleocommunity, present there even during periods of increased environmental stress represented by the flooding events, and that their demise occurred shortly before, or even during, the flood itself.

Other terrestrial taxa such as lizards or multituberculates are too poorly represented in the K2 assemblage to allow any assessment as to their membership in the proximal and/or the distal terrestrial communities. Nevertheless, it should be noted that whereas the extremely well preserved, albeit otherwise fragile, cranial remains of barbateiid lizards at K2 would indicate a short-term transport into the site (thus, their probable presence in the proximal wetlands), a larger survey concerning the distribution and preservational characteristics of multituberculate fossil remains in the Transylvanian area by [19] suggested that most local multituberculates seem to have favoured better-drained floodplain habitats. However, the dental and body-size diversity patterns recorded within this group across Haţeg Island (e.g., [19,138]) point to a relatively high degree of anatomical (thus, probably also ecological) disparity within this group, making it probable that some of these omnivorous-to-partly herbivorous animals (see [264]) might have ventured at least opportunistically, for foraging and/or shelter, into the densely vegetated and resource-rich proximal wetlands. Indeed, several vertebrate assemblages discovered in roughly comparable poorly drained, 'standing water' depositional settings have yielded decent to good samples of multituberculate remains, represented – just as in site K2 – mainly by isolated teeth (e.g., [19,146,251]), suggesting that certain multituberculates thrived (or at least were active) in the proximity of these water-logged environments as well.

Finally, we remark that the composition of the local, aquatic paleocommunity largely mirrors that of other fossiliferous localities reported throughout Haţeg Island, with frogs, albanerpetontids, dortokid turtles and semi-aquatic crocodyliforms as its main components, whereas fishes (represented primarily by gars) are strikingly rare. What stands out in this respect is the extreme rarity of the remains representing most of these aquatic taxa, not only the lepisosteids, with the notable exception of the large-sized predatory eusuchian *Allodaposuchus*, although the slightly higher abundance of the latter taxon was probably also strongly influenced by a combination between the highly attritional nature of the aquatic-semiaquatic assemblage (see above, 5.2) and the relatively high mechanical resistance of the crocodyliform remains (almost exclusively isolated teeth) to destructive taphonomic processes, compared to that of the elements representing several other taxa (especially the amphibians) from this assemblage. One last aspect worth emphasizing is the relatively high frequency (not necessarily in absolute terms, but in comparison with most other uppermost Cretaceous vertebrate accumulations from Haţeg Island) of the remains of the small-sized, semi-aquatic basal eusuchian *Acynodon* at this site – again, mirroring to a large extent its previously known occurrence patterns, its isolated teeth being reported before mainly from poorly drained floodplain environments. Such a high(er) frequency indicates that these wetland-linked water bodies, with their high primary productivity and important invertebrate biomass (as indicated by the presence of a large number and variety of gastropods in site K2) were favoured habitats of these durophagous, specialized semi-aquatic feeders.

## Conclusions

The uppermost Cretaceous Densuş-Ciula Formation of Haţeg Basin has been known for more than 100 years as an important dinosaur bone-bearing unit, but systematic excavations and detailed investigations (including sedimentology, taphonomy, malacology and palynology) of the discovered fossiliferous strata have been rare in the area, making it difficult to meaningfully evaluate and compare the local faunas of the different sites. In order to fill this gap, we report here the results of a multidisciplinary investigation of a rich (more than 800 specimens) and diverse vertebrate

material discovered from site K2 in the lower part of the Densuş-Ciula Formation. This investigation yielded the following results:

• The site K2 is currently the oldest well-documented and systematically excavated vertebrate accumulation in the entire Haţeg Basin, and one of the oldest ones across the entire Haţeg Island. Accordingly, the vertebrate assemblage recovered from this site can be considered as documenting the first stages in the assembly and evolution of the Haţeg Island faunas, and especially those from its more interior, upland areas, thus underscoring the essential contribution the thorough documentation of site K2 could make to future research into the faunal evolution of the latest Cretaceous Haţeg Island, in the wider context of the Late Cretaceous European Arhipelago.

• The sedimentological investigation pointed out that the local K2 depositional environment was represented by a small-scale lacustrine delta, where a high-diversity multixatic, multidominant mixed (macrofossil and microfossil) vertebrate thanatocoenosis was gathered due to a sudden drop in fluvial transport energy.

• According to the reconstructed taphonomical scenario leading to the accumulation of this fossil assemblage, it contains remains of a mixed origin that may represent as many as three different paleoenvironmental settings and their respective paleocommunities: 1) a local, aquatic to semi-aquatic community connected directly to the lake body and maybe to the rivers discharging into it; 2) a proximal terrestrial community that inhabited the nearshore, richly vegetated wetlands; and 3) possibly a more distal terrestrial one, albeit still representing the wetland-type environments widespread in the sedimentary subunit hosting site K2.

• The associated, even articulated, and relatively well-preserved remains of three large-sized and definitively terrestrial taxa (the turtle *Kallokibotion*, respectively the rhabdodontid and titanosaur megaherbivores) preserved at site K2 are the results of their introduction into the site following a relatively short-distance transport of still partly complete fresh skeletons ('floating carcasses') during peak flood events.

• The associated and articulated dinosaur assemblages discovered at site K2 is of exceptional importance for a better understanding of the taxonomic identity and affinities of the rhabdodontid and titanosaurian dinosaurs from Haţeg Island, because the documented skeletal parts represent some of the most complete rhabdodontid and sauropod skeletons collected from a single bonebed in Haţeg Basin, thus enabling the proper taxonomic identification of isolated remains and partial skeletons from other vertebrate sites.

• The vertebrate assemblage discovered at site K2 demonstrates that the faunas of Haţeg Island remained relatively stable on a large scale during the Maastrichtian, both in their higher-level taxonomic composition and in their ecological structuring (as exemplified by relative abundance ranking of the different taxa). The data presented herein indicate that the predominant elements of the Haţeg Basin paleofaunas were already present by the beginning of the Maastrichtian, at the latest, with no significant disparities in overall faunal composition being evident between the most ancient and the most recent latest Cretaceous local vertebrate assemblages, at least not at the level of higher taxa.

## Supporting information

**S1 File. Character list of taphonomic features recorded in the K2 vertebrate assemblage.**
(PDF)

**S1 Table. Character matrix of the taphonomic features recorded in the K2 vertebrate assemblage.**
(XLSX)

**S2 File. List and photos of the most important vertebrate taxa and specimens mentioned in the text.**
(PDF)

**S2 Table. Geological and paleontological data for different uppermost Cretaceous vertebrate sites of Haţeg Island, discussed in the text.**
(XLSX)

## Acknowledgments

We would like to thank the three reviewers for their constructive and helpful comments and suggestions which have greatly improved our resubmitted manuscript. We are grateful to the 2019–2025 field crews for their assistance in the Vălioara fieldwork, and especially to Attila Ősi (ELTE), Péter Gulyás (Mining Museum in Ajka) and Gábor Falusi (VDRG). We thank the following colleagues for their help provided during this study: Gáspár Albert (ELTE), László Makádi (SZTFH), Marcell Lebán (ELTE), Pál Pelikán† (MÁFI), Cristian Ciobanu (Haţeg Country UNESCO Global Geopark), Tibor Pecsics (VDRG), Martin Segesdi (ELTE), Maria Raluca Văcărescu (University of Bucharest), Miklós Szekeres (Budapest). This is HUN-REN-MTM-ELTE Paleo contribution No. 429.

## Author contributions

**Conceptualization:** Gábor Botfalvai.

**Data curation:** Gábor Botfalvai, Zoltán Csiki-Sava, János Magyar, Barna Páll-Gergely, Levente Koczó, Daniel Ţabără, Gergő Konecsni, Soma Budai.

**Formal analysis:** Gábor Botfalvai, Soma Budai.

**Funding acquisition:** Gábor Botfalvai.

**Investigation:** Gábor Botfalvai, Zoltán Csiki-Sava, János Magyar, Barna Páll-Gergely, Daniel Ţabără, Gergő Konecsni, Soma Budai.

**Methodology:** Gábor Botfalvai, Zoltán Csiki-Sava, Soma Budai.

**Project administration:** Gábor Botfalvai.

**Resources:** Gábor Botfalvai, Zoltán Csiki-Sava.

**Supervision:** Gábor Botfalvai.

**Validation:** Gábor Botfalvai.

**Visualization:** Gábor Botfalvai, Zoltán Csiki-Sava, János Magyar, Levente Koczó, Gergő Konecsni, Soma Budai.

**Writing – original draft:** Gábor Botfalvai, Zoltán Csiki-Sava, János Magyar, Barna Páll-Gergely, Daniel Ţabără, Soma Budai.

**Writing – review & editing:** Gábor Botfalvai, Zoltán Csiki-Sava.

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
