## [Decision Letter · Decision Letter 0]

18 Aug 2025

Dear Dr. Gábor,

Thank you for submitting your manuscript to PLOS ONE. After careful consideration, we feel that it has merit but does not fully meet PLOS ONE’s publication criteria as it currently stands. Therefore, we invite you to submit a revised version of the manuscript that addresses the points raised during the review process.

We look forward to receiving your revised manuscript.

Kind regards,

Jun Liu

Academic Editor

PLOS ONE

Journal Requirements:

2. In your manuscript, please provide additional information regarding the specimens used in your study. Ensure that you have reported human remain specimen numbers and complete repository information, including museum name and geographic location.

For more information on PLOS One's requirements for paleontology and archeology research, see https://journals.plos.org/plosone/s/submission-guidelines#loc-paleontology-and-archaeology-research .

“GB: Field and laboratory work was supported by Hungarian Scientific Research Fund and National Research, Development and Innovation Office (Grants no. NKFIH OTKA PD 131557 and FK 146097 to GB)

 ZCs-S: This research was also supported by a grant of the Romanian Ministry of Research, Innovation and Digitization, CNCS/CCCDI - UEFISCDI, project number PN-III-P4-ID-PCE-2020-2570, within PNCDI III (to ZCs-S)”

“We are grateful to the 2019-2025 field crews for their assistance in the Vălioara fieldwork, and especially to Attila Ősi (ELTE), Péter Gulyás (Mining Museum in Ajka) and Gábor  Falusi (VDRG). Field and laboratory work was supported by Hungarian Scientific Research Fund and National Research, Development and Innovation Office (Grants no. NKFIH OTKA PD 131557 and FK 146097 to GB),”

“GB: Field and laboratory work was supported by Hungarian Scientific Research Fund and National Research, Development and Innovation Office (Grants no. NKFIH OTKA PD 131557 and FK 146097 to GB)

 ZCs-S: This research was also supported by a grant of the Romanian Ministry of Research, Innovation and Digitization, CNCS/CCCDI - UEFISCDI, project number PN-III-P4-ID-PCE-2020-2570, within PNCDI III (to ZCs-S)”

6. We note that Figure 1 in your submission contain [map/satellite] images which may be copyrighted. All PLOS content is published under the Creative Commons Attribution License (CC BY 4.0), which means that the manuscript, images, and Supporting Information files will be freely available online, and any third party is permitted to access, download, copy, distribute, and use these materials in any way, even commercially, with proper attribution. For these reasons, we cannot publish previously copyrighted maps or satellite images created using proprietary data, such as Google software (Google Maps, Street View, and Earth). For more information, see our copyright guidelines: http://journals.plos.org/plosone/s/licenses-and-copyright.

1. You may seek permission from the original copyright holder of Figure 1to publish the content specifically under the CC BY 4.0 license. 

**Additional Editor Comments:**

I tryied to find experts from different aspectes but fall so nobody check the flora and invertebrate list. The reviewers are vertebrate paleontologists and sedimentologist. After reading the comments from three reviewers, your work need a minor revision.

Reviewers' comments:

Reviewer's Responses to Questions

**Comments to the Author**

1. Is the manuscript technically sound, and do the data support the conclusions?

Reviewer #1: Yes

Reviewer #2: Yes

Reviewer #3: Yes

2. Has the statistical analysis been performed appropriately and rigorously?

Reviewer #1: Yes

Reviewer #2: Yes

Reviewer #3: Yes

3. Have the authors made all data underlying the findings in their manuscript fully available?

Reviewer #1: Yes

Reviewer #2: Yes

Reviewer #3: Yes

4. Is the manuscript presented in an intelligible fashion and written in standard English?

Reviewer #1: Yes

Reviewer #2: Yes

Reviewer #3: Yes

Reviewer #1: The authors have done an excellent job documenting the vertebrate fossil assemblage and the taphonomy of the site K2 in Hateg Basin. The fossil assemblage record is exceptionally detailed. The authors also provided preliminary taxonomic identifications for various vertebrate fossils, supported by meticulous morphological descriptions. However, the quality of the figures does not fully match the descriptive text; the photos of the fossil are often too small to show the morphological details of the bones described.

The taphonomic analyses of the fossil assemblage are also highly detailed. The taphonomic and paleoecological inferences of the site K2 and Hateg Basin are robust and enjoyable to read, but I must admit that sedimentology and taphonomy are not my specialty. The authors successfully reconstructed the paleoenvironment, elucidated the taphonomic process, and provided significant insights into faunal evolutionary patterns and paleoecological dynamics of the Hateg Island fauna during the Maastrichtian. Most fossil vertebrate groups from the site K2 are effectively compared with counterparts from other fossil sites in the basin. However, comparisons for theropods and pterosaurs appear to be lacking; incorporating such information for these taxa would strengthen the study.

I also recommend a revision of the supplemental materials. Some inconsistencies create confusion: Table S2 and Table S3 both list 7 theropod records, but the details provided for these theropod fossils do not perfectly align between the tables or with the main text. Specifically, the main text mentions only 6 theropod elements (3 teeth, 2 limb bones, and 1 vertebra), while the supplementary tables list 7 records, including a theropod fibula that is not mentioned in the main text. Clarifying these discrepancies would improve the supplemental data's reliability.

A few minor formatting issues were noted in the manuscript. For instance, a space is missing in line 630, and the content in Table 4 is not correctly aligned.

Reviewer #2: This is an exceptional study of a bonebed. The inclusion of palynological data, description of invertebrates, and detailed sedimentological studies along with the description of the vertebrates makes this an outstanding example of this kind of research. I only have a few suggestions to consider. One is that I recommend that each of the crocodile tooth morphotypes that is described be figured, or a reference be given to another paper in which the morphotype is figured. Also, since the specimens are mostly fairly dark, I recommend considering formatting the figures with a white, rather than black, background. This is not a requirement, just a suggestion to consider.

The writing is excellent. I only saw one grammatical issue in the manuscript: line 2320: "dominated by far by macrovertebrate remains". I had to read this a couple of times to understand what was being said. I think it would be more effective just to say "dominated by macrovertebrate remains"

Also one formatting suggestion: when referring to a figure in another manuscript, eg, lines 406 and 841, use lower case "f" for "fig" so it is clear that it is not a figure in this paper that is being referred to.

Reviewer #3: This manuscript presents integrated sedimentological, palynological, paleontological investigations on the fossil-bearing Late Cretaceous Densuş-Ciula Formation in Haţeg Basin, western Romania. Voluminous fossils were excavated from the bonebed that deposited in a fluvial facies during the Maastrichtian stage. Based on the data, the authors reconstruct the palaeo-environment and the depositional processes of the bonebed. This work is undoubtedly reinforces the studies on the sedimentology and stratigraphy of the fossil sites as well as the faunal content/diversity in this area, it is probably suitable for publication in the journal of PLOS ONE. There are some minor concerns, in my opinion that need to be addressed before it’s publication.

1. I suggest the authors to add “N” and “E” to the latitude and longitude in Figure 1, and add more details to the stratigraphic column, to make it more clearly and more professional. If possible, please add some outcrop photos to show the contact between the sedimentary rocks and the volcanic units. Please also mark the sample names in A, and mark the fossils and sample names in the column.

2. Lines 285-339. Since the authors described the sedimentary facies, representative thin section photos under microscope might be needed here to show the readers more detailed lithological features of the bonebeds.

3. Lines 1473-1478. If possible, please provide some radiometric dating evidence on the Densuş-Ciula Formation in Haţeg Basin.

4. According to the description of different degree of abrasion of the fossils, and the relatively quick accumulations of the fossils, it raised me an hypothesis that these fossils might be transported by a pyroclastic flow that deposited within a delta due to the sudden decrease in current energy. Why there was a sudden drop in fluvial transport energy? Were there any volcanic activities nearby to provide the pyroclastic flow? If so, it will be very interesting for better understanding the mass-death event and the taphonomy.

5. Some of the discussion should be moved to the supplementary materials, just keep some key points.

**Do you want your identity to be public for this peer review?** For information about this choice, including consent withdrawal, please see our Privacy Policy

Reviewer #1: No

Reviewer #2: No

Reviewer #3: No

---

## [Author Response · Author response to Decision Letter 1]

14 Oct 2025

Comments of Reviewer 1

The quality of the figures does not fully match the descriptive text; the photos of the fossil are often too small to show the morphological details of the bones described.

In agreement with the reviewer's request, we have increased the size of the fossils shown in the figures (e.g. Figure 8). In addition, we have attached a supplementary file (S3 File) containing larger versions of the photographs of the fossils discussed in detail in the manuscript. We believe that these modifications are sufficient to make all anatomical details observable.

Most fossil vertebrate groups from the site K2 are effectively compared with counterparts from other fossil sites in the basin. However, comparisons for theropods and pterosaurs appear to be lacking; incorporating such information for these taxa would strengthen the study.

As aptly pointed out here by Reviewer 1, in the previous version of the MS we have indeed omitted to address the theropods and pterosaurs in this part of the discussion. In order to address and mitigate this shortcoming, we have now inserted two new sections discussing the significance of these two groups, as well, albeit at very different lengths given the very different taxonomic and distributional information available for them, both locally at site K2 and on a regional level. These new sections were inserted at lines 2178, respectively 2211 of the previous version (see 5.3 subchapter of the revised version), clarifying in some details the relevance and significance of these two groups for local faunal evolution.

I also recommend a revision of the supplemental materials. Some inconsistencies create confusion: Table S2 and Table S3 both list 7 theropod records, but the details provided for these theropod fossils do not perfectly align between the tables or with the main text. Specifically, the main text mentions only 6 theropod elements (3 teeth, 2 limb bones, and 1 vertebra), while the supplementary tables list 7 records, including a theropod fibula that is not mentioned in the main text. Clarifying these discrepancies would improve the supplemental data's reliability.

Thank you for pointing this out, as it was indeed an mistake in the manuscript. We have corrected the data related to theropod material and amended this in the manuscript and all tables (S2 Table and S3 File). The Theropoda material known from the site K2 consists of three teeth, three limb fragments, and one caudal vertebra.

A few minor formatting issues were noted in the manuscript. For instance, a space is missing in line 630, and the content in Table 4 is not correctly aligned.

We have corrected these errors in the revised manuscript.

Comments of Reviewer 2

I recommend that each of the crocodile tooth morphotypes that is described be figured, or a reference be given to another paper in which the morphotype is figured. Also, since the specimens are mostly fairly dark, I recommend considering formatting the figures with a white, rather than black, background. This is not a requirement, just a suggestion to consider.

Following up on the suggestion of Reviewer 2 (but also partly addressing a related one by Reviewer 1; see above) we have decided to split the previous figure 8 that also contained some of the crocodyliform teeth from site K2, into two distinct figures, removing all crocodyliform teeth from former figure 8 and adding a new figure (current figure 9) dedicated exclusively to the diverse corocdyliform tooth sample. This change allowed us both to present larger/more detailed images of other isolated vertebrate remains (current figure 8) and to document graphically all the crocodyliform morphotypes discussed in the manuscript text (current figure 9) – as indicated by the reviewer. We also headed its suggestion concerning the output style of the figures, and used white background to the new Figure 9 (as well).

I only saw one grammatical issue in the manuscript: line 2320: "dominated by far by macrovertebrate remains". I had to read this a couple of times to understand what was being said. I think it would be more effective just to say "dominated by macrovertebrate remains" Also one formatting suggestion: when referring to a figure in another manuscript, eg, lines 406 and 841, use lower case "f" for "fig" so it is clear that it is not a figure in this paper that is being referred to.

Thank you for your comments. We have corrected the problems in the revised manuscript.

Comments of Reviewer 3

I suggest the authors to add “N” and “E” to the latitude and longitude in Figure 1.

We have modified Figure 1 in the modified MS based on the reviewer's suggestions.

Add more details to the stratigraphic column, to make it more clearly and more professional. If possible, please add some outcrop photos to show the contact between the sedimentary rocks and the volcanic units. Please also mark the sample names in A, and mark the fossils and sample names in the column.

We believe that in it is original form the figure was misleading and it led to misinterpretation. Figure panel A shows the stratigraphy of the entire terrestrial unit cropping out in the Vălioara study area, amounting to ca. 1600 m in total thickness based on the work of Albert et al., 2025; while panel B shows the ca. 10 m thick studied interval at the Neagului Valley, near site K2. The figure was modified to show the difference in scale and to indicate the position of the studied section in the more comprehensive terrestrial unit more precisely.

Volcanic/volcaniclastic units are not present neither in the outcrop nor in its vicinity. We believe that the color of regional erosional surfaces in the original figure was misleading as it had a similar color to that used to indicate ‘Pyroclasts’. This detail has now been changed. Furthermore, the figure was complemented with two photos describing the sedimentology of the studied outcrop.

Lines 285-339. Since the authors described the sedimentary facies, representative thin section photos under microscope might be needed here to show the readers more detailed lithological features of the bonebeds.

The nature of the outcrop does not facilitate the creation of thin sections, due to the wetness and friable texture of the studied sediments. The bonebed is thoroughly bioturbated and very heterolithic, and thus we do not think thin sections would reveal any special texture that would add to the original interpretation. In addition, the original bonebed horizon is no longer accessible, so thin section sampling is currently not possible.

Lines 1473-1478. If possible, please provide some radiometric dating evidence on the Densuş-Ciula Formation in Haţeg Basin.

Although we have previously mentioned the existence of such radiometric evidence in the text of the MS (e.g., lines 1949-1953, 1958-1959), this was done indirectly, with reference to previously published studies. In order to address the suggestion of Reviewer 3 in this regard, however, we have decided to introduce at the highlighted part of the MS a brief new section with more details on the relevant, currently existing radiometric data (again, with references to previously published work), emphasizing the geochronological information that supports a ‘middle’ Campanian age of the beginning of the Densuș-Ciula Formation deposition. This section reads: “Most significantly in this respect, several recently acquired zircon U-Pb radiometric ages derived from primary andesitic-dacitic clasts recovered from volcaniclastic deposits belonging to the Lower Member of the Densuș-Ciula Formation [36] as well as from detritic zircon samples that originate from tuffitic intercalations in the basalmost part of the continental Densuș-Ciula succession [20] fall into the latest early to earliest late Campanian time interval (roughly 81 to 78 Ma), in good accordance with different marine micropaleontological and palynological biostratigraphic constraints currently available from both the continental and the underlying marine beds [11,20,32,35,53].”

According to the description of different degree of abrasion of the fossils, and the relatively quick accumulations of the fossils, it raised me an hypothesis that these fossils might be transported by a pyroclastic flow that deposited within a delta due to the sudden decrease in current energy. Why there was a sudden drop in fluvial transport energy? Were there any volcanic activities nearby to provide the pyroclastic flow? If so, it will be very interesting for better understanding the mass-death event and the taphonomy.

There is no geological data that would indicate a correlation between the K2 deposits and some sort of volcanic activity. The sedimentological profile of the exposed sediments at the site (see Fig. 3B) does not exhibit any indications of pyroclastic material. The youngest (highest-lying) pyroclastic beds are located approximately 500 m lower in the local stratigraphy, thus their occurrence cannot be associated in any way with the bone accumulations observed at site K2 (see Fig. 3A).

According to our interpretation, the energy of fluvial transport suddenly decreased at site K2 site because here the river flowed and discharged into a lake. This marked shift in local hydrodynamics precipitated the rapid accumulation of sediments (and, together with these, that of the transported bones), thereby leading to the formation of a vertebrate fossil-bearing lakeside delta environment.

Some of the discussion should be moved to the supplementary materials, just keep some key points.

It is acknowledged that the discussion in the present article is quite extensive, but it is believed that its contents are necessary in order to address the main objectives of the study, as these were laid out in the introduction. Transferring particular elements of the discussion to various supplementary materials would compromise the comprehensibility of the results, as many of the details discussed in its different parts are intimately interconnected. Nonetheless, the chapter is divided into subsections, thus enabling the reader to skip different parts if and as required.

---

## [Editor Report · Decision Letter 1]

19 Oct 2025

Paleontological and paleoecological significance of the oldest highly productive Upper Cretaceous (lowermost Maastrichtian) bonebed of Haţeg Basin (western Romania; Densuş-Ciula Formation)

PONE-D-25-38937R1

Dear Dr. Gábor,

We’re pleased to inform you that your manuscript has been judged scientifically suitable for publication and will be formally accepted for publication once it meets all outstanding technical requirements.

Kind regards,

Jun Liu

Academic Editor

PLOS ONE
---

## [Editor Report · Acceptance letter]

PONE-D-25-38937R1

PLOS ONE

Dear Dr. Botfalvai,

I'm pleased to inform you that your manuscript has been deemed suitable for publication in PLOS ONE. Congratulations! Your manuscript is now being handed over to our production team.

Kind regards,

on behalf of

Dr. Jun Liu

Academic Editor

PLOS ONE